# qNBO: QUASI-NEWTON MEETS BILEVEL OPTIMIZATION

**Sheng Fang**[12], **Yong-Jin Liu**[23], **Wei Yao**[54], **Chengming Yu**[56], **Jin Zhang**[457*]

[1]Public Health School, Fujian Medical University, [2]School of Mathematics and Statistics, Fuzhou University, [3]Center for Applied Mathematics of Fujian Province, [4]Department of Mathematics, SUSTech, [5]National Center for Applied Mathematics Shenzhen, SUSTech, [6]School of Science, BUPT, [7]CETC Key Laboratory of Smart City Modeling Simulation and Intelligent Technology, The Smart City Research Institute of CETC

shengfang1996@163.com, yjliu@fzu.edu.cn
yaow@sustech.edu.cn, 2024010618@bupt.cn, zhangj9@sustech.edu.cn

## ABSTRACT

Bilevel optimization, addressing challenges in hierarchical learning tasks, has gained significant interest in machine learning. The practical implementation of the gradient descent method to bilevel optimization encounters computational hurdles, notably the computation of the exact lower-level solution and the inverse Hessian of the lower-level objective. Although these two aspects are inherently connected, existing methods typically handle them separately by solving the lower-level problem and a linear system for the inverse Hessian-vector product. In this paper, we introduce a general framework to address these computational challenges in a coordinated manner. Specifically, we leverage quasi-Newton algorithms to accelerate the resolution of the lower-level problem while efficiently approximating the inverse Hessian-vector product. Furthermore, by exploiting the superlinear convergence properties of BFGS, we establish the non-asymptotic convergence analysis of the BFGS adaptation within our framework. Numerical experiments demonstrate the comparable or superior performance of the proposed algorithms in real-world learning tasks, including hyperparameter optimization, data hyper-cleaning, and few-shot meta-learning.

## 1 INTRODUCTIONS

Bilevel optimization (BLO), which addresses challenges in hierarchical decision process, has gained significant interest in many real-world applications. Typical applications in machine learning include meta-learning (Franceschi et al., 2018; Rajeswaran et al., 2019), hyperparameter optimization (Pedregosa, 2016; Franceschi et al., 2017; Okuno et al., 2021), adversarial learning (Goodfellow et al., 2014; Pfau & Vinyals, 2016), neural architecture search (Chen et al., 2019; Elsken et al., 2020), and reinforcement learning (Yang et al., 2019; Hong et al., 2023). In this study, we revisit the following BLO problem:

$$\min_{x \in \mathbb{R}^m} \quad \Phi(x) := F(x, y^*(x)) \quad \text{s.t.} \ y^*(x) = \arg\min_{y \in \mathbb{R}^n} f(x, y), \tag{1}$$

in which the upper-level (UL) objective function $F : \mathbb{R}^m \times \mathbb{R}^n \to \mathbb{R}$ is generally nonconvex, while the lower-level (LL) objective function $f : \mathbb{R}^m \times \mathbb{R}^n \to \mathbb{R}$ is strongly convex with respect to (*w.r.t.*) the LL variable $y \in \mathbb{R}^n$.

The gradient of $\Phi(x)$, known as hypergradient, is crucial not only for applying gradient descent but also for developing accelerated gradient-based methods for BLO problems. Therefore, a fundamental question in solving BLO problems is how to efficiently estimate the hypergradient. Assuming that $f$ is continuously twice differentiable, and by applying the chain rule and utilizing the first-order optimality condition $\nabla_y f(x, y^*(x)) = 0$ of the LL optimization problem (Ghadimi & Wang, 2018), the hypergradient is given by

$$\nabla \Phi(x) = \nabla_x F(x, y^*(x)) - [\nabla_{xy}^2 f(x, y^*(x))]^T [\nabla_{yy}^2 f(x, y^*(x))]^{-1} \nabla_y F(x, y^*(x)). \tag{2}$$

---

*Correspondence to Jin Zhang (zhangj9@sustech.edu.cn)

Two main challenges in estimating the hypergradient are: (**C1**) evaluating the LL solution $y^*(x)$; (**C2**) estimating the Jacobian-inverse Hessian-vector product $[\nabla^2_{xy}f(x,y)]^T[\nabla^2_{yy}f(x,y)]^{-1}\nabla_y F(x,y)$, once a good proxy $y$ for the LL solution $y^*(x)$ is obtained.

For (C1), the common approach is to perform a few additional gradient descent steps for the LL problem on the current estimate $y_k$, using it as a proxy for the LL solution. For (C2), two main approaches have been proposed in the literature. The first is to estimate the inverse Hessian using the (truncated) Neumann series (Ghadimi & Wang, 2018; Ji et al., 2021). The second approach is to compute the inverse Hessian-vector product $[\nabla^2_{yy}f(x,y)]^{-1}\nabla_y F(x,y)$ by solving the linear system $[\nabla^2_{yy}f(x,y)]z = \nabla_y F(x,y)$ for $z$, and then calculating $[\nabla^2_{xy}f(x,y)]^T z$ (Pedregosa, 2016; Arbel & Mairal, 2022; Dagréou et al., 2022). Clearly, the hypergradient approximation error depends on the errors in both (C1) and (C2). Most existing methods handle (C1) and (C2) separately, using different techniques.

A notable exception is the recent breakthrough by (Ramzi et al., 2022), which introduces a novel approach (named SHINE), specifically designed for deep equilibrium models (DEQs) (Bai et al., 2019; 2020) and BLO problems where the UL objective function does not explicitly depend on the UL variable, *i.e.*, $F(x,y)=\mathcal{L}(y)$. The novelty of SHINE lies in its approach to addressing (C1) and (C2) closely. The main idea is to use quasi-Newton (qN) matrices from the LL solution process to efficiently approximate the inverse Hessian in the direction needed for the hypergradient computation. SHINE provides three methods for approximating the hypergradient by incorporating a technique OPA with Broyden's method and BFGS. Note that in the OPA method from Ramzi et al. (2022), the qN matrices derived from the LL resolution are influenced by additional updates. This can potentially lead to incorrect inversion, as noted in Ramzi et al. (2022). To mitigate this issue, they employ a fallback strategy. In theory, SHINE demonstrates asymptotic convergence to the hypergradient under various conditions but does not guarantee convergence of the algorithmic iterates.

Therefore, inspired by SHINE, our focus is on improving hypergradient approximation and reducing barriers to solving BLO problems. In particular, our main research question is: *Can we develop a method to enhance hypergradient approximation for solving the bilevel optimization problem in (1) with a guaranteed convergence rate?*

## 1.1 MAIN CONTRIBUTIONS

Our response to this question is affirmative, and our main contributions are summarized below.

- qNBO, a new algorithmic framework utilizing quasi-Newton techniques, is proposed for solving the BLO problem (1). Unlike SHINE, qNBO includes a subroutine that applies quasi-Newton recursion schemes specifically tailored for the direction $\nabla_y F(x,y)$, avoiding incorrect inversion.
- We validate the effectiveness and efficiency of qNBO with two practical algorithms: qNBO (BFGS) and qNBO (SR1), corresponding to two prominent quasi-Newton methods. The numerical results demonstrate qNBO's comparable or superior performance compared to its closest competitors, SHINE (Ramzi et al., 2022) and PZOBO (Sow et al., 2022b), as well as other BLO algorithms, including two popular fully first-order methods: BOME (Liu et al., 2022) and F$^2$SA (Kwon et al., 2023).
- By leveraging the superlinear convergence rates of BFGS, we analyze the non-asymptotic convergence of BFGS adaptation within our framework, qNBO.

## 1.2 ADDITIONAL RELATED WORK

**Quasi-Newton methods.** Because of the low computation cost per iteration and fast convergence rate, quasi-Newton (qN) methods has been extensively studied (Nocedal & Wright, 2006). The most common qN methods are the Broyden-Fletcher-Goldfarb-Shanno (BFGS) (Broyden, 1970b;a; Fletcher, 1970; Goldfarb, 1970; Shanno, 1970), its low-memory extension L-BFGS (Liu & Nocedal, 1989), and the symmetric rank one method (SR1) (Davidon, 1991; Broyden, 1967). For BLO problems, Pedregosa (2016) first uses L-BFGS to solve the LL problem to a certain tolerance. Then a conjugate-gradient method is applied to solve the linear system $[\nabla^2_{yy}f(x,y)]z = \nabla_y F(x,y)$ through matrix-vector products. Finally, $[\nabla^2_{xy}f(x,y)]^T z$ is calculated.

**Hypergradient approximation methods.** Various bilevel methods have been proposed recently to approximate the inverse Hessian or omit some second-order derivative computations in the hypergradient. For example, FOMAML (Finn et al., 2017; Nichol et al., 2018) skips calculating all second-order derivatives. DARTS (Liu et al., 2018) solves the LL problem with just one gradient descent step. The Jacobian-Free method (JFB) (Fung et al., 2022) approximates the inverse Hessian with the identity. Giovannelli et al. (2021) proposes practical low-rank bilevel methods (BSG1 and BSG-N-FD) that use first-order approximations for second-order derivatives through a finite-difference scheme or rank-1 approximations. Recently, several zeroth-order methods have been proposed to approximate the full hypergradient, such as ES-MAML (Song et al., 2019) and HOZOG (Gu et al., 2021). Another zeroth-order method, PZOBO (Sow et al., 2022b), approximates only part of the hypergradient by comparing two optimization paths.

There is another line of research for BLO problems, which does not explicitly use the hypergradient in (2), see, *e.g.*, Liu et al. (2023); Sow et al. (2022a); Shen & Chen (2023); Liu et al. (2022); Kwon et al. (2023; 2024). These algorithms first use the value function of the lower-level problem to transform the bilevel problem into a single-level problem. Then, they apply the penalty function method or other techniques to solve the reformulated problem. For instance, BOME (Liu et al., 2022) is a novel and fast gradient-based method that uses a modified dynamic barrier gradient descent on the value-function reformulation. F$^2$SA (Kwon et al., 2023) is a fully first-order method developed from a value function-based penalty formulation. It can be implemented in a single-loop manner. Additionally, it is shown in Kwon et al. (2023) that the update direction of F$^2$SA has a global $\mathcal{O}(1/\lambda)$-approximability of the exact hypergradient, where $\lambda$ is the penalty parameter.

## 2 PROPOSED FRAMEWORK

In this section, we introduce a general framework, qNBO, to enhance hypergradient approximation. It addresses the computational challenges (C1) and (C2) using quasi-Newton techniques. We begin by rewriting the hypergradient as:

$$\nabla \Phi(x) = \nabla_x F(x, y^*(x)) - [\nabla^2_{xy} f(x, y^*(x))]^T u^*(x, y^*(x)),$$

where $u^*(x, y) := [\nabla^2_{yy} f(x, y)]^{-1} \nabla_y F(x, y)$. As in Arbel & Mairal (2022) and Dagréou et al. (2022), the proposed algorithms introduce an additional variable $u_k$ alongside $x_k$ and $y_k$. Naturally, qNBO consists of three components. The details of qNBO are presented in Algorithm 1.

---

**Algorithm 1** qNBO : quasi-Newton Meets Bilevel Optimization

---

**Input:** $x_0, y_0$; initial matrix $H_0$; stepsize $\alpha > 0$; iterates numbers $K, \{Q_k\}_{k=0}^{K-1}$
**for** $k = 0, 1, \ldots, K - 1$ **do**
    1. $y_{k+1} \leftarrow \mathcal{A}(x_k, y_k)$ # update $y_{k+1}$ by a subroutine $\mathcal{A}$
    2. if $Q_k = 1$:
        $u_{k+1} \leftarrow \mathcal{C}_{qN}\big(\nabla_y F(x_k, y_{k+1}), H_0, \{s_t, g_t\}_{t=0}^{T-1}\big)$ # share $\{s_t, g_t\}_{t=0}^{T-1}$ with $\mathcal{A}(x_k, y_k)$
        else:
        $u_{k+1} \leftarrow \mathcal{B}(x_k, y_{k+1}, H_0, \nabla_y F(x_k, y_{k+1}), Q_k)$ # improve $u_{k+1}$ by a subroutine $\mathcal{B}$
    3. $x_{k+1} \leftarrow x_k - \alpha \big(\nabla_x F(x_k, y_{k+1}) - [\nabla^2_{xy} f(x_k, y_{k+1})]^T u_{k+1}\big)$
**end for**

---

**Part 1:** qNBO updates $y_k$ towards $y^*(x_k)$ using a qN-based subroutine $\mathcal{A}(x_k, y_k)$ in Algorithm 2, starting from $y_k$. The key is a quasi-Newton recursion scheme $\mathcal{C}_{qN}$, which computes the matrix-vector product $Hd$ by performing a sequence of inner products and vector summations involving $d$ and the pairs $\{s_i, g_i\}_{i=0}^{t-1}$. Here $H$ represents the inverse Hessian approximation of the LL objective, $d = \nabla_y f(x, y_t)$, $s_i = y_{i+1} - y_i$, and $g_i = \nabla_y f(x, y_{i+1}) - \nabla_y f(x, y_i)$ in this subroutine. Two prominent quasi-Newton recursion schemes are provided in Appendix B.

**Part 2:** To update $u_{k+1}$, we provide two options: $Q_k = 1$ or $Q_k > 1$. In the case of $Q_k = 1$, qNBO updates $u_{k+1}$ similarly to SHINE. The pairs $\{s_i, g_i\}_{i=0}^{T-1}$ from $\mathcal{A}(x_k, y_k)$ are shared to approximate the inverse Hessian in the direction $\nabla_y F(x_k, y_{k+1})$. Unfortunately, incorrect inversion may occur because the pairs $\{s_i, g_i\}_{i=0}^{T-1}$ in $\mathcal{A}(x_k, y_k)$ are designed to satisfy the secant equation $H_{t+1} g_t = s_t$. To address this issue, qNBO adds a subroutine $\mathcal{B}$ in Algorithm 3, which uses quasi-Newton recursion

schemes for the direction $\nabla_y F(x_k, y_{k+1})$ when $Q_k > 1$. The price to pay is that the pairs $\{s_i, g_i\}$ in $\mathcal{A}$ cannot be shared with $\mathcal{B}$, increasing the computational cost. Thus, choosing between $Q_k = 1$ and $Q_k > 1$ involves a trade-off between performance and computational cost.

**Part 3:** After computing $y_{k+1}$ and $u_{k+1}$, qNBO updates $x_{k+1}$ by

$$x_{k+1} = x_k - \alpha_k \left( \nabla_x F(x_k, y_{k+1}) - [\nabla^2_{xy} f(x_k, y_{k+1})]^T u_{k+1} \right),$$

where $\alpha_k$ is a stepsize, and $\tilde{\nabla}\Phi(x_k) := \nabla_x F(x_k, y_{k+1}) - [\nabla^2_{xy} f(x_k, y_{k+1})]^T u_{k+1}$ is a hypergradient approximation. This update rule for $x_k$ is commonly used in gradient-based algorithms, , such as those in Arbel & Mairal (2022) and Dagréou et al. (2022).

---

**Algorithm 2** $\mathcal{A}(x, y^0)$: gradient descent steps + qN steps for the LL problem

---

**Input:** $x, y^0$; initial matrix $H_0$; stepsizes $\beta, \gamma > 0$; iterates numbers $P, T$

  1. **for** $j = 0, 1, 2, \ldots, P - 1$
      $y^{j+1} \leftarrow y^j - \beta \nabla_y f(x, y^j)$
  **end for**
  2. $y_0 \leftarrow y^P$
    **for** $t = 0, \ldots, T - 1$
      $y_{t+1} \leftarrow y_t - \gamma d_t,$
      where $d_t \leftarrow \mathcal{C}_{qN}\left( \nabla_y f(x, y_t), H_0, \{s_i, g_i\}_{i=0}^{t-1} \right)$ $(t \geq 1)$, $d_0 \leftarrow H_0 \nabla_y f(x, y_0)$
      $s_t \leftarrow y_{t+1} - y_t, g_t \leftarrow \nabla_y f(x, y_{t+1}) - \nabla_y f(x, y_t)$
    **end for**
  Return $y_T, \{s_t, g_t\}_{t=0}^{T-1}$.

---

**Algorithm 3** $\mathcal{B}(x, y, H_0, d, Q)$

---

**Input:** $x, y$; initial matrix $H_0$; vector $d$; stepsize $\xi_i > 0$; iterates number $Q$

  1. $u_0 \leftarrow H_0 d, \tilde{s}_0 \leftarrow \xi_0 u_0$ and $\tilde{g}_0 \leftarrow \nabla_y f(x, y + \tilde{s}_0) - \nabla_y f(x, y)$
  2. **for** $i = 1, 2, \ldots, Q - 1$
    $u_i \leftarrow \mathcal{C}_{qN}\left( d, H_0, \{\tilde{s}_j, \tilde{g}_j\}_{j=0}^{i-1} \right)$
    $\tilde{s}_i \leftarrow \xi_i u_i, \tilde{g}_i \leftarrow \nabla_y f(x, y + \tilde{s}_i) - \nabla_y f(x, y)$
  **end for**
  Return $u_{Q-1}$.

---

**Two practical qNBO algorithms: qNBO (BFGS) and qNBO (SR1).** We describe two prominent quasi-Newton recursion schemes, $\mathcal{C}_{qN}$, for computing the inverse Hessian approximation-vector product $H_t d$. These schemes involve a sequence of inner products and vector summations with $d$ and pairs $\{s_i, g_i\}_{i=0}^{t-1}$. One is the BFGS two-loop recursion scheme, outlined in Algorithm 4, corresponding to the BFGS updating formula (Broyden, 1970b;a; Fletcher, 1970; Goldfarb, 1970; Shanno, 1970):

$$H_{t+1} = \left( I - \rho_t s_t g_t^T \right) H_t \left( I - \rho_t g_t s_t^T \right) + \rho_t s_t s_t^T, \quad \rho_t = \frac{1}{g_t^T s_t}. \tag{3}$$

The second algorithm is presented in Algorithm 5, which corresponds to the symmetric-rank-one (SR1) updating formula (Davidon, 1991; Broyden, 1967):

$$H_{t+1} = H_t + \frac{(s_t - H_t g_t)(s_t - H_t g_t)^T}{(s_t - H_t g_t)^T g_t}. \tag{4}$$

**Implementation and improvement.** Several details and enhancements are needed for an efficient implementation of qNBO.

First, the purpose of including a few gradient descent steps in subroutine $\mathcal{A}$ is to bring the iterators closer to a neighborhood of the LL solution, enabling superlinear convergence in subsequent quasi-Newton steps. A warm-start for $y^0$ is effective because $y^*(x_{k+1})$ remains close to $y^*(x_k)$ when $x_{k+1}$ is near $x_k$. This is guaranteed by the Lipschitz continuity of $y^*(x)$. In practice, we conjecture

that a few initial gradient descent steps are sufficient, although they are necessary for the theoretical analysis.

Second, because $f(x, y)$ exhibits strong convexity *w.r.t.* $y$ in our context, the curvature condition $s_t^T g_t > 0$, required for BFGS, is consistently satisfied. This allows the use of a fixed step size, eliminating the need for time-consuming line searches. Furthermore, as the solution approaches a region conducive to superlinear convergence, employing a few quasi-Newton steps is sufficient to achieve a satisfactory solution.

Third, in experiments, the initial matrix $H_0$ is chosen as a scalar multiple of the identity matrix. This simplification ensures that the recursion algorithms, Algorithms 4 and 5, involve only vector inner products, significantly reducing computational costs. In Algorithm 3, the parameter $\xi_i$ is typically set to either 1 or $\|u_i\|$ in most cases.

Fourth, qNBO is a flexible framework that can integrate other quasi-Newton methods, such as limited memory BFGS (L-BFGS) (Liu & Nocedal, 1989). It also supports a "non-loop" implementation of L-BFGS by representing quasi-Newton matrices in outer-product form (Byrd et al., 1994).

Finally, qNBO consists of three parts, with the first two utilizing quasi-Newton recursion schemes. A stochastic adaptation involves replacing these schemes with stochastic methods (e.g., K-BFGS (Goldfarb et al., 2020), Stochastic Block BFGS(Gower et al., 2016)) and using stochastic gradients in Part 3, aligning with Dagréou et al. (2022). Key challenges in implementing the stochastic adaptation include constructing effective unbiased or biased estimators in Part 3 using techniques like variance reduction and momentum, and analyzing the convergence rate and complexity of the proposed stochastic algorithms in a bilevel setting that leverages noisy second-order information. Addressing these theoretical issues may require breakthroughs beyond the scope of this work.

## 3 THEORETICAL ANALYSIS

In this section, we analyze the non-asymptotic convergence of the qNBO algorithm, as outlined in Algorithm 1, under standard assumptions commonly used in BLO literature (Ghadimi & Wang, 2018; Ji et al., 2021; Chen et al., 2022; Dagréou et al., 2022; Ji et al., 2022).

### 3.1 ASSUMPTIONS

**Assumption 3.1.** Assume that the UL objective function $F$ satisfies the following properties:

(i) For all $x$, the gradients $\nabla_x F(x, y)$ and $\nabla_y F(x, y)$ are Lipschitz continuous *w.r.t.* $y$, with Lipschitz constants $L_{F_x}$ and $L_{F_y}$, respectively.

(ii) For all $y$, $\nabla_y F(x, y)$ is Lipschitz continuous *w.r.t.* $x$, with a Lipschitz constant $\bar{L}_{F_y}$.

(iii) There exists a constant $C_{F_y}$ such that $\|\nabla_y F(x, y)\| \leq C_{F_y}$ for all $x, y$.

**Assumption 3.2.** Assume that the LL objective function $f$ has the following properties:

(i) For all $x$ and $y$, $f$ is continuously twice differentiable in $(x, y)$.

(ii) For all $x$, $f(x, y)$ is strongly convex *w.r.t.* $y$ with parameter $\mu > 0$. Moreover, $\nabla_y f(x, y)$ and $\nabla_{yy}^2 f(x, y)$ are Lipschitz continuous *w.r.t.* $y$ with parameter $L$ and $L_{f_{yy}}$, respectively.

(iii) For all $x$, $\nabla_{xy}^2 f(x, y)$ is Lipschitz continuous *w.r.t.* $y$ with constant $L_{f_{xy}}$.

(iv) For all $x, y$, we have $\|\nabla_{xy}^2 f(x, y)\| \leq M_{f_{xy}}$ for some constant $M_{f_{xy}}$.

(v) For all $y$, $\nabla_{xy}^2 f(x, y)$ and $\nabla_{yy}^2 f(x, y)$ are Lipschitz continuous *w.r.t.* $x$ with constants $\bar{L}_{f_{xy}}$ and $\bar{L}_{f_{yy}}$, respectively.

**Assumption 3.3.** There exists $\iota \in \mathbb{R}$ such that $\inf_x \Phi(x) \geq \iota$.

## 3.2 CONVERGENCE RESULTS

The evolving nature of inverse Hessian approximations through updating formulas in qNBO significantly complicates the analysis of non-asymptotic convergence. Additionally, various update formulas present different challenges, similar to studying the convergence rates of quasi-Newton methods. In this section, we focus on the non-asymptotic convergence of qNBO (BFGS), leveraging the superlinear convergence rates of BFGS. Some results can also be extended to qNBO (SR1). Comprehensive proofs of these results are provided in Appendix D.

Let $L_\Phi$ be the Lipschitz constant of $\nabla\Phi$ given in Lemma D.12. We first present the convergence results of qNBO (BFGS) for solving the bilevel problem (1), where the LL objective function is quadratic.

**Theorem 3.4** (quadratic case). *Suppose that $f$ in (1) takes the following quadratic form:*

$$f(x, y) = \frac{1}{2}y^T A y - y^T x, \tag{5}$$

*where $\mu I \preceq A \preceq LI$. Assume that Assumption 3.1 and 3.3 hold. Set $Q_k = k + 1$ and $H_0 = LI$. Let $\kappa := L/\mu$, $t_b := 4n\ln\kappa$, $c_t := 2t_b^{\frac{T}{2}}$, and $\omega := c_1(1 + \frac{1}{\varepsilon})c_t^2\kappa^3(\frac{1}{T})^T$, where $c_1$ is a positive constant given in Theorem D.19. We can choose positive parameters $\alpha$, $\varepsilon$ and $T$ such that $\tau := c_t^2\kappa^3(\frac{1}{T})^T\big((1 + \varepsilon) + (1 + \frac{1}{\varepsilon})\alpha^2 c_1\big) < 1$ and $\alpha L_\Phi + \omega\alpha^2\left(\frac{1}{2} + \alpha L_\Phi\right)\frac{1}{1-\tau} \leq \frac{1}{4}$. Then the iterates $x_k$ generated by qNBO (BFGS) satisfy:*

$$\frac{1}{K}\sum_{k=0}^{K-1}\|\nabla\Phi(x_k)\|^2 \leq \frac{4(\Phi(x_0) - \inf_x \Phi(x))}{\alpha K} + \frac{3\delta_0}{K(1-\tau)} + \frac{18nLC_{F_y}^2 \ln K}{\mu^3 K}, \tag{6}$$

*with the initial error $\delta_0 = 3c_t^2\kappa^3(\frac{1}{T})^T c_2\|y_0^* - y_0\|^2$, where $c_2$ is a constant.*

*Remark* 3.5. For the quadratic case, quasi-Newton methods achieve global superlinear convergence (Ye et al., 2023; Rodomanov & Nesterov, 2022; 2021b), allowing $P = 0$ in Algorithm 2.

*Remark* 3.6. The convergence rate of qNBO (SR1) for the quadratic case is similar to that in Theorem 3.4. However, for the general case, the qNBO (SR1) algorithm lacks convergence guarantees without specific corrections used to achieve numerical stability, as noted in Ye et al. (2023).

Next, we explore the case where the LL objective function in (1) takes a general form. While the convergence rate of qNBO (BFGS) resembles that of the previous quadratic case, it is much more challenging. The specific convergence rate for this general case is detailed in the following theorem.

**Theorem 3.7** (general case). *Suppose that Assumptions 3.1, 3.2 and 3.3 hold. Set $Q_k = k + 1$. Choose the parameters $\beta$ and $P$ such that $(1 - \beta\mu)^P\|y_k - y_k^*\| \leq \frac{1}{300\sqrt{\mu}}$, and assume $H_0$ satisfies: $\|\nabla_{yy}^2 f(x_k, y^*(x_k))^{-1/2}\big(H_0^{-1} - \nabla_{yy}^2 f(x_k, y^*(x_k))\big)\nabla_{yy}^2 f(x_k, y^*(x_k))^{-1/2}\|_F \leq \frac{1}{7}$. Define $\tau := \kappa(\frac{1}{T})^T(1 - \beta\mu)^P\big((1 + \varepsilon) + (1 + \frac{1}{\varepsilon})\alpha^2 c_3\big)$ and $\omega := c_3(1 + \frac{1}{\varepsilon})\kappa(\frac{1}{T})^T(1 - \beta\mu)^P$, with a constant $c_3$ given in Theorem D.24. We can choose positive parameters $\alpha$, $\varepsilon$ and $T$ such that $\tau < 1$ and $\alpha L_\Phi + \omega\alpha^2\left(\frac{1}{2} + \alpha L_\Phi\right)\frac{1}{1-\tau} \leq \frac{1}{4}$. Then the iterates $x_k$ generated by qNBO (BFGS) satisfy:*

$$\frac{1}{K}\sum_{k=0}^{K-1}\|\nabla\Phi(x_k)\|^2 \leq \frac{4(\Phi(x_0) - \inf_x \Phi(x))}{\alpha K} + \frac{3\delta_0}{K(1-\tau)} + \frac{18nLM_{f_{xy}}^2 C_{F_y}^2 \ln K}{\mu^3\tilde{\xi}K}, \tag{7}$$

*where $\delta_0 = 3\kappa(\frac{1}{T})^T(1 - \beta\mu)^P c_4\|y_0^* - y_0\|^2$ is the initial error with constant $c_4$. The constant $\tilde{\xi}$ is related to the property of $f$, as given in (29).*

The proof sketch of Theorem 3.7 can be found in Appendix D.2. The complete version of the parameter specifications and the proofs of Theorems 3.4 and 3.7 are provided in Appendices D.3 and D.4, respectively.

**Discussion on convergence rate and complexity.** Choose $T = \Theta(\ln\kappa)$ and the step size $\alpha = \Theta(\kappa^{-3})$ such that $\tau < 1$ and $\alpha L_\Phi + \omega\alpha^2\left(\frac{1}{2} + \alpha L_\Phi\right)\frac{1}{1-\tau} \leq \frac{1}{4}$. Under the same setting as Theorem 3.7, we have $\frac{1}{K}\sum_{k=0}^{K-1}\|\nabla\Phi(x_k)\|^2 = \mathcal{O}\left(\frac{\kappa^3}{K} + \frac{\kappa^3\ln K}{K}\right)$. To achieve an $\epsilon$-stationary point, it is required that $K = \tilde{\mathcal{O}}(\kappa^3\epsilon^{-1})$, resulting in a gradient complexity of $Gc(f, \epsilon) = \tilde{\mathcal{O}}(\kappa^6\epsilon^{-2})$ and $Gc(F, \epsilon) = \tilde{\mathcal{O}}(\kappa^3\epsilon^{-1})$, as well as a Jacobian-vector product complexity of $JV(\epsilon) = \tilde{\mathcal{O}}(\kappa^3\epsilon^{-1})$.

The details for ensuring $\tau < 1$ and $\alpha L_\Phi + \omega\alpha^2 \left(\frac{1}{2} + \alpha L_\Phi\right) \frac{1}{1-\tau} \leq \frac{1}{4}$, as well as the specific complexity analysis, can be found in Appendix E.

**Theoretical comparisons.** Our analysis provides a non-asymptotic convergence rate, superior to that of SHINE (Ramzi et al., 2022). It is established in Ji et al. (2022) that the fastest deterministic convergence rate, $\frac{1}{K} \sum_{k=0}^{K-1} \|\nabla\Phi(x_k)\|^2 = \mathcal{O}(\frac{1}{K})$, is achievable. In comparison, BOME (Liu et al., 2022) reaches a convergence rate of $\mathcal{O}(K^{-1/4})$, $F^2SA$ (Kwon et al., 2023) attains $\mathcal{O}(\frac{\ln K}{K^{2/3}})$, and SABA (Dagréou et al., 2022) achieves $\mathcal{O}(\frac{1}{K})$. To achieve an $\epsilon$-stationary point, the matrix-vector complexity for qNBO is $\tilde{\mathcal{O}}(\kappa^3 \epsilon^{-1})$, primarily due to the $[\nabla_{xy}^2 f(x,y)]^T u$ calculations. It is worth noting that the number of Jacobian-vector products and Hessian-vector products for AID-BIO (Ji et al., 2021) are of the orders $\mathcal{O}(\kappa^3 \epsilon^{-1})$ and $\mathcal{O}(\kappa^{3.5} \epsilon^{-1})$, respectively. Although the gradient complexity $Gc(f, \epsilon)$ for qNBO is higher than that in AID-BIO (Ji et al., 2021), the computation of gradients is generally less complex than performing matrix-vector operations.

## 4 NUMERICAL EXPERIMENT

In this section, we conduct numerical experiments to evaluate the performance of the qNBO algorithms in solving bilevel optimization problems. We first validate the theoretical convergence through experiments on a toy example, followed by an assessment of efficiency by comparing qNBO with its closest competitor, SHINE (Ramzi et al., 2022), as well as other bilevel optimization (BLO) algorithms such as AID-BIO (Ji et al. (2021) with CG method), AID-TN (Ji et al. (2021) with Truncated Neumann method), AMIGO (Arbel & Mairal (2022) with CG method), SABA (Dagréou et al., 2022) and BSG1 (Giovannelli et al., 2021). Additionally, we compare qNBO with two widely used fully first-order algorithms, BOME (Liu et al., 2022) and $F^2SA$ (Kwon et al., 2023), in real-world applications including hyperparameter optimization and data hyper-cleaning. Finally, we explore qNBO's applicability to complex machine learning tasks by exclusively comparing it with PZOBO (Sow et al., 2022b) in a meta-learning experiment, where PZOBO is regarded as the leading algorithm for few-shot meta-learning. Details of all experimental specifications are provided in Appendix C. Additionally, we perform an ablation study in Appendix C.5.

### 4.1 TOY EXAMPLE

In this section, we consider a quadratic bilevel problem where both the UL and LL objective functions are quadratic. Given $z_0 \in \mathbb{R}^n$ and the symmetric positive definite matrix $A \in \mathbb{R}^{n \times n}$, the problem is formulated as follows:

$$\min_{x \in \mathbb{R}^n} \frac{1}{2}\|x - z_0\|^2 + \frac{1}{2}y^*(x)^T A y^*(x) \;\text{ s.t. }\; y^*(x) = \arg\min_{y \in \mathbb{R}^n} \frac{1}{2}y^T A y - x^T y. \quad (8)$$

In the experiment, the vector $z_0$ and the matrix $A$ are randomly generated, with $n = 1000$. The hypergradient is given by $\nabla\Phi(x) = (A^{-1} + I)x - z_0$, which yields the unique solution $(x^*, y^*) = \left((A^{-1} + I)^{-1}z_0, A^{-1}(A^{-1} + I)^{-1}z_0\right)$. To evaluate the performance of various methods in solving this problem, we analyze the hypergradient estimation errors and the distances between the iterates $(x_k, y_k)$ and the optimal solution $(x^*, y^*)$. The results, shown in Figure 1(a), indicate that AID-BIO, AMIGO-CG and qNBO (SR1) achieve smaller hypergradient errors and produce iterates closer to the optimal solutions compared to other methods. Furthermore, we analyze the impact of parameter $\{Q_k\}_{k=0}^{K-1}$ in Algorithm 1 on the performance of qNBO (BFGS). As depicted in Figure 1(d) shows that the hypergradient $\nabla\Phi(x_k)$ generally decreases as $Q_k$ increases, with $\{Q_k\}_{k=0}^{K-1} = \{k + 1\}_{k=0}^{K-1}$ performing the best, thereby supporting the claims of Theorem 3.4.

### 4.2 HYPERPARAMETER OPTIMIZATION IN LOGISTIC REGRESSION

We perform hyperparameter optimization for $l_2$-regularized logistic regression on the 20News (Lang, 1995) and Real-sim (Chang & Lin) datasets, formulated as a bilevel problem:

$$\min_{x \in \mathbb{R}} \sum_{i=1}^{n'} \ell(a_i', b_i', y^*(x)) \;\text{ s.t. }\; y^*(x) = \arg\min_{y \in \mathbb{R}^n} \sum_{i=1}^{n} \ell(a_i, b_i, y) + \frac{\exp(x)}{2}\|y\|^2, \quad (9)$$

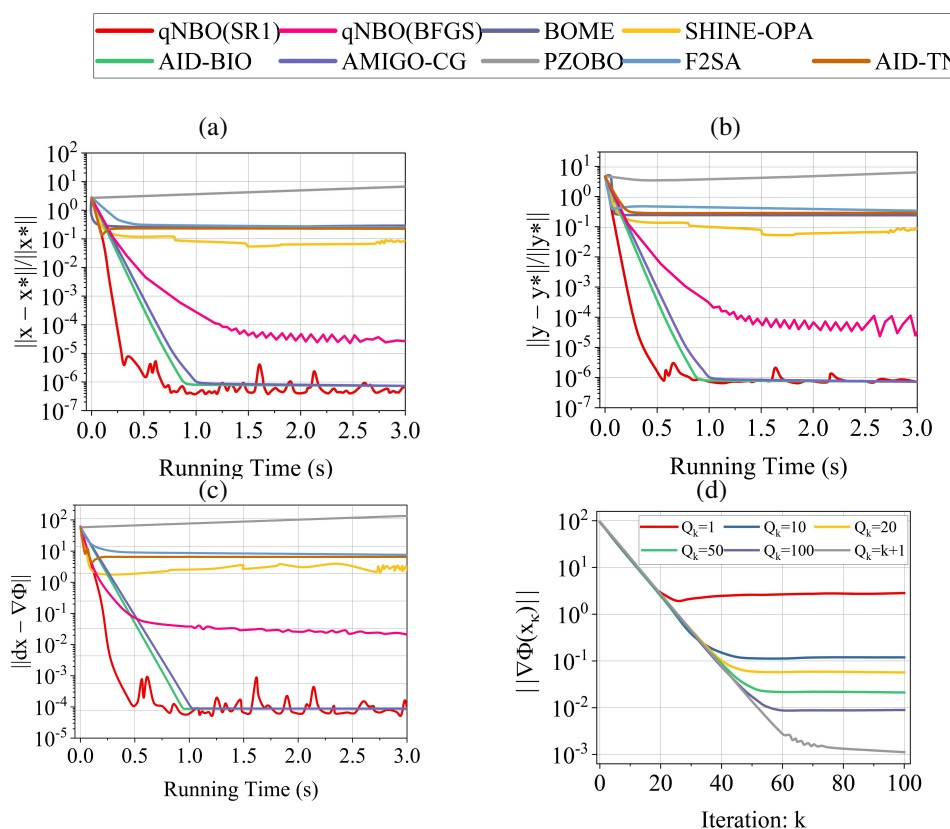

Figure 1: Numerical results on toy example. (d) Testing results on the impact of the parameter $\{Q_k\}_{k=0}^{K-1}$ in qNBO (BFGS).

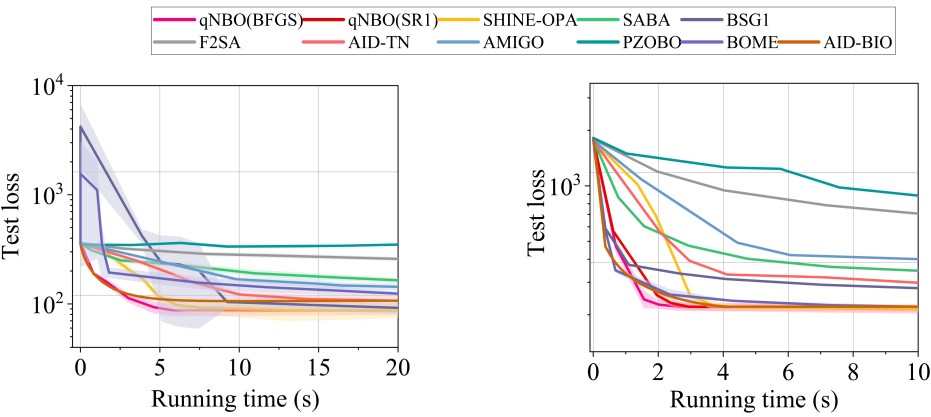

Figure 2: Hyperparameter optimization experiments for $l_2$-regularized logistic regression on two datasets (Left: **20News**; Right: **Real-sim**).

where $(a_i, b_i) \in \mathcal{D}_{train}$ and $(a_i', b_i') \in \mathcal{D}_{val}$ are the training data and validation data respectively, and $\ell(a_i, b_i, y) := \log\left(1 + \exp\left(-b_i a_i^T y\right)\right)$. The LL variable $y$ is the model's parameter, while the UL variable $x$ refers to the regularization hyperparameter.

The performance of different methods on the unseen dataset $\mathcal{D}_{test}$ is shown in Figure 2, where the results over 10 runs are plotted for each method. Here AMIGO refers to the algorithm AMIGO in Arbel & Mairal (2022) that employs gradient descent to solve the auxiliary quadratic problem. The

results show that qNBO (BFGS) reaches its lowest loss faster than the other methods. Notably, the performance of qNBO (SR1) on the 20News dataset was omitted due to its ineffectiveness in this experiment, which resulted in oscillations. This issue arises from the numerical instability of SR1 when solving general functions without a correction strategy (Ye et al., 2023).

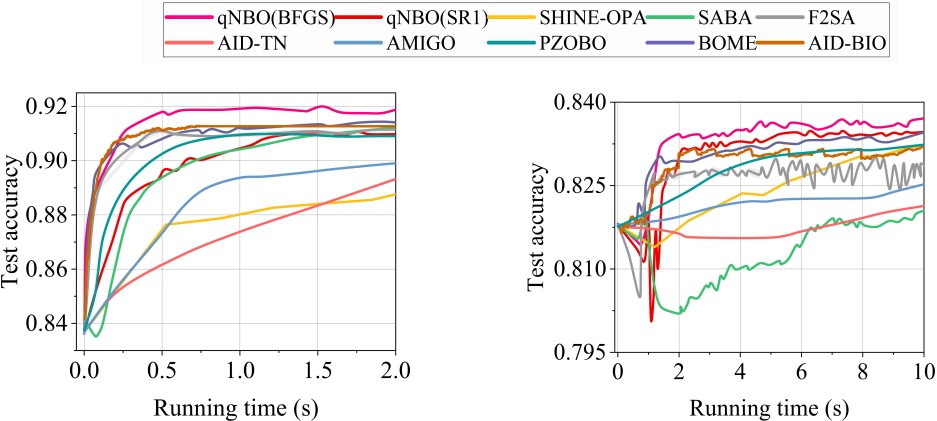

Figure 3: Data hyper-cleaning results on two datasets. (Left: **MNIST**; Right: **FashionMNIST**).

### 4.3 DATA HYPER-CLEANING

This subsection focuses on data hyper-cleaning for the MNIST (Deng, 2012) and FashionMNIST (Xiao et al., 2017) to enhance model accuracy, using a noisy training set $\mathcal{D}_{\text{train}} := \{a_i, b_i\}_{i=1}^m$ and a clean validation set $\mathcal{D}_{\text{val}}$. The objective is to adjust the training data weights to improve performance on $\mathcal{D}_{\text{val}}$. This task can be formalized as the bilevel problem:

$$\min_x \ \ell^{\text{val}}(y^*(x)) \ \text{s.t.} \ y^*(x) = \arg\min_y \{\ell^{\text{train}}(x, y) + c\|y\|^2\}, \tag{10}$$

where $\ell^{\text{val}}$ is the validation loss on $\mathcal{D}_{\text{val}}$ and $\ell^{\text{train}} = \sum_{i=1}^m \sigma(x_i)\ell(a_i, b_i, y)$ is a weighted training loss with $\sigma(x) = \text{Clip}(x, [0, 1])$ and $x \in \mathbb{R}^m$. In the experiment, both $\ell(a_i, b_i, y)$ and $\ell^{\text{val}}$ are the cross entropy loss, with $c = 0.001$.

Table 1: Comparison of results for hyper-cleaning. We compare the time and F1 score of various algorithms in achieving specific test accuracies (91.50% for MNIST and 83.00% for FashionMNIST). Bold font indicates the **fastest time** to reach the target accuracy. If an algorithm fails to reach the required test accuracy, the time is recorded up to the highest accuracy it achieves.

| | MNIST | | | FashionMNIST | | |
|---|---|---|---|---|---|---|
| Method | Time (s) | Acc. (%) | F1 score | Time (s) | Acc. (%) | F1 score |
| qNBO (BFGS) | **0.42** | 91.54 | 95.34 | **0.83** | 83.04 | 93.56 |
| qNBO (SR1) | 3.68 | 91.51 | 94.59 | 1.53 | 83.02 | 94.09 |
| BOME | 6.31 | 91.50 | 94.94 | 3.59 | 83.00 | 93.48 |
| SABA | 3.35 | 91.44 | 94.79 | 44.29 | 82.79 | 88.81 |
| F$^2$SA | 8.06 | 91.46 | 93.31 | 8.72 | 82.98 | 86.55 |
| SHINE-OPA | 20.25 | 91.51 | 95.44 | 9.96 | 83.07 | 93.87 |
| PZOBO | 1.05 | 91.46 | 95.46 | 2.96 | 83.05 | 93.77 |

As shown in Figure 3, qNBO (BFGS) significantly outperforms other methods, achieving lower test loss and higher test accuracy more quickly. All results are averaged over 10 random trials. Table 1 illustrates that while qNBO, BOME, and SHINE-OPA are able to achieve the required accuracy, qNBO (BFGS) does so in the shortest time. Notably, both F$^2$SA and SABA fail to reach the target accuracy on either dataset. For example, on the MNIST dataset, qNBO (BFGS) requires less than one-tenth of the time taken by BOME, the second-fastest method. The exclusion of BSG1's performance from this experiment is due to its ineffectiveness in addressing these data hyper-cleaning problems.

## 4.4 META-LEARNING

In this subsection, we consider the few-shot meta-learning, which can be described as:

$$\min_x \ \frac{1}{m}\sum_{i=1}^m \mathcal{L}_{\mathcal{D}_i}(x, y_i^*(x)) \ \text{ s.t. } y^*(x) = \arg\min_y \frac{1}{m}\sum_{i=1}^m \mathcal{L}_{\mathcal{S}_i}(x, y_i).$$

Due to the superior performance demonstrated by PZOBO compared to the baseline methods (MAML (Finn et al., 2017) and ANIL (Raghu et al., 2020)), the comparison of qNBO (BFGS) is exclusively conducted against PZOBO, excluding the aforementioned baseline methods.

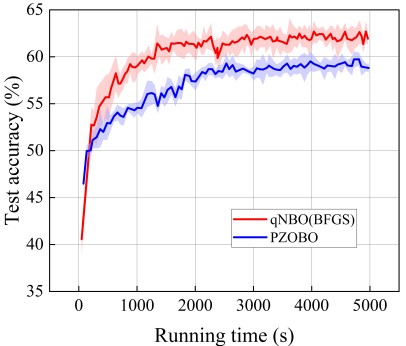 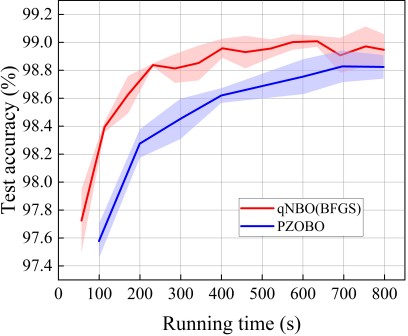

Figure 4: 5-way 5-shot experiments on two datasets. (Left: **miniImageNet**; Right: **Omniglot**.)

As shown in Figure 4 and Table 2, qNBO (BFGS) achieves superior accuracy compared to PZOBO on the miniImageNet (Vinyals et al., 2016) and Omniglot (Lake et al., 2015) datasets. Results are averaged over 5 runs, with all algorithms starting from the same initial point with a test accuracy of 20%. For clarity, the graphs begin at the second data point, omitting the initial one. It is reported that qNBO (BFGS) reaches peak test accuracies on the Omniglot dataset within 800 seconds, after which performance declines, likely due to overfitting or other factors. Notably, on the miniImageNet dataset, qNBO (BFGS) attains a test accuracy exceeding 60%, while PZOBO fails to achieve comparable results. Furthermore, the results in Table 2 show that qNBO (BFGS) reaches higher test accuracy in significantly less time compared to PZOBO as the number of ways increases. We do not plot the curves for other methods like SABA, SHINE-OPA, and BOME, as they are difficult to converge under various hyperparameter configurations using their source codes.

Table 2: Few-shot meta-learning on the Omniglot dataset: highest test accuracy and time required by each algorithm.

| 5-shot | PZOBO | | qNBO (BFGS) | |
|---|---|---|---|---|
| | Acc. (%) | Time (s) | Acc. (%) | Time (s) |
| 5-way | **99.31** | 11124 | 99.14 | **772** |
| 20-way | 97.94 | 17869 | **99.14** | **1648** |
| 30-way | 97.15 | 21043 | **99.05** | **2978** |

## 5 CONCLUSION

This paper introduces qNBO, a flexible algorithmic framework for improving hypergradient approximation. It leverages quasi-Newton techniques to accelerate the solution of the lower-level problem and efficiently approximates the inverse Hessian-vector product in hypergradient computation. Notably, qNBO includes a subroutine using quasi-Newton recursion schemes specifically tailored for the direction $\nabla_y F(x, y)$ to avoid incorrect inversion. Furthermore, in addition to the prominent BFGS and SR1 methods, qNBO can integrate other quasi-Newton methods such as limited-memory BFGS (L-BFGS) and single-loop implementations. Extensive numerical results verify the efficiency of the proposed algorithms.

## ACKNOWLEDGMENTS

Authors listed in alphabetical order. This work is supported by National Key R & D Program of China (2023YFA1011400), National Natural Science Foundation of China (12222106, 12271097, 12326605, 62331014, 12371305), Guangdong Basic and Applied Basic Research Foundation (No. 2022B1515020082), the Key Program of National Science Foundation of Fujian Province of China (Grant No. 2023J02007), the Central Guidance on Local Science and Technology Development Fund of Fujian Province (Grant No. 2023L3003), and the Fujian Alliance of Mathematics (Grant No. 2023SXLMMS01). We sincerely thank the anonymous reviewers for their insightful comments and suggestions on this work.

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

## A  OUTLINE OF APPENDIX

The appendix is organized as follows:

- Appendix B presents details of recursive algorithms for computing the inverse quasi-Newton matrix-vector product.
- Appendix C provides details of the numerical experiments from Section 4.
- Appendix D includes the proofs of the theorems from Section 3.2.
  - Appendix D.1 reviews some useful results of the BFGS method;
  - Appendix D.2 provides the proof sketch of Theorem 3.7;
  - Appendix D.3 presents the proof of Theorem 3.4;
  - Appendix D.4 presents the proof of Theorem 3.7.
- Appendix E contains the theoretical discussion and complexity analysis of the proposed algorithms.

## B  RECURSIVE PROCEDURE TO COMPUTE THE INVERSE HESSIAN APPROXIMATION-VECTOR PRODUCT

Due to the low-rank structure of the updates in (3) and (4), for any vector $d$, $r = H_{t+1}d$ can be efficiently computed using the recursive methods detailed in Algorithm 4 for the BFGS update (Nocedal, 1980), and Algorithm 5 for the SR1 update (Erway & Marcia, 2017), respectively. Note that Algorithms 4 and 5 involve only the computation of first-order information, provided $H_0$ is a scalar multiple of the identity matrix. Consequently, by avoiding the storage and computation of the full Hessian, computational costs can be significantly reduced.

---

**Algorithm 4** $\mathcal{C}_b(d, H_0, \{s_i, g_i\}_{i=0}^{t-1})$: Two-loop recursion for computing $r = H_t d$ when $H_t$ is the inverse of the BFGS matrix.

---

1: $q = d$;
2: **for** $i = t-1, t-2, \ldots, 0$
     $\alpha_i = (s_i^T q)/(g_i^T s_i)$;
     $q = q - \alpha_i g_i$;
   **end for**
3: $r = H_0 q$;
4: **for** $i = 0, \ldots, t-1$
     $\beta = (g_i^T r)/(g_i^T s_i)$;
     $r = r + (\alpha_i - \beta)s_i$;
   **end for**
**Return** $r = H_t d$.

---

---

**Algorithm 5** $\mathcal{C}_s(d, H_0, \{s_i, g_i\}_{i=0}^{t-1})$: Computing $r = H_t d$ when $H_t$ is the inverse of an SR1 matrix.

---

1: **for** $i = 0, \ldots, t-1$
     $p_i = s_i - H_0 g_i$;
2: **for** $j = 0, \ldots, i-1$
     $p_i = p_i - ((p_j^T g_i)/(p_j^T g_j))p_j$;
   **end for**
3: **end for**
**Return** $r = H_0 d + \sum_{i=0}^{t-1}((p_i^T d)/(p_i^T g_i))p_i$.

---

## C  DETAILS ON EXPERIMENTS

In this section, we additionally compare more algorithms, such as AMIGO (Arbel & Mairal (2022) with CG method), AID-BIO (Ji et al. (2021) with CG method) and AID-TN (Ji et al. (2021) with

Truncated Neumann method). At the same time, we add meta-learning experiments comparing with the PZOBO algorithm. All experiments are conducted on a server equipped with two NVIDIA A40 GPUs, an Intel(R) Xeon(R) Gold 6326 CPU, and 256 GB of RAM.

## C.1 DETAILS OF THE TOY EXAMPLE

In this experiment, the initial point for all algorithms is $(x_0, y_0) = (2\mathbf{e}, 2\mathbf{e})$ where $\mathbf{e}$ denotes the vector of all ones.

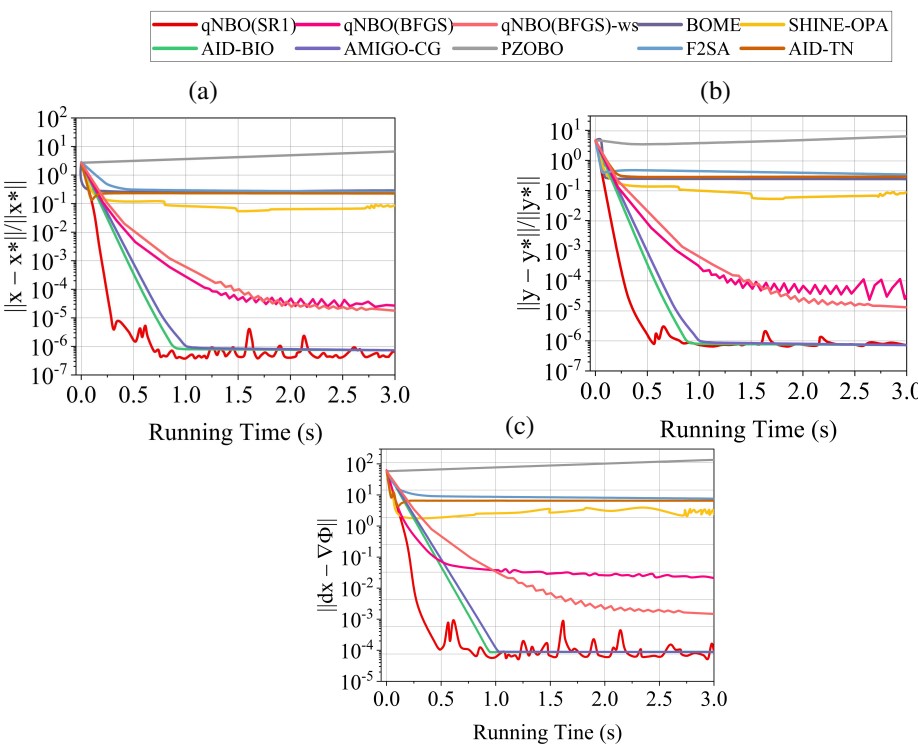

Figure 5: Numerical results on toy example: Comparison of qNBO (SR1), qNBO (BFGS) and qNBO (BFGS)-ws with other bilevel optimization methods in a toy experiment. Here, "ws" indicates that a warm-start strategy is applied for $u_k$, with $Q_k = \min(k + 1, 60)$.

**BOME:** The maximum number of outer iterations is $K = 5000$, the number of inner iterations is $T = 100$, the inner step size is $\alpha = 0.1$, the outer step size is $\xi = 0.1$, and

$$\lambda_k = \max \left\{ \frac{0.0001 \hat{q}(x_k, y_k) - \langle \nabla F(x_k, y_k), \nabla \hat{q}(x_k, y_k) \rangle}{\|\nabla \hat{q}(x_k, y_k)\|^2}, 0 \right\}.$$

**F²SA:** The maximum number of outer iterations is $K = 5000$, the number of inner iterations is $T = 10$, the inner step sizes are $\gamma_k = \alpha_k = 0.1$, the initial multiplier is $\lambda_0 = 0.1$, the multiplier increment is $\delta_k = 0.001$, and the step size ratio is $\xi = 1$.

**SHINE-OPA:** The maximum number of outer iterations is $K = 5000$, the maximum number of inner iterations is $T = 100$, the inner stopping criterion is $\|\nabla_y f(x_k, y_{k+1})\| \leq \frac{1}{k+1}$, the inner step size is determined using strong Wolfe line search, the number of extra updates of upper-level information in the BFGS algorithm is 5 (i.e., every 5 steps of BFGS iteration includes an update with UL gradient $\nabla_y F$), the initial matrix is $H_0 = I$, and the outer step size is $\alpha_k = 0.1$.

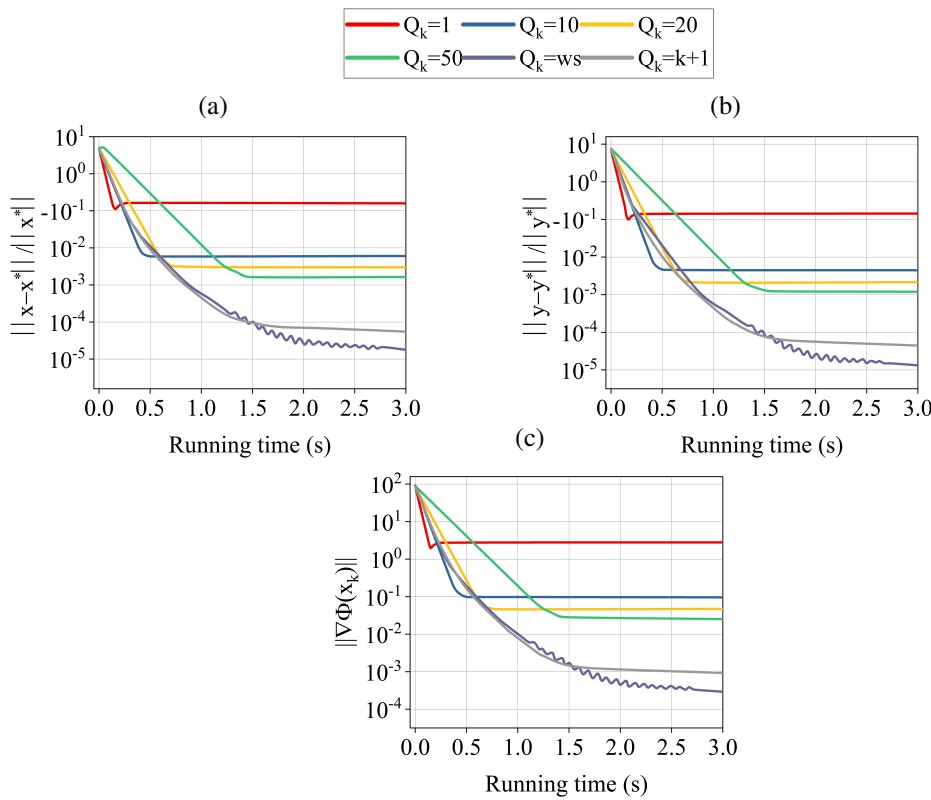

Figure 6: Testing results on the impact of the parameter $Q_k$ in qNBO (BFGS) in the toy example. (The qNBO (BFGS) parameters are the same as those in Figure 1, except for $Q_k$. Additionally, $Q_k = ws$ means that $Q_k = \min(k+1, 60)$, where the warm start strategy is used for $u_k$.)

**AID-TN:** The maximum number of outer iterations is $K = 5000$, the number of inner iterations is $T = 1$, the TN iteration $P = 1$, the inner step size is $\beta = 0.01$ and the outer step size is $\alpha = 0.2$.

**AID-BIO/AMIGO-CG:** The maximum number of outer iterations is $K = 5000$, the number of inner iterations is $T = 1$, the CG iteration $P = 1$, the inner step size is $\beta = 0.01$ and the outer step size is $\alpha = 0.2$.

**PZOBO:** The maximum number of outer iterations is $K = 5000$, the number of inner iterations is $Q = N = 10$, the parameter $\mu = 100$, the inner step size is $\alpha = 0.01$ and the outer step size is $\beta = 0.01$.

**qNBO (SR1):** The maximum number of outer iterations is $K = 5000$, the number of inner iterations is $T = 6$, the warm up iterations $P = 9$, the number of iterations is $Q_k = 25$, the inner step sizes are $\beta = 0.1, \gamma = 1$, the initial matrix is $H_0 = I$, and the outer step size is $\alpha = 0.4$.

**qNBO (BFGS):** The maximum number of outer iterations is $K = 5000$, the number of inner iterations is $T = 15$, the warm up iterations $P = 1$, the number of iterations $Q_k = k + 1$, the inner step sizes are $\beta = 0.1, \gamma = 1$, the initial matrix is $H_0 = I$, and the outer step size is $\alpha = 0.1$.

## C.2 FURTHER SPECIFICATIONS FOR LOGISTIC REGRESSION

### C.2.1 IMPLEMENTATIONS AND HYPERPARAMETER SETTINGS

This section introduces the specific parameters of different algorithms used in the logistic regression experiment, formulated as a bilevel problem:

$$\min_{x \in \mathbb{R}} \sum_{i=1}^{n'} \ell(a_i', b_i', y^*(x)) \ \text{s.t.} \ y^*(x) = \arg\min_{y \in \mathbb{R}^n} \sum_{i=1}^{n} \ell(a_i, b_i, y) + \frac{\exp(x)}{2} \|y\|^2, \qquad (11)$$

where $(a_i, b_i) \in \mathcal{D}_{train}$ and $(a_i', b_i') \in \mathcal{D}_{val}$ are the training data and validation data respectively, and $\ell(a_i, b_i, y) := \log\big(1 + \exp\big(-b_i a_i^T y\big)\big)$. The LL variable $y$ is the model's parameter, while the UL variable $x$ refers to the regularization hyperparameter.

In this task, we consider two dataset, 20news and Real-sim. The 20news dataset (Lang, 1995) comprises a total of 18,846 samples with 130,107 features. It is divided into three subsets: the training set $\mathcal{D}_{train}$ has 16961 samples, the validation set $\mathcal{D}_{val}$ has 943 samples, and the test set $\mathcal{D}_{test}$ has 942 samples. Similarly, the Real-sim dataset (Chang & Lin) contains 72,309 samples, each with 20,958 features. This dataset is also split into three parts: the training set $\mathcal{D}_{train}$ has 65078 samples, the validation set $\mathcal{D}_{val}$ has 3616 samples, and the test set $\mathcal{D}_{test}$ has 3615 samples. For all algorithms, we set $x_0$ to 0 and $y_0$ to a random value. Unless otherwise stated, the batch size of algorithm is assumed to be 200. In following, AMIGO refers to the algorithm AMIGO (Arbel & Mairal (2022)) that employs stochastic gradient descent to solve the auxiliary quadratic problem.

**BOME:** The maximum number of outer iterations is $K = 200$, the number of inner iterations is $T = 10$, the inner step size is $\alpha = 0.01$, the outer step sizes are $\xi_x = 0.1$ and $\xi_y = 0.01$. The update rule for $\lambda_k$ is:

$$\lambda_k = \max\left\{ \frac{0.5\hat{q}(x_k, y_k) - \langle \nabla F(x_k, y_k), \nabla \hat{q}(x_k, y_k) \rangle}{\|\nabla \hat{q}(x_k, y_k)\|^2}, 0 \right\}.$$

**F$^2$SA:** The maximum number of outer iterations is $K = 2000$, the number of inner iterations is $T = 10$, starting point $z_0 = y_0$, inner step sizes $\gamma_k = \alpha_k = 0.1$, the initial multiplier $\lambda_0 = 0.1$, multiplier increment $\delta_k = 0.0001$, step size ratio $\xi = 1$, and the batch size is 1000.

**SABA:** The maximum number of outer iterations is $K = 20000$, the initial point $v_0 = \mathbf{0}$, step sizes $\alpha_k = 0.125$ and $\beta_k = \beta_k^v = 0.125$, and the batch size is 32.

**BSG1:** The maximum number of outer iterations is $K = 300$, the number of inner iterations is $T = 10$, inner step size $\beta_k = 0.01$, outer step size $\alpha_k = 0.01$.

**SHINE-OPA:** The maximum number of outer iterations is $K = 30$, maximum number of inner iterations is $T = 1000$. The initial matrix $H_0 = I$, for more details see SHINE code.[†]

**qNBO (BFGS):** The maximum number of outer iterations is $K = 50$, the number of inner iterations is $T = 9$, warm-up iteration steps $P = 1$, iteration steps $Q_k = 1$, inner step sizes $\beta = 0.0001/(j + 1), \gamma = 0.1$, outer step size selection strategy is the same as the SHINE-OPA algorithm. When the dataset is the 20news dataset, the initial matrix $H_0 = 0.1I$; when the dataset is the Real-sim dataset, $H_0 = 0.01I$.

**qNBO (SR1):** The maximum number of outer iterations is $K = 100$, the number of inner iterations is $T = 7$, warm-up iteration steps $P = 3$, iteration steps $Q_k = 3$, inner step sizes $\beta = 0.0001/(j + 1), \gamma = 0.1$, initial matrix $H_0 = 0.1I$ and outer step size selection strategy is the same as the SHINE-OPA algorithm.

**AID-BIO/AMIGO-CG:** The maximum number of outer iterations is $K = 1000$, the number of inner iterations is $T = 1$, the CG iteration $P = 1$, inner step sizes $\beta = 0.01$ and outer step size $\alpha = 0.01$.

---

[†] https://github.com/zaccharieramzi/hoag/tree/shine

**AMIGO:** The maximum number of outer iterations is $K = 1000$, the batch size is 32, the number of inner iterations is $T = 1$, inner step sizes $\beta = 0.01$ and outer step size $\alpha = 0.1$.

**AID-TN:** The maximum number of outer iterations is $K = 1000$, the number of inner iterations is $T = 1$, the TN iteration $P = 1$, inner step sizes $\beta = 0.01$ and outer step size $\alpha = 0.1$.

**PZOBO:** The maximum number of outer iterations is $K = 5000$, the number of inner iterations is $Q = N = 10$, the parameter $\mu = 10$, the inner step size is $\alpha = 0.01$ and the outer step size is $\beta = 0.03$.

### C.3 FURTHER SPECIFICATIONS ON DATA HYPER-CLEANING EXPERIMENTS

This subsection focuses on data hyper-cleaning to enhance model accuracy, using a noisy training set $\mathcal{D}_{\text{train}} := \{a_i, b_i\}_{i=1}^m$ and a clean validation set $\mathcal{D}_{\text{val}}$. The goal is to adjust training data weights to enhance performance on $\mathcal{D}_{\text{val}}$. The task can be formalized as the bilevel problem:

$$\min_x \ \ell^{\text{val}}(y^*(x)) \ \text{ s.t. } y^*(x) = \arg\min_y \{\ell^{\text{train}}(x, y) + c\|y\|^2\}, \tag{12}$$

where $\ell^{\text{val}}$ is the validation loss on $\mathcal{D}_{\text{val}}$ and $\ell^{\text{train}} = \sum_{i=1}^m \sigma(x_i)\ell(a_i, b_i, y)$ is a weighted training loss with $\sigma(x) = \text{Clip}(x, [0, 1])$ and $x \in \mathbb{R}^m$. In the experiment, both $\ell(a_i, b_i, y)$ and $\ell^{\text{val}}$ are the cross entropy loss, with $c = 0.001$.

Experiments are conducted using MNIST (Deng, 2012) and FashionMNIST (Xiao et al., 2017) datasets, with 50% of the training data corrupted by randomly assigning them sampled labels. The data is divided into four parts: training set, validation sets 1 and 2, and the test set. The training set comprises 50000 samples, while the validation and test sets contain 5000 and 10000 samples, respectively. For each method, model training is conducted on the training set, with the tuning of hyperparameter $x$ using validation set 1. The LL variable $y = (W, b)$ denotes the parameters of the linear model with weight $W \in \mathbb{R}^{10 \times 784}$ and bias $b \in \mathbb{R}^{10}$.

#### C.3.1 IMPLEMENTATIONS AND HYPERPARAMETER SETTINGS

The initial point $y_0$ for all algorithms is obtained from a pretrained initialization model, and the initial weight vector $x_0 = 0.5\mathbf{e} \in \mathbb{R}^{50000}$.

**BOME:** The maximum number of outer iterations $K = 10000$, the number of inner iterations $T = 1$, the inner step size $\alpha = 0.01$, the outer step sizes $\xi_x = 100, \xi_y = 0.01$, and

$$\lambda_k = \max \left( \frac{0.1\hat{q}(x_k, y_k) - \langle \nabla F(x_k, y_k), \nabla \hat{q}(x_k, y_k) \rangle}{\|\nabla \hat{q}(x_k, y_k)\|^2}, 0 \right).$$

Details can be seen in the code.‡

**F²SA:** The maximum number of outer iterations $K = 7000$, the number of inner iterations $T = 1$, the initial point $z_0 = y_0$, the inner step size $\gamma_k = \alpha_k = 0.01$, the initial multiplier $\lambda_0 = 0.1$, the difference in multiplier $\delta_k = 0.001$, the step size ratio $\xi = 10000$, and the batch size is 2000.

**SABA:** The maximum number of outer iterations $K = 10000$, the initial point $v_0 = \mathbf{0}$, the batch size is 2000. For the MNIST dataset, the step sizes $\alpha_k = 10, \beta_k = 0.01, \beta_k^v = 0.1$; otherwise, the step sizes $\alpha_k = 100, \beta_k = \beta_k^v = 0.001$.

**SHINE-OPA:** The maximum number of outer iterations $K = 50$, the maximum number of inner iterations $T = 1000$, the inner stopping criterion $\|\nabla_y f(x_k, y_{k+1})\| \leq 1/(100k)$, the inner step size is determined using strong Wolfe line search, the number of extra updates in the BFGS algorithm is 5 (i.e., upper-level information is introduced in the BFGS iterations for every 5 steps), the initial matrix $H_0 = I$, and the outer step size is 100.

---

‡ https://github.com/Cranial-XIX/BOME

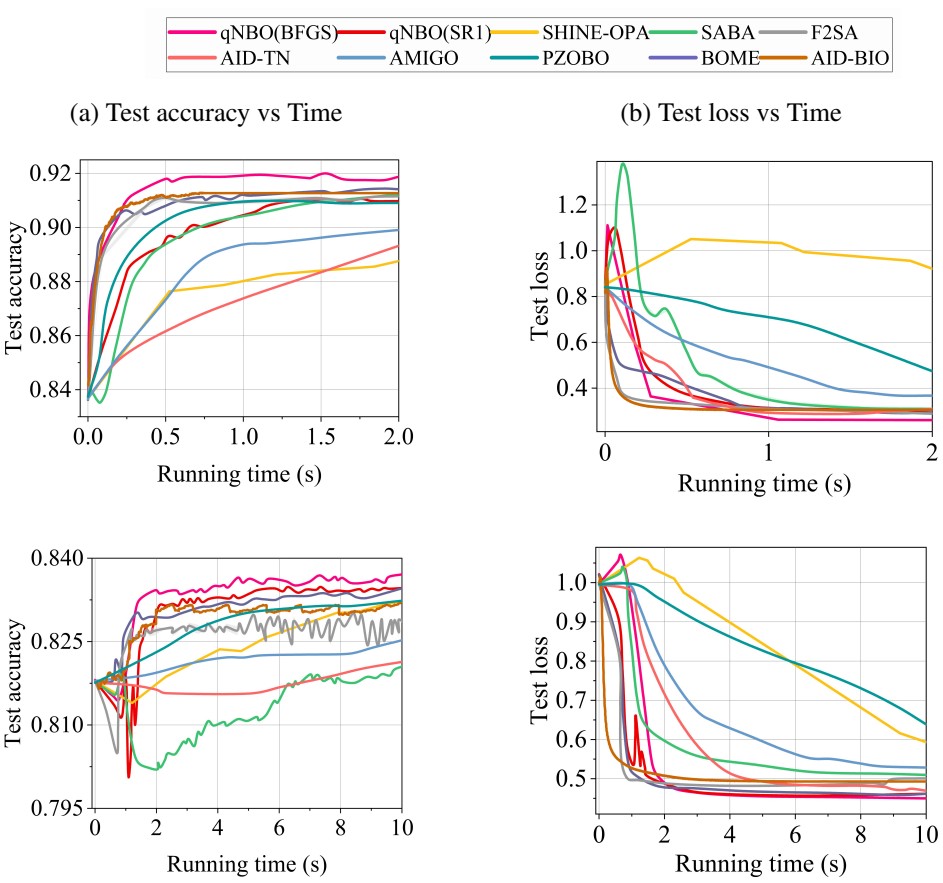

Figure 7: Data hyper-cleaning on two datasets. (First row: **MNIST**; Second row: **FashionMNIST**. All results are averaged over 10 random trials. The exclusion of BSG1's performance in this experiment is due to its ineffectiveness in addressing these data hyper-cleaning problems.)

**qNBO (BFGS):** The number of iterations $Q_k = 1$, the inner step sizes $\beta = 0.1, \gamma = 0.1$, the outer step size $\alpha = 100$, the initial matrix $H_0 = I$. For the MNIST dataset, the maximum number of outer iterations $K = 5000$, the number of inner iterations $T = 7$, the warm-up iteration count $P = 3$; otherwise, the maximum number of outer iterations $K = 600$, the number of inner iterations $T = 47$, and the warm-up iteration count $P = 3$.

**qNBO (SR1):** The number of inner iterations $T = 17$, the warm-up iteration count $P = 3$, the number of iterations $Q_k = 3$, the inner iteration step sizes $\beta = 0.1, \gamma = 0.1$, the outer iteration step size $\alpha = 100$, the initial matrix $H_0 = 0.01I$. In addition, the inner iteration will terminate early if $\|\nabla_y f(x_k, y_{k+1})\| \leq 0.1$. For the MNIST dataset, the maximum number of outer iterations $K = 5000$; otherwise, the maximum number of outer iterations $K = 2000$.

**AID-BIO/AMIGO-CG:** The maximum number of outer iterations is $K = 1000$, the number of inner iterations is $T = 1$, the CG iteration steps $P = 1$, inner step sizes $\beta = 0.01$ and outer step size $\alpha = 10$.

**AMIGO:** The maximum number of outer iterations is $K = 1000$, the batch size is 200, the number of inner iterations is $T = 1$, inner step sizes $\beta = 0.01$ and outer step size $\alpha = 100$.

**AID-TN:** The maximum number of outer iterations is $K = 1000$, the number of inner iterations is $T = 1$, the TN iteration steps $P = 1$, inner step sizes $\beta = 0.01$ and outer step size $\alpha = 30$.

**PZOBO:** The maximum number of outer iterations is $K = 5000$, the number of inner iterations is $Q = N = 10$, the parameter $\mu = 0.01$, the inner step size is $\alpha = 0.01$ and the outer step size is $\beta = 10$.

## C.4 META-LEARNING

In this subsection, we consider the few-shot meta-learning, which can be described as:

$$\min_x \; \frac{1}{m} \sum_{i=1}^m \mathcal{L}_{\mathcal{D}_i}(x, y_i^*(x)) \; \text{ s.t. } y^*(x) = \arg\min_y \frac{1}{m} \sum_{i=1}^m \mathcal{L}_{\mathcal{S}_i}(x, y_i),$$

where $\mathcal{L}_{\mathcal{D}_i}(x, y_i^*) = \frac{1}{|\mathcal{D}_i|} \sum_{\xi \in \mathcal{D}_i} \mathcal{L}(x, y_i^*; \xi)$ is the validation loss function and $\mathcal{L}_{\mathcal{S}_i}(x, y_i) = \frac{1}{|\mathcal{S}_i|} \sum_{\xi \in \mathcal{S}_i} (\mathcal{L}(x, y_i; \xi) + \mathcal{R}(y_i))$ is the training loss with the classification loss $\mathcal{L}$ and the strongly-convex regularizer $\mathcal{R}(y_i)$. In the experiment, $\mathcal{L}$ is the cross-entropy function and $\mathcal{R}$ is the $\ell_2$ norm. In our experimental setting, the task-specific parameters $y$ denote the weights of the last linear layer of a neural work and $x$ are the parameters of a 4-layer convolutional neural networks (CNN4).

The few-shot meta-learning has $m$ tasks $\{\mathcal{T}_i, i = 1, \cdots, m\}$ sampled over a distribution $\mathcal{P}_{\mathcal{T}}$. Each task $\mathcal{T}_i$ has a loss function $\mathcal{L}(x, y_i; \xi)$ with data sample $\xi$, the task-specific parameters $y_i$ and the parameters $x$ of an embedding model shared by all tasks.

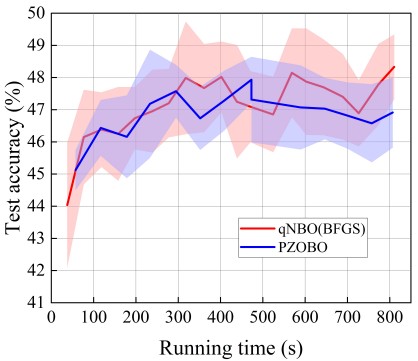

Figure 8: 5way-5shot on FC100 datasets. Results are averaged over 5 runs, with all algorithms starting from the same initial point with a test accuracy of 20%. For clarity, graphs begin at the second data point, omitting the initial one. We report that qNBO (BFGS) reaches peak test accuracies within 800 seconds, after which performance declines, likely due to overfitting or other factors.

### C.4.1 DATASETS

**miniImageNet:** The miniImageNet dataset (Vinyals et al., 2016), derived from ImageNet (Russakovsky et al., 2015), is a large-scale benchmark for few-shot learning. The dataset comprises 100 classes, each encompassing 600 images of size 84 × 84. Following Arnold et al. (2020), we partition the classes into 64 classes for meta-training, 16 classes for meta-validation, and 20 classes for meta-testing. In the experiment, CNN4 has four convolutional blocks, in which each convolutional block contains a 3×3 convolution (padding=1), ReLU activation, 2×2 max pooling and batch normalization. Each convolutional layer has 32 filters.

**FC100:** The FC100 dataset (Oreshkin et al., 2018), generated from Krizhevsky & Hinton (2009), consists of 100 classes with each class containing 600 images of size 32. Following Oreshkin et al. (2018), the classes are split into 60 classes for meta-training, 20 classes for meta-validation, and 20 classes for meta-testing. Each convolutional block of CNN4 comprises a 3×3 convolutional layer (with padding set to 1 and a stride of 2), subsequent batch normalization, ReLU activation, and 2×2 max pooling. Each convolutional layer has 64 filters.

**Omniglot:** Comprising 1623 character classes derived from 50 diverse alphabets, the Omniglot dataset (Lake et al., 2015) contains 20 samples within each class. The classes are divided into three parts: 1100 classes for meta-training, 100 classes for meta-validation, and 423 classes for meta-testing. The CNN4 network is identical to that used on the miniImageNet dataset but has 64 filters per layer.

### C.4.2 EXPERIMENTAL SETUP

The meta learning experiment is carried out using the code available at the website.[§] At each meta-iteration, a batch of 16 training tasks is sampled and the parameters are updated based on these tasks. The max outer steps $K$ is set to 6000 for both algorithms. The initial parameter $y_0$ is selected as 0.

**PZOBO**: The parameters involved in the algorithm are the same as those in Sow et al. (2022b).

**qNBO (BFGS)**: For all three datasets, the inner steps T is set to 20, the hypergradient updates $Q_k$ is 3, the step sizes $\beta$ and $\gamma$ are both specified as 0.1, and the Hessian matrix is initialized as $H_0 = 0.01I$. In addition, the inner iteration will terminate early if $\|A\| + \|b\| \leq \text{tol}$, where $A$ and $b$ are the components of the parameter $y$ denoting the weights of the final linear layer in a neural work. The specific values of tol and other parameters can be found in the code provided in the supplementary materials.

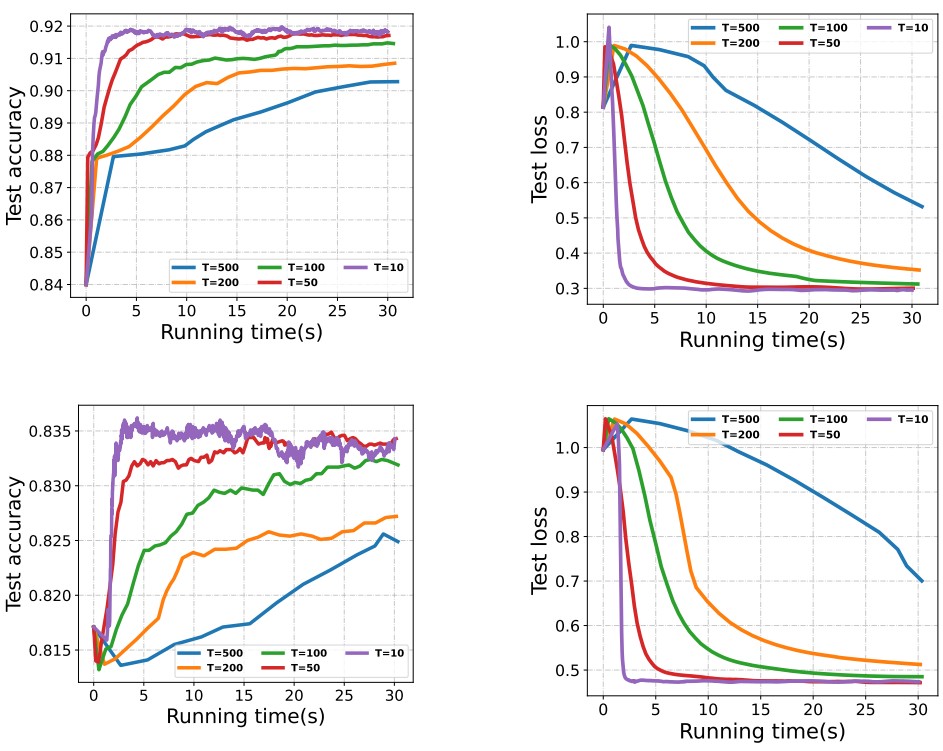

Figure 9: Ablation study on the iteration number $T$ of qNBO (BFGS). (The first row illustrates the test accuracy and test loss for MNIST, and the second row shows these values for FashionMNIST.)

### C.5 ABLATION STUDY

### C.5.1 TOY EXAMPLE

In this subsection, we conduct an ablation study to assess the impact of the parameters $Q_k$ on the performance of the qNBO (BFGS) algorithm in the toy experiment. As illustrated in Figure 6, the setting $Q_k = k + 1$ outperforms the others. Furthermore, employing the warm start strategy for $u_k$ (denoted as $Q_k = \text{ws}$) enhances the performance of qNBO (BFGS), suggesting the potential benefits of this strategy.

---

[§]https://github.com/sowmaster/esjacobians

### C.5.2 DATA HYPER-CLEANING

In this subsection, we conduct an ablation study to assess the impact of the parameters $T$ and $Q_k$ within the qNBO (BFGS) algorithm on its performance in the data hyper-cleaning experiment. As illustrated in Figures 9 and 10, smaller values of $T$ and $Q_k$ lead to improved performance in terms of both accuracy and loss across the two datasets, thereby indicating the efficiency of qNBO (BFGS).

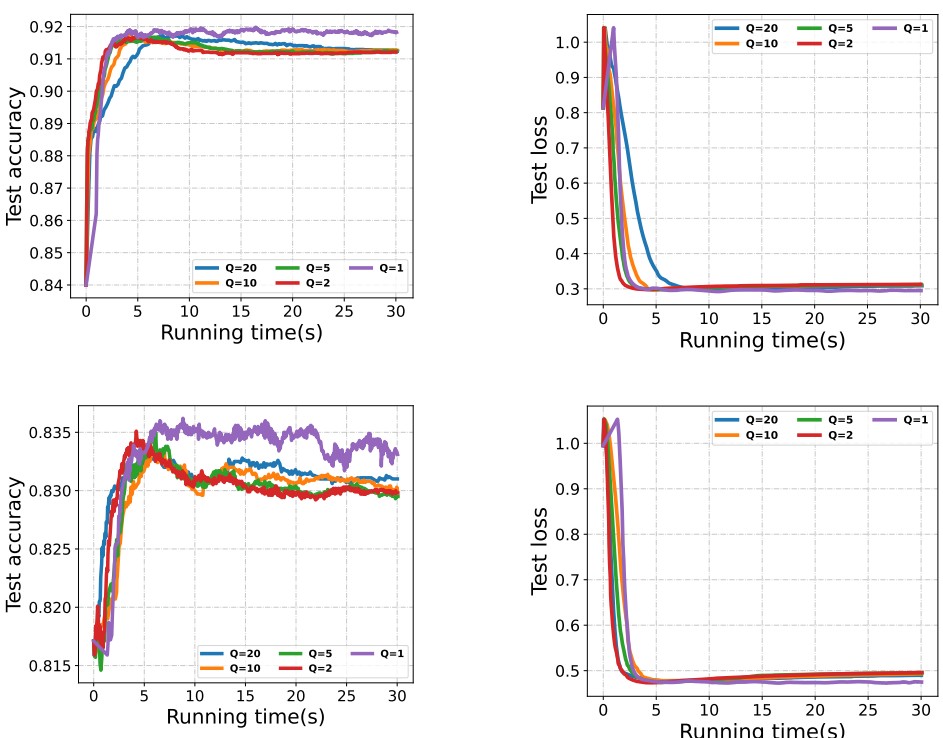

Figure 10: Ablation study on the iteration number $Q$ of qNBO (BFGS). (The first row shows the test accuracy and test loss for the MNIST dataset, and the second row shows the same metrics for the FashionMNIST dataset.)

## D PROOF OF THE RESULTS IN SECTION 3.2

### D.1 RESULTS OF QUASI-NEWTON METHOD

In this subsection, we first summarize the convergence properties of BFGS method for solving the problem:

$$\min_{y \in \mathbb{R}^n} g(y). \tag{13}$$

Besides, to derive the upper bound of the hypergradient estimation error, we review some conclusions of BFGS update presented in Jin & Mokhtari (2023); Rodomanov & Nesterov (2022; 2021b).

### D.1.1 CONVERGENCE RESULTS OF BFGS METHOD

**Assumption D.1.** Assume that $g$ has the following properties:

(i) $g(y)$ is strongly convex *w.r.t.* $y$ with parameter $\mu > 0$, i.e., $\mu I \preceq \nabla^2 g(y)$. Moreover, $\nabla g(y)$ is Lipschitz continuous *w.r.t.* y with parameter $L > 0$ (i.e., $\nabla^2 g(y) \preceq LI$ ).

(ii) The Hessian $\nabla^2 g(y)$ satisfies:

$$\nabla^2 g(y_1) - \nabla^2 g(y_2) \preceq M \|y_1 - y_2\|_z \nabla^2 g(w), \forall y_1, y_2, z, w \in \mathbb{R}^n,$$

where $\|y\|_z := \langle \nabla^2 g(z)y, y \rangle^{1/2}$ and $M > 0$.

**Lemma D.2.** *(Rodomanov & Nesterov (2021b), Global convergence) If the function $g$ has the following quadratic form:*

$$g(y) = \frac{1}{2}y^T A y - y^T x, \tag{14}$$

*where $\mu I \preceq A \preceq LI$ such that Assumption D.1 holds. If the BFGS method is used to solve the problem (13), and if $H_0 = LI$ and the number of iterations $i \geq 4n \ln \frac{L}{\mu}$, then for all $i \geq 4n \ln \frac{L}{\mu}$, the following inequality holds:*

$$\|y_i - y^*\| \leq 2\kappa^{3/2}(\frac{t_b}{i})^{\frac{i}{2}}\|y_0 - y^*\|, \tag{15}$$

*with $\kappa = \frac{L}{\mu}$ and $t_b = 4n\ln\frac{L}{\mu}$.*

**Lemma D.3.** *(Rodomanov & Nesterov (2021b), local convergence) Suppose that Assumption D.1 on $g$ holds. If $H_0 = LI$ and the initial point $y_0$ satisfies:*

$$\|y_0 - y^*\| \leq K_1, K_1 = \frac{2\ln^{\frac{3}{2}}\sqrt{L}}{3^{\frac{3}{2}}M\mu}\max\{\frac{\mu}{2L}, \frac{1}{K_0 + 9}\}, \tag{16}$$

*where $K_0 = 8n\ln\frac{2L}{\mu}$ and $i \geq K_0$, then for all $i \geq K_0$, the iterate generated by the BFGS algorithm has the following superlinear convergence rate:,*

$$\|y_i - y^*\| \leq 2\kappa^{3/2}(\frac{t_c}{i})^{\frac{i}{2}}\|y_0 - y^*\|, \tag{17}$$

*with $t_c = \frac{9}{8}K_0$ and $\kappa = \frac{L}{\mu}$.*

**Lemma D.4.** *(Theorem 3 of Jin & Mokhtari (2023)) Suppose that Assumption D.1 on $g$ holds. If the initial point $y_0$ and the initial Hessian approximation matrix $B_0$ (with $H_0 = B_0^{-1}$) satisfy:*

$$\begin{aligned}
\|\nabla^2 g(y^*)^{1/2}(y_0 - y^*)\| &\leq \frac{\epsilon}{6}, \\
\|\nabla^2 g(y^*)^{-1/2}\big(B_0 - \nabla^2 g(y^*)\big)\nabla^2 g(y^*)^{-1/2}\|_F &\leq \delta,
\end{aligned} \tag{18}$$

*where $\epsilon, \delta \in (0, \frac{1}{2}), \rho \in (0, 1)$,*

$$\frac{(3 + \epsilon)\epsilon}{(1 - \epsilon)(1 - \rho)} \leq \delta, \text{and } \frac{\epsilon}{3} + 2\delta \leq (1 - 2\delta)\rho,$$

*then the iterate generated by the BFGS algorithm has the following superlinear convergence rate:*

$$\|y_i - y^*\| \leq \sqrt{\frac{L}{\mu}}\big(\frac{C_1 q\sqrt{i} + C_2}{i}\big)^i\|y_0 - y^*\|, \tag{19}$$

*where $C_1 = 2\sqrt{2}\delta(1 + \rho)(1 + \frac{\epsilon}{3})$, $C_2 = \frac{(1+\rho)(1+\frac{\epsilon}{3})\epsilon}{3(1-\rho)}$ and $q = \sqrt{\frac{1+2\delta}{1-2\delta}}$.*

To be specific, the superlinear convergence rate of BFGS algorithm can also be expressed in the following alternative form.

**Lemma D.5.** *(Corollary 4 of Jin & Mokhtari (2023)) Suppose that Assumption D.1 on $g$ holds. If the initial point $y_0$ and the initial BFGS matrix $B_0$ satisfy:*

$$\begin{aligned}
\|\nabla^2 g(y^*)^{1/2}(y_0 - y^*)\| &\leq \frac{1}{300}, \\
\|\nabla^2 g(y^*)^{-1/2}\big(B_0 - \nabla^2 g(y^*)\big)\nabla^2 g(y^*)^{-1/2}\|_F &\leq \frac{1}{7},
\end{aligned} \tag{20}$$

*then the iterate solved by the BFGS algorithm exhibits the following convergence rate:*

$$\|y_i - y^*\| \leq \sqrt{\frac{L}{\mu}}(\frac{1}{i})^{\frac{i}{2}}\|y_0 - y^*\|. \tag{21}$$

### D.1.2 PROPERTIES OF THE BFGS UPDATES

**Lemma D.6.** *If the function g in the problem (13) has the following quadratic form:*

$$g(y) = \frac{1}{2}y^T A y - y^T x, \tag{22}$$

*with $\mu I \preceq A \preceq LI$, then the BFGS matrix $B_i$ satisfies:*

$$\sum_{i=0}^{k-1} \frac{(B_i s_i - A s_i)^T A^{-1}(B_i s_i - A s_i)}{s_i^T B_i s_i} \leq \frac{nL}{\mu}. \tag{23}$$

*Proof.* Define $\sigma_i := \sigma(A, B_i) = \langle A^{-1}, B_i - A \rangle = \mathrm{Tr}(A^{-1}(B_i - A)) \geq 0$, then

$$\begin{aligned}
\sigma(A, B_i) - \sigma(A, B_{i+1}) &= \langle A^{-1}, B_i - B_{i+1} \rangle \\
&= \frac{\langle B_i A^{-1} B_i s_i, s_i \rangle}{\langle B_i s_i, s_i \rangle} - 1 \\
&= \frac{\langle B_i \left(A^{-1} - B_i^{-1}\right) B_i s_i, s_i \rangle}{\langle B_i s_i, s_i \rangle} \\
&\geq \frac{\langle (B_i - A) A^{-1}(B_i - A) s_i, s_i \rangle}{\langle B_i s_i, s_i \rangle},
\end{aligned} \tag{24}$$

where the last inequality follows from the fact that

$$\begin{aligned}
(B_i - A) A^{-1}(B_i - A) &= B_i A^{-1} B_i - 2 B_i + A \\
&\stackrel{A \preceq B_i}{\preceq} B_i A^{-1} B_i - B_i = B_i(A^{-1} - B_i^{-1}) B_i.
\end{aligned}$$

Thus, it is derived that

$$\sigma_i - \sigma_{i+1} \geq \frac{\langle (B_i - A) A^{-1}(B_i - A) s_i, s_i \rangle}{\langle B_i s_i, s_i \rangle}, \quad \forall 0 \leq i \leq k - 1.$$

Finally, summing the above inequality over $i$ yields:

$$\begin{aligned}
\sum_{i=0}^{k-1} \frac{\langle (B_i - A) A^{-1}(B_i - A) s_i, s_i \rangle}{\langle B_i s_i, s_i \rangle} &\leq \sigma_0 - \sigma_q \leq \sigma_0 = \sigma(A, LI) = \langle A^{-1}, LI - A \rangle \\
&\leq \left\langle A^{-1}, \frac{L}{\mu} A - A \right\rangle = n\left(\frac{L}{\mu} - 1\right) \leq \frac{nL}{\mu}.
\end{aligned}$$

$\square$

**Definition D.7.** Define

$$\psi(A, G) \triangleq \langle A^{-1}, G - A \rangle - \ln \mathrm{Det}(A^{-1}G), \tag{25}$$

where $\mathrm{Det}$ denotes the determinant of the matrix.

**Definition D.8.** Define

$$\theta(A, B, u) := \left[ \frac{\langle (B - A) A^{-1}(B - A) u, u \rangle}{\langle B A^{-1} B u, u \rangle} \right]^{1/2},$$

and let $\vartheta : (-1, +\infty) \to \mathbb{R}$ be the univarite function:

$$\vartheta(t) := t - \ln(1 + t) \geq 0.$$

*Remark* D.9. On the interval $[0, +\infty)$, $\vartheta(t)$ satisfies:

$$\frac{t^2}{2(1 + t)} \leq \vartheta(t) \leq \frac{t^2}{2 + t}. \tag{26}$$

**Lemma D.10.** (Rodomanov & Nesterov (2021b), Lemma 5.2 ) *Define* $J_i := \int_0^1 \nabla^2 g(y_i + ts_i)\, dt$ *and* $y_{i+1} = y_i + s_i$. *Then,* $J_i s_i = \nabla g(y_{i+1}) - \nabla g(y_i)$. *If* $B_0 = LI$, *then* $\forall i \geq 0$, *the* BFGS *matrix* $B_i$ *satisfies:*

$$\frac{1}{\xi_i} \nabla^2 g(y_i) \preceq B_i \preceq \xi_i \frac{L}{\mu} \nabla^2 g(y_i), \tag{27}$$

$$\frac{1}{\xi_{i+1}} J_i \preceq B_i \preceq \xi_{i+1} \frac{L}{\mu} J_i, \tag{28}$$

*where* $r_i := \|s_i\|_{y_i}$, $\quad \xi_i := e^{M \sum_{j=0}^{i-1} r_j}$ $\quad (\geq 1)$ *and the strongly self-concordant constant* $M$ *of* $g$.

**Lemma D.11.** *When* $B_0 = LI$, *the* BFGS *matrix* $B_i$ *satisfies (27) and (28). If* $\bar{\xi} = \max\limits_{i=0,\cdots,k-1} \xi_{i+1} \leq 2$ *in (28), then the following inequality holds:*

$$\tilde{\xi} \sum_{i=0}^{k-1} \theta_i^2 \leq n \left( \frac{L}{\mu} - 1 \right) + \sum_{i=0}^{k-1} \Delta_i, \tag{29}$$

*where* $\tilde{\xi} = \frac{1}{2(\bar{\xi}^2 + \bar{\xi})}$, $\theta_i := \theta(J_i, B_i, u_i)$, $\psi_i := \psi(J_i, B_i)$, $\tilde{\psi}_{i+1} := \psi(J_i, B_{i+1})$, *and* $\Delta_i := \psi_{i+1} - \tilde{\psi}_{i+1}$.

*Proof.* Note that

$$\frac{1}{\xi_{i+1}} J_i \preceq B_i \preceq \frac{\xi_{i+1} L}{\mu} J_i. \tag{30}$$

From Lemma 2.4 of Rodomanov & Nesterov (2022), it can be further deduced that:

$$\psi_i - \tilde{\psi}_{i+1} \geq \vartheta \left( \frac{1}{\xi_{i+1}} \theta_i \right). \tag{31}$$

Since $\bar{\xi} = \max\limits_{i=0,\cdots,k-1} \xi_{i+1} \leq 2$, it follows from the definition of $\theta_i$ that:

$$\theta_i^2 = \frac{\langle (B_i - J_i) J_i^{-1} (B_i - J_i) u_i, u_i \rangle}{\langle B_i J_i^{-1} B_i u_i, u_i \rangle} = 1 - \frac{\langle (2B_i - J_i) u_i, u_i \rangle}{\langle B_i J_i^{-1} B_i u_i, u_i \rangle} \overset{(30)}{\leq} 1. \tag{32}$$

Then, it is derived that:

$$\vartheta \left( \frac{1}{\xi_{i+1}} \theta_i \right) \overset{(26)}{\geq} \frac{\frac{1}{\xi_{i+1}^2} \theta_i^2}{2 \left( 1 + \frac{1}{\xi_{i+1}} \theta_i \right)} \geq \frac{\frac{1}{\xi_{i+1}^2}}{2 \left( 1 + \frac{1}{\xi_{i+1}} \right)} \theta_i^2 = \frac{1}{2 \left( \xi_{i+1}^2 + \xi_{i+1} \right)} \theta_i^2. \tag{33}$$

Thus, it holds that:

$$\tilde{\xi} \theta_i^2 \leq \psi_i - \tilde{\psi}_{i+1} = \psi_i - \psi_{i+1} + \Delta_i, \quad \forall i \in \{0, \ldots, k-1\}, \tag{34}$$

where $\tilde{\xi} = \frac{1}{2(\bar{\xi}^2 + \bar{\xi})}$ and

$$\Delta_i := \psi_{i+1} - \tilde{\psi}_{i+1} = \langle J_{i+1}^{-1} - J_i^{-1}, B_{i+1} \rangle + \ln \mathrm{Det}(J_i^{-1}, J_{i+1}). \tag{35}$$

By summing equation (34) over $i$ and given that $\psi_k \geq 0$, it follows that:

$$
\begin{aligned}
\tilde{\xi} \sum_{i=0}^{k-1} \theta_i^2 &\leq \psi_0 - \psi_k + \sum_{i=0}^{k-1} \Delta_i \leq \psi_0 + \sum_{i=0}^{k-1} \Delta_i \\
&= \psi(J_0, LI) + \sum_{i=0}^{k-1} \Delta_i \\
&= \langle J_0^{-1}, LI - J_0 \rangle - \ln \mathrm{Det}(J_0^{-1}, LI) + \sum_{i=0}^{k-1} \Delta_i \\
&\leq \langle J_0^{-1}, LI - J_0 \rangle + \sum_{i=0}^{k-1} \Delta_i \\
&\leq \langle J_0^{-1}, \frac{L}{\mu} J_0 - J_0 \rangle + \sum_{i=0}^{k-1} \Delta_i \\
&= n\left(\frac{L}{\mu} - 1\right) + \sum_{i=0}^{k-1} \Delta_i.
\end{aligned}
\tag{36}
$$

$\square$

**Notations:** In step 1 of Algorithm 1, $y_k^0 = y_k$ establishes the initial value of $y$ at the start of the $k$-th iteration. After $P$ warm-up iterations, denoted as $y_{k,0} = y_k^P$, the term $y_{k,T}$ refers to the state of $y$ following $T$ further iterations with $x_k$ fixed. In the second step of Algorithm 1, $u_{k,Q_k}$ represents the state of $u$ after $Q_k$ iterations, with both $x_k$ and $y_{k+1}$ fixed.

### D.2 PROOF SKETCH OF THEOREM 3.7

The proofs of Theorem 3.4 and 3.7 encompasses three critical steps: first, it splits the hypergradient approximation error into the T-step error of estimating the lower level solution $\|y_{k,T} - y_k^*\|^2$ and the $Q_k$-step error $\|u_{k,Q_k} - u_k^*\|^2$; second, it establishes upper bounds for these errors based on previous iteration errors; finally, it combines the above results to substantiate the theorem's convergence. Since the proofs of Theorem 3.4 and 3.7 are similar, we only elaborate on the proof sketch of Theorem 3.7 below.

**Step 1:** Decomposing the hypergradient estimation error.

First, the hypergradient estimation error at the $k$th iteration can be bounded by:

$$
\|\tilde{\nabla}\Phi(x_k) - \nabla\Phi(x_k)\|^2 \leq 3\left(L_{F_x}^2 + \frac{L_{f_{xy}}^2 C_{F_y}^2}{\mu^2}\right)\|y_{k,T} - y_k^*\|^2 + 3M_{f_{xy}}^2\|u_{k,Q_k} - u_k^*\|^2.
\tag{37}
$$

**Step 2:** Upper-bounding the error.

The T-step error of estimating the lower level solution $\|y_{k,T} - y_k^*\|^2$ is bounded by:

$$
\begin{aligned}
\|y_k^* - y_{k,T}\|^2 &\leq \tau\|y_{k-1}^* - y_{k-1,T}\|^2 + 2(1 + \frac{1}{\varepsilon})\kappa(\frac{1}{T})^T(1 - \beta\mu)^P L_y^2\alpha^2\|\nabla\Phi(x_{k-1})\|^2 \\
&\quad + 12(1 + \frac{1}{\varepsilon})\kappa(\frac{1}{T})^T(1 - \beta\mu)^P L_y^2\alpha^2 \frac{nLM_{f_{xy}}^2 C_{F_y}^2}{\mu^3\tilde{\xi}Q_{k-1}},
\end{aligned}
\tag{38}
$$

where $\tau = \kappa(\frac{1}{T})^T(1 - \beta\mu)^P\left((1 + \varepsilon) + 6(1 + \frac{1}{\varepsilon})L_y^2\alpha^2(L_{F_x}^2 + \frac{L_{f_{xy}}^2 C_{F_y}^2}{\mu^2} + \frac{4M_{f_{xy}}^2 L_{F_y}^2}{\mu^2} + \frac{4M_{f_{xy}}^2 C_{F_y}^2 L_{f_{yy}}^2}{\mu^4})\right)$.

The $Q_k$-step error $\|u_{k,Q_k} - u_k^*\|$ is bounded by:

$$
\|u_{k,Q_k} - u_k^*\|^2 \leq 2\frac{nLC_{F_y}^2}{\tilde{\xi}\mu^3 Q_k} + 4\left(\frac{L_{F_y}^2}{\mu^2} + \frac{C_{F_y}^2 L_{f_{yy}}^2}{\mu^4}\right)\|y_k^* - y_{k,T}\|^2.
\tag{39}
$$

**Step 3:** Combining Step 1 and Step 2.

Combining (37), (38) and (39), the upper bound of the hypergradient estimation error is derived as:

$$\|\tilde{\nabla}\Phi(x_k) - \nabla\Phi(x_k)\|^2 \leq \delta_0\tau^k + \omega\alpha^2\sum_{j=0}^{k-1}\tau^j\|\nabla\Phi(x_{k-1-j})\|^2$$

$$+ 6\omega\alpha^2\frac{nLM_{f_{xy}}^2C_{F_y}^2}{\mu^3\tilde{\xi}}\sum_{j=0}^{k-1}\tau^j\frac{1}{Q_{k-1-j}} + 6\frac{nLM_{f_{xy}}^2C_{F_y}^2}{\mu^3\tilde{\xi}Q_k},$$

with

$$\delta_0 = 3\kappa(\frac{1}{T})^T(1-\beta\mu)^P(L_{F_x}^2 + \frac{L_{f_{xy}}^2C_{F_y}^2}{\mu^2} + \frac{4M_{f_{xy}}^2L_{F_y}^2}{\mu^2} + \frac{4M_{f_{xy}}^2C_{F_y}^2L_{f_{yy}}^2}{\mu^4})\|y_0^* - y_0\|^2$$

and

$$\omega = 6(L_{F_x}^2 + \frac{L_{f_{xy}}^2C_{F_y}^2}{\mu^2} + \frac{4M_{f_{xy}}^2L_{F_y}^2}{\mu^2} + \frac{4M_{f_{xy}}^2C_{F_y}^2L_{f_{yy}}^2}{\mu^4})(1+\frac{1}{\varepsilon})\kappa(\frac{1}{T})^T(1-\beta\mu)^PL_y^2.$$

Due to the $L_\Phi$-smoothness of $\Phi$, the final convergence result can be proved.

To prove Theorems 3.4 and 3.7, we first present the following lemma.

**Lemma D.12.** (Lemma 2.2 of Ghadimi & Wang (2018)) *Under Assumptions 3.1 and 3.2, we have:*

- *For all $x, y$,*
$$\|\bar{\nabla}F(x;y) - \bar{\nabla}F(x;y^*(x))\| \leq C\|y^*(x) - y\|,$$
*where $\bar{\nabla}F(x;y) = \nabla_xF(x,y) - [\nabla_{xy}^2f(x,y)]^T[\nabla_{yy}^2f(x,y)]^{-1}\nabla_yF(x,y)$ and $C = L_{F_x} + \frac{L_{F_y}M_{f_{xy}}}{\mu} + C_{F_y}\left(\frac{L_{f_{xy}}}{\mu} + \frac{L_{f_{yy}}M_{f_{xy}}}{\mu^2}\right)$.*

- *$y^*(x)$ is $L_y$-Lipschitz continuous in $x$:*
$$\|y^*(x_1) - y^*(x_2)\| \leq L_y\|x_1 - x_2\|,$$
*where $L_y = \frac{M_{f_{xy}}}{\mu}$.*

- *$\nabla\Phi$ is $L_\Phi$-Lipschitz continuous in $x$:*
$$\|\nabla\Phi(x_1) - \nabla\Phi(x_2)\| \leq L_\Phi\|x_1 - x_2\|,$$
*where $L_\Phi = \frac{(\bar{L}_{F_y}+C)M_{f_{xy}}}{\mu} + L_{F_x} + C_{F_y}\left(\frac{\bar{L}_{f_{xy}}C_{F_y}}{\mu} + \frac{\bar{L}_{f_{yy}}M_{f_{xy}}}{\mu^2}\right)$.*

### D.3 PROOF OF THEOREM 3.4

**Lemma D.13.** *Suppose that Assumptions 3.1 and 3.2 hold. The error between the approximate hypergradient $\tilde{\nabla}\Phi(x_k)$ and the true hypergradient in Algorithm 1 can be bounded by:*

$$\|\tilde{\nabla}\Phi(x_k) - \nabla\Phi(x_k)\|^2 \leq 3\big(L_{F_x}^2 + \frac{L_{f_{xy}}^2C_{F_y}^2}{\mu^2}\big)\|y_{k,T} - y_k^*\|^2 + 3M_{f_{xy}}^2\|u_{k,Q_k} - u_k^*\|^2. \quad (40)$$

*Proof.* Let $u_k^* = [\nabla_{yy}^2f(x_k, y_k^*)]^{-1}\nabla_yF(x_k, y_k^*)$, then

$$\|\tilde{\nabla}\Phi(x_k) - \nabla\Phi(x_k)\|^2$$

$$= \big\|\nabla_xF(x_k, y_{k,T}) - [\nabla_{xy}^2f(x_k, y_{k,T})]^Tu_{k,Q_k} - \big(\nabla_xF(x_k, y_k^*) - [\nabla_{xy}^2f(x_k, y_k^*)]^Tu_k^*\big)\big\|^2$$

$$= \big\|\nabla_xF(x_k, y_{k,T}) - \nabla_xF(x_k, y_k^*) - \big([\nabla_{xy}^2f(x_k, y_{k,T})]^Tu_{k,Q_k} - [\nabla_{xy}^2f(x_k, y_k^*)]^Tu_k^*\big)\big\|^2$$

$$= \big\|\nabla_xF(x_k, y_{k,T}) - \nabla_xF(x_k, y_k^*)$$

$$- \big([\nabla_{xy}^2f(x_k, y_{k,T})]^Tu_{k,Q_k} - [\nabla_{xy}^2f(x_k, y_{k,T})]^Tu_k^* - ([\nabla_{xy}^2f(x_k, y_k^*)]^Tu_k^* - [\nabla_{xy}^2f(x_k, y_{k,T})]^Tu_k^*)\big)\big\|^2$$

$$= \big\|\nabla_xF(x_k, y_{k,T}) - \nabla_xF(x_k, y_k^*)$$

$$- [\nabla_{xy}^2f(x_k, y_{k,T})]^T(u_{k,Q_k} - u_k^*) - \big([\nabla_{xy}^2f(x_k, y_{k,T})]^T - [\nabla_{xy}^2f(x_k, y_k^*)]^T\big)u_k^*\big\|^2$$

$$\leq 3\|\nabla_xF(x_k, y_{k,T}) - \nabla_xF(x_k, y_k^*)\|^2 + 3\|\nabla_{xy}^2f(x_k, y_{k,T})\|^2\|u_{k,Q_k} - u_k^*\|^2$$

$$+ 3\|\nabla_{xy}^2f(x_k, y_{k,T}) - \nabla_{xy}^2f(x_k, y_k^*)\|^2\|u_k^*\|^2.$$

Based on Assumptions 3.1 and 3.2, it can be derived that:

$$\|\tilde{\nabla}\Phi(x_k) - \nabla\Phi(x_k)\|^2$$

$$\leq 3L_{F_x}^2\|y_{k,T} - y_k^*\|^2 + 3M_{f_{xy}}^2\|u_{k,Q_k} - u_k^*\|^2 + 3\frac{L_{f_{xy}}^2 C_{F_y}^2}{\mu^2}\|y_{k,T} - y_k^*\|^2$$

$$= 3\big(L_{F_x}^2 + \frac{L_{f_{xy}}^2 C_{F_y}^2}{\mu^2}\big)\|y_{k,T} - y_k^*\|^2 + 3M_{f_{xy}}^2\|u_{k,Q_k} - u_k^*\|^2.$$

$\square$

**Lemma D.14.** *When the qNBO algorithm (Algorithm 1) is applied to solve the problem (1), if the LL objective function $f$ takes the quadratic form (22) and $H_0 = (1/L)I$, it holds that:*

$$\sum_{i=1}^{Q_k}\|A^{-1}\nabla_y F(x_k, y_{k+1}) - u_{k,i}\| \leq \frac{C_{F_y}}{\mu}\sqrt{\frac{nLQ_k}{\mu}},$$

*with $Q_k > 1$.*

*Proof.* From Lemma D.6, it follows that:

$$\sum_{i=1}^{Q_k}\frac{(A^{-1}\nabla_y F(x_k, y_{k+1}) - u_{k,i})^T A(A^{-1}\nabla_y F(x_k, y_{k+1}) - u_{k,i})}{u_{k,i}^T B_{k,i} u_{k,i}}$$

$$= \sum_{i=1}^{Q_k}\frac{(A^{-1}\nabla_y F(x_k, y_{k+1}) - u_{k,i})^T A(A^{-1}\nabla_y F(x_k, y_{k+1}) - u_{k,i})}{\nabla_y F(x_k, y_{k+1})^T B_{k,i}^{-1}\nabla_y F(x_k, y_{k+1})}$$

$$\leq \frac{nL}{\mu},$$

with $B_{k,i} = H_{k,i}^{-1}$.

Since $\mu I \preceq A \preceq LI$ and $A \preceq B_{k,i} \preceq \frac{L}{\mu}A$, we have:

$$\sum_{i=1}^{Q_k}\frac{\mu\|A^{-1}\nabla_y F(x_k, y_{k+1}) - u_{k,i}\|^2}{\frac{1}{\mu}\|\nabla_y F(x_k, y_{k+1})\|^2} \leq \frac{nL}{\mu}.$$

Since $\|\nabla_y F(x_k, y_{k+1})\| \leq C_{F_y}$, it can be further derived that:

$$\sum_{i=1}^{Q_k}\|A^{-1}\nabla_y F(x_k, y_{k+1}) - u_{k,i}\|^2 \leq \frac{nLC_{F_y}^2}{\mu^3}. \tag{41}$$

Finally, by applying the Cauchy-Schwarz inequality, we can deduce that

$$\sum_{i=1}^{Q_k}\|A^{-1}\nabla_y F(x_k, y_{k+1}) - u_{k,i}\| \leq \frac{C_{F_y}}{\mu}\sqrt{\frac{nLQ_k}{\mu}}. \tag{42}$$

$\square$

**Lemma D.15.** *Choose the parameters $\beta$ and $P$ such that $(1 - \beta\mu)^P\|y_k - y_k^*\| \leq \frac{1}{300\sqrt{\mu}}$, and ensure $H_0$ satisfies: $\|\nabla_{yy}^2 f(x_k, y^*(x_k))^{-1/2}\big(H_0^{-1} - \nabla_{yy}^2 f(x_k, y^*(x_k))\big)\nabla_{yy}^2 f(x_k, y^*(x_k))^{-1/2}\|_F \leq \frac{1}{7}$. Then, under Assumptions 3.1 and 3.2, it holds that*

$$\|y_k^* - y_{k,T}\|^2 \leq (1+\varepsilon)\kappa(\frac{1}{T})^T(1-\beta\mu)^P\|y_{k-1,T} - y_{k-1}^*\|^2 + (1+\frac{1}{\varepsilon})\kappa(\frac{1}{T})^T(1-\beta\mu)^P L_y^2\|x_k - x_{k-1}\|^2, \tag{43}$$

*with a positive constant $\varepsilon$.*

*Proof.* Under the setting of parameters $\beta, P$ and $H_0$, the condition (20) is satisfied. Furthermore, based on Lemma D.5 and the fact that $y_{k,0} = y_k^P$ and $y_k = y_k^0$, it holds that:

$$\|y_{k,T} - y_k^*\|^2 \leq \kappa(\frac{1}{T})^T \|y_{k,0} - y_k^*\|^2 \leq \kappa(\frac{1}{T})^T (1 - \beta\mu)^P \|y_k - y_k^*\|^2,$$

where $\kappa = \frac{L}{\mu}$. Finally, since $y_k = y_{k-1,T}$, using Young's inequality yields:

$$\|y_{k,T} - y_k^*\|^2 \leq (1 + \varepsilon)\kappa(\frac{1}{T})^T (1 - \beta\mu)^P \|y_{k-1,T} - y_{k-1}^*\|^2 + (1 + \frac{1}{\varepsilon})\kappa(\frac{1}{T})^T (1 - \beta\mu)^P \|y_{k-1}^* - y_k^*\|^2$$

$$\leq (1 + \varepsilon)\kappa(\frac{1}{T})^T (1 - \beta\mu)^P \|y_{k-1,T} - y_{k-1}^*\|^2 + (1 + \frac{1}{\varepsilon})\kappa(\frac{1}{T})^T (1 - \beta\mu)^P L_y^2 \|x_{k-1} - x_k\|^2,$$

where $\varepsilon$ is a positive constant and the last inequality follows from Lemma 2.2 in Ghadimi & Wang (2018). $\qquad\square$

**Lemma D.16.** *If the LL function $f$ takes the quadratic form and $T \geq t_b$, under Assumptions 3.1 and 3.2, it is derived that*

$$\|y_k^* - y_{k,T}\|^2 \leq (1 + \varepsilon)c_t^2 \kappa^3 (\frac{1}{T})^T \|y_{k-1,T} - y_{k-1}^*\|^2 + (1 + \frac{1}{\varepsilon})c_t^2 \kappa^3 (\frac{1}{T})^T L_y^2 \|x_k - x_{k-1}\|^2, \quad (44)$$

*with $t_b = 4n\ln\frac{L}{\mu}$ and a positive constant $\varepsilon$.*

*Proof.* From Lemma D.2, if $T \geq t_b$, it holds that:

$$\|y_{k,T} - y_k^*\|^2 \leq c_t^2 \kappa^3 (\frac{1}{T})^T \|y_{k,0} - y_k^*\|^2,$$

where $c_t = 2t_b^{\frac{T}{2}}$. Furthermore, since $y_{k,0} = y_{k-1,T}$, using Young's inequality yields:

$$\|y_{k,T} - y_k^*\|^2 \leq (1 + \varepsilon)c_t^2 \kappa^3 (\frac{1}{T})^T \|y_{k-1,T} - y_{k-1}^*\|^2 + (1 + \frac{1}{\varepsilon})c_t^2 \kappa^3 (\frac{1}{T})^T \|y_{k-1}^* - y_k^*\|^2$$

$$\leq (1 + \varepsilon)c_t^2 \kappa^3 (\frac{1}{T})^T \|y_{k-1,T} - y_{k-1}^*\|^2 + (1 + \frac{1}{\varepsilon})c_t^2 \kappa^3 (\frac{1}{T})^T L_y^2 \|x_{k-1} - x_k\|^2,$$

where $\varepsilon$ is a positive constant and the last inequality follows from Lemma 2.2 in Ghadimi & Wang (2018). $\qquad\square$

**Lemma D.17.** *Choose the parameters $\beta, P$ such that $(1 - \beta\mu)^P \|y_k - y_k^*\| \leq K_1$, and $H_0 = LI$, under Assumptions 3.1 and 3.2, it is derived that*

$$\|y_k^* - y_{k,T}\|^2 \leq (1+\varepsilon)c_l^2 \kappa^3 (1-\beta\mu)^P (\frac{1}{T})^T \|y_{k-1,T} - y_{k-1}^*\|^2 + (1+\frac{1}{\varepsilon})c_l^2 \kappa^3 (1-\beta\mu)^P (\frac{1}{T})^T L_y^2 \|x_{k-1} - x_k\|^2,$$

$$(45)$$

*with a positive constant $\varepsilon$, $T \geq K_0$, $K_0 := 8n\ln\frac{2L}{\mu}$, $c_l = 2t_c^{\frac{T}{2}}$.*

*Proof.* From Lemma D.3, if $T \geq K_0$, it holds that:

$$\|y_{k,T} - y_k^*\|^2 \leq c_l^2 \kappa^3 (\frac{1}{T})^T \|y_{k,0} - y_k^*\|^2 \leq c_l^2 \kappa^3 (\frac{1}{T})^T (1 - \beta\mu)^P \|y_k - y_k^*\|^2,$$

where $c_l = 2t_c^{\frac{T}{2}}$. Furthermore, using Young's inequality yields:

$$\|y_{k,T} - y_k^*\|^2 \leq (1 + \varepsilon)c_l^2 \kappa^3 (1 - \beta\mu)^P (\frac{1}{T})^T \|y_{k-1,T} - y_{k-1}^*\|^2 + (1 + \frac{1}{\varepsilon})c_l^2 \kappa^3 (1 - \beta\mu)^P (\frac{1}{T})^T \|y_{k-1}^* - y_k^*\|^2$$

$$\leq (1 + \varepsilon)c_l^2 \kappa^3 (1 - \beta\mu)^P (\frac{1}{T})^T \|y_{k-1,T} - y_{k-1}^*\|^2 + (1 + \frac{1}{\varepsilon})c_l^2 \kappa^3 (1 - \beta\mu)^P (\frac{1}{T})^T L_y^2 \|x_{k-1} - x_k\|^2,$$

where $\varepsilon$ is a positive constant and the last inequality follows from Lemma 2.2 in Ghadimi & Wang (2018). $\qquad\square$

**Lemma D.18.** *(Error of $u_{k,Q_k}$) Suppose that the lower level function $f$ has the quadratic form and Assumptions 3.1 and 3.2 hold. If $u_{k,Q_k} = \bar{u}_k$ and*

$$\bar{u}_k := \operatorname*{argmin}_i \|A^{-1}\nabla_y F(x_k, y_{k+1}) - u_{k,i}\|, \tag{46}$$

*then for $Q_k > 1$, the following inequality holds:*

$$\|u_{k,Q_k} - u_k^*\|^2 \leq 2\frac{nLC_{F_y}^2}{\mu^3 Q_k} + 2\frac{L_{F_y}^2}{\mu^2}\|y_k^* - y_{k,T}\|^2. \tag{47}$$

*Proof.* From Lemma D.14, we have:

$$\|A^{-1}\nabla_y F(x_k, y_{k+1}) - \bar{u}_k\|^2 \leq \frac{nLC_{F_y}^2}{\mu^3 Q_k}. \tag{48}$$

Moreover, under Assumption 3.1 and given that $y_{k+1} = y_{k,T}$, it holds that:

$$\|u_{k,Q_k} - u_k^*\|^2 \leq 2\|\bar{u}_k - A^{-1}\nabla_y F(x_k, y_{k+1})\|^2 + 2\|A^{-1}\nabla_y F(x_k, y_{k+1}) - A^{-1}\nabla_y F(x_k, y_k^*)\|^2$$

$$\leq 2\frac{nLC_{F_y}^2}{\mu^3 Q_k} + 2\frac{L_{F_y}^2}{\mu^2}\|y_k^* - y_{k,T}\|^2.$$

$\square$

**Theorem D.19.** *(Restatement of Theorem 3.4 with full parameter specifications) Suppose that the LL function $f$ in (1) takes the quadratic form:*

$$f(x, y) = \frac{1}{2}y^T A y - y^T x, \tag{49}$$

*where $\mu I \preceq A \preceq LI$ such that Assumption 3.2 holds. Choose the stepsize $\alpha > 0$, the positive constant $\varepsilon > 0$, $\bar{H}_0 = LI$ and $T \geq t_b$ ($t_b = 4n\ln\kappa$) such that*

$$\tau < 1 \quad \text{and} \quad \alpha L_\Phi + \omega\alpha^2\left(\frac{1}{2} + \alpha L_\Phi\right)\frac{1}{1-\tau} \leq \frac{1}{4},$$

*where $\tau = c_t^2\kappa^3(\frac{1}{T})^T\left((1+\varepsilon) + 6(1+\frac{1}{\varepsilon})L_y^2\alpha^2(L_{F_x}^2 + \frac{L_{f_{xy}}^2 C_{F_y}^2}{\mu^2} + \frac{2M_{f_{xy}}^2 L_{F_y}^2}{\mu^2})\right)$, $\omega = 6(L_{F_x}^2 + \frac{L_{f_{xy}}^2 C_{F_y}^2}{\mu^2} + \frac{2M_{f_{xy}}^2 L_{F_y}^2}{\mu^2})(1+\frac{1}{\varepsilon})c_t^2\kappa^3(\frac{1}{T})^T L_y^2$, $\kappa = \frac{L}{\mu}$ and $c_t = 2t_b^{\frac{T}{2}}$. Then, under Assumptions 3.1 and 3.3, the iterate generated by the qNBO (BFGS) algorithm (Algorithm 1) has the following convergence rate:*

$$\frac{1}{K}\sum_{k=0}^{K-1}\|\nabla\Phi(x_k)\|^2 \leq \frac{4(\Phi(x_0) - \Phi(x^*))}{\alpha K} + \frac{3\delta_0}{K(1-\tau)} + \frac{1}{K}\sum_{k=0}^{K-1}\frac{18nLM_{f_{xy}}^2 C_{F_y}^2}{\mu^3 Q_k}, \tag{50}$$

*with the initial error $\delta_0 = 3c_t^2\kappa^3(\frac{1}{T})^T(L_{F_x}^2 + \frac{L_{f_{xy}}^2 C_{F_y}^2}{\mu^2} + \frac{2M_{f_{xy}}^2 L_{F_y}^2}{\mu^2})\|y_0^* - y_0\|^2$. Specifically, if $Q_k = k + 1$, we have:*

$$\frac{1}{K}\sum_{k=0}^{K-1}\|\nabla\Phi(x_k)\|^2 \leq \frac{4(\Phi(x_0) - \inf_x \Phi(x))}{\alpha K} + \frac{3\delta_0}{K(1-\tau)} + \frac{18nLM_{f_{xy}}^2 C_{F_y}^2 \ln K}{\mu^3 K}. \tag{51}$$

*Proof.* Substituting the inequality (47) into (40) yields:

$$\begin{aligned}
\|\tilde{\nabla}\Phi(x_k) - \nabla\Phi(x_k)\|^2 &\leq 3\left(L_{F_x}^2 + \frac{L_{f_{xy}}^2 C_{F_y}^2}{\mu^2}\right)\|y_{k,T} - y_k^*\|^2 \\
&\quad + 3M_{f_{xy}}^2\left(2\frac{nLC_{F_y}^2}{\mu^3 Q_k} + 2\frac{L_{F_y}^2}{\mu^2}\|y_k^* - y_{k,T}\|^2\right) \\
&\leq \left(3L_{F_x}^2 + \frac{3L_{f_{xy}}^2 C_{F_y}^2}{\mu^2} + \frac{6M_{f_{xy}}^2 L_{F_y}^2}{\mu^2}\right)\|y_k^* - y_{k,T}\|^2 \\
&\quad + 6\frac{nLM_{f_{xy}}^2 C_{F_y}^2}{\mu^3 Q_k}.
\end{aligned} \tag{52}$$

Then, by plugging the inequality (52) into (44), we obtain that:

$$
\begin{aligned}
\|y_k^* - y_{k,T}\|^2 &\leq (1+\varepsilon)c_t^2\kappa^3(\tfrac{1}{T})^T\|y_{k-1,T} - y_{k-1}^*\|^2 + 2(1+\tfrac{1}{\varepsilon})c_t^2\kappa^3(\tfrac{1}{T})^T L_y^2\alpha^2\|\nabla\Phi(x_{k-1})\|^2 \\
&\quad + 2(1+\tfrac{1}{\varepsilon})c_t^2\kappa^3(\tfrac{1}{T})^T L_y^2\alpha^2\|\tilde{\nabla}\Phi(x_{k-1}) - \nabla\Phi(x_{k-1})\|^2 \\
&\leq \tau\|y_{k-1}^* - y_{k-1,T}\|^2 + 2(1+\tfrac{1}{\varepsilon})c_t^2\kappa^3(\tfrac{1}{T})^T L_y^2\alpha^2\|\nabla\Phi(x_{k-1})\|^2 \\
&\quad + 12(1+\tfrac{1}{\varepsilon})c_t^2\kappa^3(\tfrac{1}{T})^T L_y^2\alpha^2 \frac{nLM_{f_{xy}}^2 C_{F_y}^2}{\mu^3 Q_{k-1}},
\end{aligned}
\tag{53}
$$

where $\tau = c_t^2\kappa^3(\tfrac{1}{T})^T\big((1+\varepsilon) + 6(1+\tfrac{1}{\varepsilon})L_y^2\alpha^2(L_{F_x}^2 + \tfrac{L_{f_{xy}}^2 C_{F_y}^2}{\mu^2} + \tfrac{2M_{f_{xy}}^2 L_{F_y}^2}{\mu^2})\big)$.

By telescoping (53) over $k$, it follows that:

$$
\begin{aligned}
\|y_k^* - y_{k,T}\|^2 &\leq \tau^k\|y_0^* - y_{0,T}\|^2 + 2(1+\tfrac{1}{\varepsilon})c_t^2\kappa^3(\tfrac{1}{T})^T L_y^2\alpha^2 \sum_{j=0}^{k-1}\tau^j\|\nabla\Phi(x_{k-1-j})\|^2 \\
&\quad + 12(1+\tfrac{1}{\varepsilon})c_t^2\kappa^3(\tfrac{1}{T})^T L_y^2\alpha^2 \frac{nLM_{f_{xy}}^2 C_{F_y}^2}{\mu^3}\sum_{j=0}^{k-1}\tau^j \frac{1}{Q_{k-1-j}}.
\end{aligned}
$$

Combining the inequality (52) and $\|y_0^* - y_{0,T}\|^2 \leq c_t^2\kappa^3(\tfrac{1}{T})^T\|y_0^* - y_{0,0}\|^2$, we can further derive that

$$
\begin{aligned}
\|\tilde{\nabla}\Phi(x_k) - \nabla\Phi(x_k)\|^2 &\leq \delta_0\tau^k + \omega\alpha^2\sum_{j=0}^{k-1}\tau^j\|\nabla\Phi(x_{k-1-j})\|^2 \\
&\quad + 6\omega\alpha^2 \frac{nLM_{f_{xy}}^2 C_{F_y}^2}{\mu^3}\sum_{j=0}^{k-1}\tau^j \frac{1}{Q_{k-1-j}} + 6\frac{nLM_{f_{xy}}^2 C_{F_y}^2}{\mu^3 Q_k},
\end{aligned}
\tag{54}
$$

with $\delta_0 = 3c_t^2\kappa^3(\tfrac{1}{T})^T(L_{F_x}^2 + \tfrac{L_{f_{xy}}^2 C_{F_y}^2}{\mu^2} + \tfrac{2M_{f_{xy}}^2 L_{F_y}^2}{\mu^2})\|y_0^* - y_{0,0}\|^2$ and $\omega = 6(L_{F_x}^2 + \tfrac{L_{f_{xy}}^2 C_{F_y}^2}{\mu^2} + \tfrac{2M_{f_{xy}}^2 L_{F_y}^2}{\mu^2})(1+\tfrac{1}{\varepsilon})c_t^2\kappa^3(\tfrac{1}{T})^T L_y^2$.

Since $\nabla\Phi(\cdot)$ is $L_\Phi-$lipschitz continuous (Lemma 2.2 in Ghadimi & Wang (2018)), we have:

$$
\begin{aligned}
\Phi(x_{k+1}) &\leq \Phi(x_k) + \langle\nabla\Phi(x_k), x_{k+1} - x_k\rangle + \frac{L_\Phi}{2}\|x_{k+1} - x_k\|^2 \\
&\leq \Phi(x_k) - \alpha\langle\nabla\Phi(x_k), \tilde{\nabla}\Phi(x_k) - \nabla\Phi(x_k)\rangle - \alpha\|\nabla\Phi(x_k)\|^2 + \alpha^2 L_\Phi\|\nabla\Phi(x_k)\|^2 \\
&\quad + \alpha^2 L_\Phi\|\nabla\Phi(x_k) - \tilde{\nabla}\Phi(x_k)\|^2 \\
&\leq \Phi(x_k) - \left(\frac{\alpha}{2} - \alpha^2 L_\Phi\right)\|\nabla\Phi(x_k)\|^2 + \left(\frac{\alpha}{2} + \alpha^2 L_\Phi\right)\|\nabla\Phi(x_k) - \tilde{\nabla}\Phi(x_k)\|^2.
\end{aligned}
\tag{55}
$$

Using the inequality (54), it holds that:

$$
\begin{aligned}
\Phi(x_{k+1}) &\leq \Phi(x_k) - \left(\frac{\alpha}{2} - \alpha^2 L_\Phi\right)\|\nabla\Phi(x_k)\|^2 + \left(\frac{\alpha}{2} + \alpha^2 L_\Phi\right)\|\nabla\Phi(x_k) - \tilde{\nabla}\Phi(x_k)\|^2 \\
&\leq \Phi(x_k) - \left(\frac{\alpha}{2} - \alpha^2 L_\Phi\right)\|\nabla\Phi(x_k)\|^2 + \left(\frac{\alpha}{2} + \alpha^2 L_\Phi\right)\delta_0\tau^k \\
&\quad + \omega\alpha^2\left(\frac{\alpha}{2} + \alpha^2 L_\Phi\right)\sum_{j=0}^{k-1}\tau^j\|\nabla\Phi(x_{k-1-j})\|^2 + \left(\frac{\alpha}{2} + \alpha^2 L_\Phi\right)\frac{6nLM_{f_{xy}}^2 C_{F_y}^2}{\mu^3 Q_k} \\
&\quad + 6\omega\left(\frac{\alpha}{2} + \alpha^2 L_\Phi\right)\alpha^2 \frac{nLM_{f_{xy}}^2 C_{F_y}^2}{\mu^3}\sum_{j=0}^{k-1}\tau^j \frac{1}{Q_{k-1-j}}.
\end{aligned}
\tag{56}
$$

Finally, summing the inequality (56) from $k = 0$ to $k = K - 1$ yields:

$$
\begin{aligned}
\left(\frac{\alpha}{2} - \alpha^2 L_\Phi\right) \sum_{k=0}^{K-1} \|\nabla\Phi(x_k)\|^2 \leq & \Phi(x_0) - \Phi(x_K) + \left(\frac{\alpha}{2} + \alpha^2 L_\Phi\right) \frac{\delta_0}{1 - \tau} \\
& + \omega\alpha^2 \left(\frac{\alpha}{2} + \alpha^2 L_\Phi\right) \sum_{k=0}^{K-1} \sum_{j=0}^{k-1} \tau^j \|\nabla\Phi(x_{k-1-j})\|^2 \\
& + 6\omega \left(\frac{\alpha}{2} + \alpha^2 L_\Phi\right) \alpha^2 \frac{nLM_{f_{xy}}^2 C_{F_y}^2}{\mu^3} \sum_{k=0}^{K-1} \sum_{j=0}^{k-1} \tau^j \frac{1}{Q_{k-1-j}} \\
& + \sum_{k=0}^{K-1} \left(\frac{\alpha}{2} + \alpha^2 L_\Phi\right) \frac{6nLM_{f_{xy}}^2 C_{F_y}^2}{\mu^3 Q_k}.
\end{aligned}
\tag{57}
$$

Furthermore, from the inequality $\sum_{k=0}^{K-1} \sum_{j=0}^{k-1} a_j b_{k-1-j} \leq \sum_{k=0}^{K-1} a_k \sum_{j=0}^{K-1} b_j$, we obtain

$$
\sum_{k=0}^{K-1} \sum_{j=0}^{k-1} \tau^j \|\nabla\Phi(x_{k-1-j})\|^2 \leq \sum_{k=0}^{K-1} \tau^k \sum_{k=0}^{K-1} \|\nabla\Phi(x_k)\|^2 \leq \frac{1}{1 - \tau} \sum_{k=0}^{K-1} \|\nabla\Phi(x_k)\|^2,
$$

$$
\sum_{k=0}^{K-1} \sum_{j=0}^{k-1} \tau^j \frac{1}{Q_{k-1-j}} \leq \sum_{k=0}^{K-1} \tau^k \sum_{k=0}^{K-1} \frac{1}{Q_k} \leq \frac{1}{1 - \tau} \sum_{k=0}^{K-1} \frac{1}{Q_k}.
$$

Thus, we can conclude that

$$
\begin{aligned}
& \left(\frac{1}{2} - \alpha L_\Phi - \omega\alpha^2 \left(\frac{1}{2} + \alpha L_\Phi\right) \frac{1}{1 - \tau}\right) \frac{1}{K} \sum_{k=0}^{K-1} \|\nabla\Phi(x_k)\|^2 \\
\leq & \frac{\Phi(x_0) - \Phi(x_K)}{\alpha K} + \left(\frac{1}{2} + \alpha L_\Phi\right) \frac{\delta_0}{K(1 - \tau)} \\
& + \left(\frac{1}{2} + \alpha L_\Phi\right) \frac{1}{K} \sum_{k=0}^{K-1} \frac{6nLM_{f_{xy}}^2 C_{F_y}^2}{\mu^3 Q_k} + \frac{6\omega}{1 - \tau} \left(\frac{1}{2} + \alpha L_\Phi\right) \alpha^2 \frac{1}{K} \sum_{k=0}^{K-1} \frac{nLM_{f_{xy}}^2 C_{F_y}^2}{\mu^3 Q_k}.
\end{aligned}
\tag{58}
$$

Moreover, if $\alpha L_\Phi + \omega\alpha^2 \left(\frac{1}{2} + \alpha L_\Phi\right) \frac{1}{1 - \tau} \leq \frac{1}{4}$, then

$$
\frac{1}{K} \sum_{k=0}^{K-1} \|\nabla\Phi(x_k)\|^2 \leq \frac{4(\Phi(x_0) - \Phi(x^*))}{\alpha K} + \frac{3\delta_0}{K(1 - \tau)} + \frac{1}{K} \sum_{k=0}^{K-1} \frac{18nLM_{f_{xy}}^2 C_{F_y}^2}{\mu^3 Q_k},
\tag{59}
$$

where $\Phi(x^*) = \inf_x \Phi(x)$. Finally, if $Q_k = k + 1$, then (51) is established.

$\square$

Next, we consider the case with warm-up steps and provide the corresponding theorem, which proves to be similar to Theorem D.24, so a detailed proof is not provided.

**Theorem D.20.** *(Warm-up for quadratic $f$) Suppose that the LL function $f$ in (1) takes the quadratic form:*

$$
f(x, y) = \frac{1}{2} y^T A y - y^T x,
\tag{60}
$$

*where $\mu I \preceq A \preceq LI$ such that Assumption 3.2 holds. Choose the stepsize $\beta$ and warm-start iteration steps $P$ such that $(1 - \beta\mu)^P \|y_k - y_k^*\| \leq \frac{1}{300\sqrt{\mu}}$, and ensure the initial Hessian approximation matrix $H_0$ satisfies: $\|\nabla_{yy}^2 f(x_k, y^*(x_k))^{-1/2} \left(H_0^{-1} - \nabla_{yy}^2 f(x_k, y^*(x_k))\right) \nabla_{yy}^2 f(x_k, y^*(x_k))^{-1/2}\|_F \leq \frac{1}{7}$. Choose the stepsize $\alpha > 0$ and the positive constant $\varepsilon > 0$ such that*

$$
\tau < 1 \quad \text{and} \quad \alpha L_\Phi + \omega\alpha^2 \left(\frac{1}{2} + \alpha L_\Phi\right) \frac{1}{1 - \tau} \leq \frac{1}{4},
$$

where $\tau = \kappa(\frac{1}{T})^T (1-\beta\mu)^P \big((1+\varepsilon) + 6(1+\frac{1}{\varepsilon})L_y^2\alpha^2(L_{F_x}^2 + \frac{L_{f_{xy}}^2 C_{F_y}^2}{\mu^2} + \frac{2M_{f_{xy}}^2 L_{F_y}^2}{\mu^2})\big)$, $\omega = 6(L_{F_x}^2 + \frac{L_{f_{xy}}^2 C_{F_y}^2}{\mu^2} + \frac{2M_{f_{xy}}^2 L_{F_y}^2}{\mu^2})(1+\frac{1}{\varepsilon})\kappa(\frac{1}{T})^T(1-\beta\mu)^P L_y^2$ and $\kappa = \frac{L}{\mu}$. Then, under Assumptions 3.1 and 3.3, the iterate generated by the qNBO (BFGS) algorithm (Algorithm 1) has the following convergence rate:

$$\frac{1}{K}\sum_{k=0}^{K-1}\|\nabla\Phi(x_k)\|^2 \leq \frac{4(\Phi(x_0)-\Phi(x^*))}{\alpha K} + \frac{3\delta_0}{K(1-\tau)} + \frac{1}{K}\sum_{k=0}^{K-1}\frac{18nLM_{f_{xy}}^2 C_{F_y}^2}{\mu^3 Q_k}, \qquad (61)$$

with the initial error $\delta_0 = 3\kappa(\frac{1}{T})^T(1-\beta\mu)^P(L_{F_x}^2 + \frac{L_{f_{xy}}^2 C_{F_y}^2}{\mu^2} + \frac{2M_{f_{xy}}^2 L_{F_y}^2}{\mu^2})\|y_0^* - y_0\|^2$.

### D.4 PROOF OF THEOREM 3.7

**Proposition D.21.** (Example 4.1 of Rodomanov & Nesterov (2021a)) *Suppose that $\forall x$, the LL function $f$ is $\mu$-strongly convex w.r.t. $y$ and its Hessian is $L_{f_{yy}}$-Lipschitz continuous w.r.t. $y$. Then $f$ is strongly self-concordant with constant $M = \frac{L_{f_{yy}}}{\mu^{3/2}}$, i.e.,*

$$\nabla_{yy}^2 f(x,y_1) - \nabla_{yy}^2 f(x,y_2) \preceq M\|y_1 - y_2\|_z \nabla_{yy}^2 f(x,w), \forall y_1, y_2, z, w \in \mathbb{R}^n,$$

*where $\|y\|_z := \langle\nabla_{yy}^2 f(x,z)y, y\rangle^{1/2}$.*

*Proof.* Using the Lipschitz continuity of the Hessian, we have

$$\nabla_{yy}^2 f(x,y_1) - \nabla_{yy}^2 f(x,y_2) \preceq L_{f_{yy}}\|y_1 - y_2\|I$$
$$\preceq \frac{L_{f_{yy}}}{\mu^{1/2}}\langle\nabla_{yy}^2 f(x,z)(y_1-y_2), y_1-y_2\rangle^{1/2}I$$
$$= \frac{L_{f_{yy}}}{\mu^{1/2}}\|y_1 - y_2\|_z I \preceq \frac{L_{f_{yy}}}{\mu^{3/2}}\|y_1 - y_2\|_z\nabla_{yy}^2 f(x,w),$$

where the second and the last inequalities follow from the fact that $\mu I \preceq \nabla_{yy}^2 f(x,y)$. This demonstrates that $f$ is strongly self-concordant with constant $M = \frac{L_{f_{yy}}}{\mu^{3/2}}$. $\qquad\square$

**Lemma D.22.** *If the Assumptions 3.1 and 3.2 hold, then $u_{k,i}$ generated in the step 2 of the Algorithm 1 satisfies:*

$$\sum_{i=1}^{Q_k}\|\nabla_{yy}^2 f(x_k, y_{k+1})^{-1}\nabla_y F(x_k, y_{k+1}) - u_{k,i}\|^2 \leq \frac{nLC_{F_y}^2}{\tilde{\xi}\mu^3}, \qquad (62)$$

*where $Q_k > 1$, $\tilde{\xi} = \min_{i=1,\cdots,Q_k}\frac{1}{2(\xi_i^2+\xi_i)}$ and $\xi_i = e^{M\sum_{j=0}^{i-1}\|\zeta_j u_{k,j}\|_{y_{k+1}}}$.*

*Proof.* Note that in Algorithm 3:

$$J_i := \int_0^1\nabla_{yy}^2 f(x_k, y_{k+1} + ts_i)\,dt, \quad J_{i+1} := \int_0^1\nabla_{yy}^2 f(x_k, y_{k+1} + ts_{i+1})\,dt,$$

with $s_i = \zeta_i H_{k,i}\nabla_y F(x_k, y_{k+1})$.

When the step size $\zeta_i$ is chosen appropriately, based on the definition of $J_i$ and the properties of $f$ as stated in Assumption 3.2, it can be concluded that $J_i$ is nearly equal to $J_{i+1}$, i.e., $\Delta_i \approx 0$, and $\mu I \preceq J_i \preceq LI$. From Lemma D.11, it follows that

$$\sum_{i=1}^{Q_k}\frac{(J_i^{-1}\nabla_y F(x_k, y_{k+1}) - u_{k,i})^T J_i(J_i^{-1}\nabla_y F(x_k, y_{k+1}) - u_{k,i})}{\nabla_y F(x_k, y_{k+1})^T J_i^{-1}\nabla_y F(x_k, y_{k+1})} \leq \frac{nL}{\tilde{\xi}\mu}. \qquad (63)$$

Since $\mu I \preceq J_i \preceq LI$, we have

$$\sum_{i=1}^{Q_k}\frac{\mu\|J_i^{-1}\nabla_y F(x_k, y_{k+1}) - u_{k,i}\|^2}{\frac{1}{\mu}\|\nabla_y F(x_k, y_{k+1})\|^2} \leq \frac{nL}{\tilde{\xi}\mu}. \qquad (64)$$

Moreover, it follows from Assumption 3.1:

$$\sum_{i=1}^{Q_k} \|J_i^{-1} \nabla_y F(x_k, y_{k+1}) - u_{k,i}\|^2 \leq \frac{nLC_{F_y}^2}{\tilde{\xi}\mu^3}.$$

If the parameter $M$ of function $f$ or $\zeta_i u_{k,i}$ is sufficiently small, $J_i$ can be considered as an approximation of $\nabla_{yy}^2 f(x_k, y_{k+1})$. Therefore, $\theta(J_i, B_i, u_i)$ can be used to characterize the approximation between $H_{k,i}$ and $[\nabla_{yy}^2 f(x_k, y_{k+1})]^{-1}$ along the gradient direction $\nabla_y F(x_k, y_{k+1})$, i.e.,

$$\sum_{i=1}^{Q_k} \|\nabla_{yy}^2 f(x_k, y_{k+1})^{-1} \nabla_y F(x_k, y_{k+1}) - u_{k,i}\|^2 \leq \frac{nLC_{F_y}^2}{\tilde{\xi}\mu^3}, \tag{65}$$

where $\tilde{\xi} = \min_{i=1,\cdots,Q_k} \frac{1}{2(\xi_i^2 + \xi_i)}$ and $\xi_i = e^{M \sum_{j=0}^{i-1} \|\zeta_j u_{k,j}\|_{y_{k+1}}}$. $\qquad\square$

**Lemma D.23.** *Suppose that Assumption 3.2 holds. Note that $u_{k+1} = u_{k,Q_k}$ in the step 2 of Algorithm 1. If $u_{k,Q_k} = \bar{u}_k$ with*

$$\bar{u}_k := \operatorname*{argmin}_i \|\nabla_{yy}^2 f(x_k, y_{k+1})^{-1} \nabla_y F(x_k, y_{k+1}) - u_{k,i}\|^2, \tag{66}$$

*then*

$$\|u_{k,Q_k} - u_k^*\|^2 \leq 2\frac{nLC_{F_y}^2}{\tilde{\xi}\mu^3 Q_k} + 4\left(\frac{L_{F_y}^2}{\mu^2} + \frac{C_{F_y}^2 L_{f_{yy}}^2}{\mu^4}\right)\|y_k^* - y_{k,T}\|^2, \forall Q_k > 1, \tag{67}$$

*where $\tilde{\xi} = \min_{i=1,\cdots,Q_k} \frac{1}{2(\xi_i^2 + \xi_i)}$ and $\xi_i = e^{M \sum_{j=0}^{i-1} \|\zeta_j u_{k,j}\|_{y_{k+1}}}$.*

*Proof.* Combining Lemma D.22 and the definition of $\bar{u}_k$ yields:

$$\|\nabla_{yy}^2 f(x_k, y_{k+1})^{-1} \nabla_y F(x_k, y_{k+1}) - u_{k,Q_k}\|^2 \leq \frac{nLC_{F_y}^2}{\tilde{\xi}\mu^3 Q_k}. \tag{68}$$

Under Assumptions 3.1 and 3.2, since $y_{k+1} = y_{k,T}$, it holds that

$$\begin{aligned}
&\|\nabla_{yy}^2 f(x_k, y_{k+1})^{-1} \nabla_y F(x_k, y_{k+1}) - \nabla_{yy}^2 f(x_k, y_k^*)^{-1} \nabla_y F(x_k, y_k^*)\|^2 \\
\leq& 2\|\nabla_{yy}^2 f(x_k, y_{k+1})^{-1} \nabla_y F(x_k, y_{k+1}) - \nabla_{yy}^2 f(x_k, y_{k+1})^{-1} \nabla_y F(x_k, y_k^*)\|^2 \\
&+ 2\|\nabla_{yy}^2 f(x_k, y_{k+1})^{-1} \nabla_y F(x_k, y_k^*) - \nabla_{yy}^2 f(x_k, y_k^*)^{-1} \nabla_y F(x_k, y_k^*)\|^2 \\
\leq& 2\frac{L_{F_y}^2}{\mu^2}\|y_k^* - y_{k,T}\|^2 + 2C_{F_y}^2 \|\nabla_{yy}^2 f(x_k, y_{k+1})^{-1} - \nabla_{yy}^2 f(x_k, y_k^*)^{-1}\|^2 \\
\leq& 2\frac{L_{F_y}^2}{\mu^2}\|y_k^* - y_{k,T}\|^2 \\
&+ 2C_{F_y}^2 \|\nabla_{yy}^2 f(x_k, y_{k+1})^{-1}\|^2 \|\nabla_{yy}^2 f(x_k, y_{k+1}) - \nabla_{yy}^2 f(x_k, y_k^*)\|^2 \|\nabla_{yy}^2 f(x_k, y_k^*)^{-1}\|^2 \\
\leq& 2\frac{L_{F_y}^2}{\mu^2}\|y_k^* - y_{k,T}\|^2 + 2\frac{C_{F_y}^2 L_{f_{yy}}^2}{\mu^4}\|y_k^* - y_{k,T}\|^2 \\
=& 2\left(\frac{L_{F_y}^2}{\mu^2} + \frac{C_{F_y}^2 L_{f_{yy}}^2}{\mu^4}\right)\|y_k^* - y_{k,T}\|^2.
\end{aligned} \tag{69}$$

Finally, from Assumption 3.1 and the inequality (69), it is derived that

$$\begin{aligned}
\|u_{k,Q_k} - u_k^*\|^2 \leq& 2\|u_{k,Q_k} - \nabla_{yy}^2 f(x_k, y_{k+1})^{-1} \nabla_y F(x_k, y_{k+1})\|^2 \\
&+ 2\|\nabla_{yy}^2 f(x_k, y_{k+1})^{-1} \nabla_y F(x_k, y_{k+1}) - \nabla_{yy}^2 f(x_k, y_k^*)^{-1} \nabla_y F(x_k, y_k^*)\|^2 \\
\leq& 2\frac{nLC_{F_y}^2}{\tilde{\xi}\mu^3 Q_k} + 4\left(\frac{L_{F_y}^2}{\mu^2} + \frac{C_{F_y}^2 L_{f_{yy}}^2}{\mu^4}\right)\|y_k^* - y_{k,T}\|^2.
\end{aligned} \tag{70}$$

$\qquad\square$

**Theorem D.24.** *(Restatement of Theorem 3.7 with full parameter specifications) Suppose that Assumptions 3.1, 3.2 and 3.3 hold. Choose the stepsize $\beta$ and warm-up iteration steps $P$ such that $(1 - \beta\mu)^P \|y_k - y_k^*\| \leq \frac{1}{300\sqrt{\mu}}$, and ensure the initial Hessian approximation matrix $H_0$ satisfies: $\|\nabla_{yy}^2 f(x_k, y^*(x_k))^{-1/2} \big( H_0^{-1} - \nabla_{yy}^2 f(x_k, y^*(x_k)) \big) \nabla_{yy}^2 f(x_k, y^*(x_k))^{-1/2}\|_F \leq \frac{1}{7}$. Define $\tau = \kappa(\frac{1}{T})^T (1 - \beta\mu)^P \big( (1 + \varepsilon) + 6(1 + \frac{1}{\varepsilon}) L_y^2 \alpha^2 (L_{F_x}^2 + \frac{L_{f_{xy}}^2 C_{F_y}^2}{\mu^2} + \frac{4M_{f_{xy}}^2 L_{F_y}^2}{\mu^2} + \frac{4M_{f_{xy}}^2 C_{F_y}^2 L_{f_{yy}}^2}{\mu^4}) \big)$ and $\omega = 6(L_{F_x}^2 + \frac{L_{f_{xy}}^2 C_{F_y}^2}{\mu^2} + \frac{4M_{f_{xy}}^2 L_{F_y}^2}{\mu^2} + \frac{4M_{f_{xy}}^2 C_{F_y}^2 L_{f_{yy}}^2}{\mu^4})(1 + \frac{1}{\varepsilon})\kappa(\frac{1}{T})^T (1 - \beta\mu)^P L_y^2$. Choose the stepsize $\alpha > 0$, the positive constant $\varepsilon > 0$ and iterate $T > 0$ such that*

$$\tau < 1 \quad \text{and} \quad \alpha L_\Phi + \omega\alpha^2 \left( \frac{1}{2} + \alpha L_\Phi \right) \frac{1}{1 - \tau} \leq \frac{1}{4}.$$

*Then, the solution $x_k$ generated by Algorithm 1 achieves the following convergence rate:*

$$\frac{1}{K} \sum_{k=0}^{K-1} \|\nabla\Phi(x_k)\|^2 \leq \frac{4(\Phi(x_0) - \inf_x \Phi(x))}{\alpha K} + \frac{3\delta_0}{K(1 - \tau)} + \frac{1}{K} \sum_{k=0}^{K-1} \frac{18nLM_{f_{xy}}^2 C_{F_y}^2}{\mu^3 \tilde{\xi} Q_k}, \quad (71)$$

*where $\delta_0 = 3\kappa(\frac{1}{T})^T (1 - \beta\mu)^P (L_{F_x}^2 + \frac{L_{f_{xy}}^2 C_{F_y}^2}{\mu^2} + \frac{4M_{f_{xy}}^2 L_{F_y}^2}{\mu^2} + \frac{4M_{f_{xy}}^2 C_{F_y}^2 L_{f_{yy}}^2}{\mu^4}) \|y_0^* - y_0\|^2$ is the initial error. Specifically, if $Q_k = k + 1$, we have:*

$$\frac{1}{K} \sum_{k=0}^{K-1} \|\nabla\Phi(x_k)\|^2 \leq \frac{4(\Phi(x_0) - \Phi(x^*))}{\alpha K} + \frac{3\delta_0}{K(1 - \tau)} + \frac{18nLM_{f_{xy}}^2 C_{F_y}^2 \ln K}{\mu^3 \tilde{\xi} K}. \quad (72)$$

*Proof.* Substituting the inequality (67) into (40) yields:

$$
\begin{aligned}
\|\tilde{\nabla}\Phi(x_k) - \nabla\Phi(x_k)\|^2 &\leq 3\big(L_{F_x}^2 + \frac{L_{f_{xy}}^2 C_{F_y}^2}{\mu^2}\big) \|y_{k,T} - y_k^*\|^2 \\
&\quad + 3M_{f_{xy}}^2 \left( 2\frac{nLC_{F_y}^2}{\tilde{\xi}\mu^3 Q_k} + 4\big(\frac{L_{F_y}^2}{\mu^2} + \frac{C_{F_y}^2 L_{f_{yy}}^2}{\mu^4}\big) \|y_k^* - y_{k,T}\|^2 \right) \\
&\leq \left( 3L_{F_x}^2 + \frac{3L_{f_{xy}}^2 C_{F_y}^2}{\mu^2} + 12M_{f_{xy}}^2 \big(\frac{L_{F_y}^2}{\mu^2} + \frac{C_{F_y}^2 L_{f_{yy}}^2}{\mu^4}\big) \right) \|y_k^* - y_{k,T}\|^2 \\
&\quad + 6\frac{nLM_{f_{xy}}^2 C_{F_y}^2}{\mu^3 \tilde{\xi} Q_k}.
\end{aligned}
$$
(73)

Then, based on Lemma D.15, substituting the above inequality into (43) yields:

$$
\begin{aligned}
\|y_k^* - y_{k,T}\|^2 &\leq (1 + \varepsilon)\kappa(\frac{1}{T})^T (1 - \beta\mu)^P \|y_{k-1,T} - y_{k-1}^*\|^2 + 2(1 + \frac{1}{\varepsilon})\kappa(\frac{1}{T})^T (1 - \beta\mu)^P L_y^2 \alpha^2 \|\nabla\Phi(x_{k-1})\|^2 \\
&\quad + 2(1 + \frac{1}{\varepsilon})\kappa(\frac{1}{T})^T (1 - \beta\mu)^P L_y^2 \alpha^2 \|\tilde{\nabla}\Phi(x_{k-1}) - \nabla\Phi(x_{k-1})\|^2 \\
&\leq \tau \|y_{k-1}^* - y_{k-1,T}\|^2 + 2(1 + \frac{1}{\varepsilon})\kappa(\frac{1}{T})^T (1 - \beta\mu)^P L_y^2 \alpha^2 \|\nabla\Phi(x_{k-1})\|^2 \\
&\quad + 12(1 + \frac{1}{\varepsilon})\kappa(\frac{1}{T})^T (1 - \beta\mu)^P L_y^2 \alpha^2 \frac{nLM_{f_{xy}}^2 C_{F_y}^2}{\mu^3 \tilde{\xi} Q_{k-1}},
\end{aligned}
$$
(74)

where $\tau = \kappa(\frac{1}{T})^T (1 - \beta\mu)^P \big( (1 + \varepsilon) + 6(1 + \frac{1}{\varepsilon}) L_y^2 \alpha^2 (L_{F_x}^2 + \frac{L_{f_{xy}}^2 C_{F_y}^2}{\mu^2} + \frac{4M_{f_{xy}}^2 L_{F_y}^2}{\mu^2} + \frac{4M_{f_{xy}}^2 C_{F_y}^2 L_{f_{yy}}^2}{\mu^4}) \big)$.

Summing the inequality (74) from 0 to $k$ results in:

$$
\begin{aligned}
\|y_k^* - y_{k,T}\|^2 &\leq \tau^k \|y_0^* - y_{0,T}\|^2 + 2(1 + \frac{1}{\varepsilon})\kappa(\frac{1}{T})^T (1 - \beta\mu)^P L_y^2 \alpha^2 \sum_{j=0}^{k-1} \tau^j \|\nabla\Phi(x_{k-1-j})\|^2 \\
&\quad + 12(1 + \frac{1}{\varepsilon})\kappa(\frac{1}{T})^T (1 - \beta\mu)^P L_y^2 \alpha^2 \frac{nLM_{f_{xy}}^2 C_{F_y}^2}{\mu^3 \tilde{\xi}} \sum_{j=0}^{k-1} \tau^j \frac{1}{Q_{k-1-j}}.
\end{aligned}
$$

Combining the inequality (73) and $\|y_0^* - y_{0,T}\|^2 \le \kappa(\frac{1}{T})^T(1-\beta\mu)^P\|y_0^* - y_0\|^2$, it follows that

$$\|\tilde\nabla\Phi(x_k) - \nabla\Phi(x_k)\|^2 \le \delta_0\tau^k + \omega\alpha^2\sum_{j=0}^{k-1}\tau^j\|\nabla\Phi(x_{k-1-j})\|^2$$
$$+ 6\omega\alpha^2\frac{nLM_{f_{xy}}^2 C_{F_y}^2}{\mu^3\tilde\xi}\sum_{j=0}^{k-1}\tau^j\frac{1}{Q_{k-1-j}} + 6\frac{nLM_{f_{xy}}^2 C_{F_y}^2}{\mu^3\tilde\xi Q_k}, \tag{75}$$

where $\delta_0 = 3\kappa(\frac{1}{T})^T(1-\beta\mu)^P(L_{F_x}^2 + \frac{L_{f_{xy}}^2 C_{F_y}^2}{\mu^2} + \frac{4M_{f_{xy}}^2 L_{F_y}^2}{\mu^2} + \frac{4M_{f_{xy}}^2 C_{F_y}^2 L_{f_{yy}}^2}{\mu^4})\|y_0^* - y_0\|^2$ and $\omega = 6(L_{F_x}^2 + \frac{L_{f_{xy}}^2 C_{F_y}^2}{\mu^2} + \frac{4M_{f_{xy}}^2 L_{F_y}^2}{\mu^2} + \frac{4M_{f_{xy}}^2 C_{F_y}^2 L_{f_{yy}}^2}{\mu^4})(1+\frac{1}{\varepsilon})\kappa(\frac{1}{T})^T(1-\beta\mu)^P L_y^2$.

Since $\nabla\Phi(\cdot)$ is $L_\Phi$−Lipschitz, it can be obtained that

$$\Phi(x_{k+1}) \le \Phi(x_k) + \langle\nabla\Phi(x_k), x_{k+1} - x_k\rangle + \frac{L_\Phi}{2}\|x_{k+1} - x_k\|^2$$
$$\le \Phi(x_k) - \alpha\langle\nabla\Phi(x_k), \tilde\nabla\Phi(x_k) - \nabla\Phi(x_k)\rangle - \alpha\|\nabla\Phi(x_k)\|^2 + \alpha^2 L_\Phi\|\nabla\Phi(x_k)\|^2$$
$$+ \alpha^2 L_\Phi\|\nabla\Phi(x_k) - \tilde\nabla\Phi(x_k)\|^2 \tag{76}$$
$$\le \Phi(x_k) - \left(\frac{\alpha}{2} - \alpha^2 L_\Phi\right)\|\nabla\Phi(x_k)\|^2 + \left(\frac{\alpha}{2} + \alpha^2 L_\Phi\right)\|\nabla\Phi(x_k) - \tilde\nabla\Phi(x_k)\|^2.$$

Using the inequality (75) yields:

$$\Phi(x_{k+1}) \le \Phi(x_k) - \left(\frac{\alpha}{2} - \alpha^2 L_\Phi\right)\|\nabla\Phi(x_k)\|^2 + \left(\frac{\alpha}{2} + \alpha^2 L_\Phi\right)\|\nabla\Phi(x_k) - \tilde\nabla\Phi(x_k)\|^2$$
$$\le \Phi(x_k) - \left(\frac{\alpha}{2} - \alpha^2 L_\Phi\right)\|\nabla\Phi(x_k)\|^2 + \left(\frac{\alpha}{2} + \alpha^2 L_\Phi\right)\delta_0\tau^k$$
$$+ \omega\alpha^2\left(\frac{\alpha}{2} + \alpha^2 L_\Phi\right)\sum_{j=0}^{k-1}\tau^j\|\nabla\Phi(x_{k-1-j})\|^2 + \left(\frac{\alpha}{2} + \alpha^2 L_\Phi\right)\frac{6nLM_{f_{xy}}^2 C_{F_y}^2}{\mu^3\tilde\xi Q_k} \tag{77}$$
$$+ 6\omega\left(\frac{\alpha}{2} + \alpha^2 L_\Phi\right)\alpha^2\frac{nLM_{f_{xy}}^2 C_{F_y}^2}{\mu^3\tilde\xi}\sum_{j=0}^{k-1}\tau^j\frac{1}{Q_{k-1-j}}.$$

Finally, by telescoping the inequality (77) from $k = 0$ to $k = K - 1$, it is derived that

$$\left(\frac{\alpha}{2} - \alpha^2 L_\Phi\right)\sum_{k=0}^{K-1}\|\nabla\Phi(x_k)\|^2 \le \Phi(x_0) - \Phi(x_K) + \left(\frac{\alpha}{2} + \alpha^2 L_\Phi\right)\frac{\delta_0}{1-\tau}$$
$$+ \omega\alpha^2\left(\frac{\alpha}{2} + \alpha^2 L_\Phi\right)\sum_{k=0}^{K-1}\sum_{j=0}^{k-1}\tau^j\|\nabla\Phi(x_{k-1-j})\|^2$$
$$+ \sum_{k=0}^{K-1}\left(\frac{\alpha}{2} + \alpha^2 L_\Phi\right)\frac{6nLM_{f_{xy}}^2 C_{F_y}^2}{\mu^3\tilde\xi Q_k} \tag{78}$$
$$+ 6\omega\left(\frac{\alpha}{2} + \alpha^2 L_\Phi\right)\alpha^2\frac{nLM_{f_{xy}}^2 C_{F_y}^2}{\mu^3\tilde\xi}\sum_{k=0}^{K-1}\sum_{j=0}^{k-1}\tau^j\frac{1}{Q_{k-1-j}}.$$

Moreover, due to $\sum_{k=0}^{K-1}\sum_{j=0}^{k-1}a_j b_{k-1-j} \le \sum_{k=0}^{K-1}a_k\sum_{j=0}^{K-1}b_j$, we can deduce that

$$\sum_{k=0}^{K-1}\sum_{j=0}^{k-1}\tau^j\|\nabla\Phi(x_{k-1-j})\|^2 \le \sum_{k=0}^{K-1}\tau^k\sum_{k=0}^{K-1}\|\nabla\Phi(x_k)\|^2 \le \frac{1}{1-\tau}\sum_{k=0}^{K-1}\|\nabla\Phi(x_k)\|^2,$$
$$\sum_{k=0}^{K-1}\sum_{j=0}^{k-1}\tau^j\frac{1}{Q_{k-1-j}} \le \sum_{k=0}^{K-1}\tau^k\sum_{k=0}^{K-1}\frac{1}{Q_k} \le \frac{1}{1-\tau}\sum_{k=0}^{K-1}\frac{1}{Q_k}.$$

Then, the following inequality holds:

$$\left(\frac{1}{2} - \alpha L_\Phi - \omega\alpha^2\left(\frac{1}{2} + \alpha L_\Phi\right)\frac{1}{1-\tau}\right)\frac{1}{K}\sum_{k=0}^{K-1}\|\nabla\Phi(x_k)\|^2$$

$$\leq \frac{\Phi(x_0) - \Phi(x_K)}{\alpha K} + \left(\frac{1}{2} + \alpha L_\Phi\right)\frac{\delta_0}{K(1-\tau)} \tag{79}$$

$$+ \frac{1}{K}\sum_{k=0}^{K-1}\left(\frac{1}{2} + \alpha L_\Phi\right)\frac{6nLM_{f_{xy}}^2 C_{F_y}^2}{\mu^3\tilde{\xi}Q_k} + \frac{1}{K}\sum_{k=0}^{K-1}\frac{6\omega}{1-\tau}\left(\frac{1}{2} + \alpha L_\Phi\right)\alpha^2\frac{nLM_{f_{xy}}^2 C_{F_y}^2}{\mu^3\tilde{\xi}Q_k}.$$

If $\alpha L_\Phi + \omega\alpha^2\left(\frac{1}{2} + \alpha L_\Phi\right)\frac{1}{1-\tau} \leq \frac{1}{4}$, then

$$\frac{1}{K}\sum_{k=0}^{K-1}\|\nabla\Phi(x_k)\|^2 \leq \frac{4(\Phi(x_0) - \inf_x \Phi(x))}{\alpha K} + \frac{3\delta_0}{K(1-\tau)} + \frac{1}{K}\sum_{k=0}^{K-1}\frac{18nLM_{f_{xy}}^2 C_{F_y}^2}{\mu^3\tilde{\xi}Q_k}. \tag{80}$$

Finally, by substituting $Q_k = k + 1$ into (80), (72) is derived. $\qquad\square$

In addition, we present another theorem that does not require the strict assumption on the initial Hessian matrix $H_0$, but does require that $T \geq 8n\ln\left(\frac{2L}{\mu}\right)$.

**Theorem D.25.** *Suppose that Assumptions 3.1, 3.2 and 3.3 hold. Choose the stepsize $\beta$ and warm-up iteration steps $P$ such that $(1-\beta\mu)^P\|y_k - y_k^*\| \leq K_1$, where $K_1$ is defined in (16). Set $H_0 = LI$ and $T \geq 8n\ln\frac{2L}{\mu}$. Define $\tau = c_l^2\kappa^3(\frac{1}{T})^T(1-\beta\mu)^P\left((1+\varepsilon) + 6(1+\frac{1}{\varepsilon})L_y^2\alpha^2(L_{F_x}^2 + \frac{L_{f_{xy}}^2 C_{F_y}^2}{\mu^2} + \frac{4M_{f_{xy}}^2 L_{F_y}^2}{\mu^2} + \frac{4M_{f_{xy}}^2 C_{F_y}^2 L_{f_{yy}}^2}{\mu^4})\right)$ and $\omega = 6(L_{F_x}^2 + \frac{L_{f_{xy}}^2 C_{F_y}^2}{\mu^2} + \frac{4M_{f_{xy}}^2 L_{F_y}^2}{\mu^2} + \frac{4M_{f_{xy}}^2 C_{F_y}^2 L_{f_{yy}}^2}{\mu^4})(1 + \frac{1}{\varepsilon})c_l^2\kappa^3(\frac{1}{T})^T(1-\beta\mu)^P L_y^2$. Choose the stepsize $\alpha > 0$, the positive constant $\varepsilon > 0$ and iterate $T > 0$ such that*

$$\tau < 1 \quad\text{and}\quad \alpha L_\Phi + \omega\alpha^2\left(\frac{1}{2} + \alpha L_\Phi\right)\frac{1}{1-\tau} \leq \frac{1}{4}.$$

*Then, the solution $x_k$ generated by Algorithm 1 achieves the following convergence rate:*

$$\frac{1}{K}\sum_{k=0}^{K-1}\|\nabla\Phi(x_k)\|^2 \leq \frac{4(\Phi(x_0) - \Phi(x^*))}{\alpha K} + \frac{3\delta_0}{K(1-\tau)} + \frac{1}{K}\sum_{k=0}^{K-1}\frac{18nLM_{f_{xy}}^2 C_{F_y}^2}{\mu^3\tilde{\xi}Q_k}, \tag{81}$$

*where $\delta_0 = 3c_l^2\kappa^3\kappa(\frac{1}{T})^T(1-\beta\mu)^P(L_{F_x}^2 + \frac{L_{f_{xy}}^2 C_{F_y}^2}{\mu^2} + \frac{4M_{f_{xy}}^2 L_{F_y}^2}{\mu^2} + \frac{4M_{f_{xy}}^2 C_{F_y}^2 L_{f_{yy}}^2}{\mu^4})\|y_0^* - y_0\|^2$ is the initial error. Specifically, if $Q_k = k + 1$, we have:*

$$\frac{1}{K}\sum_{k=0}^{K-1}\|\nabla\Phi(x_k)\|^2 \leq \frac{4(\Phi(x_0) - \inf_x \Phi(x))}{\alpha K} + \frac{3\delta_0}{K(1-\tau)} + \frac{18nLM_{f_{xy}}^2 C_{F_y}^2 \ln K}{\mu^3\tilde{\xi}K}. \tag{82}$$

*Proof.* Substituting the inequality (67) into (40) yields:

$$\|\tilde{\nabla}\Phi(x_k) - \nabla\Phi(x_k)\|^2 \leq 3\left(L_{F_x}^2 + \frac{L_{f_{xy}}^2 C_{F_y}^2}{\mu^2}\right)\|y_{k,T} - y_k^*\|^2$$

$$+ 3M_{f_{xy}}^2\left(2\frac{nLC_{F_y}^2}{\tilde{\xi}\mu^3 Q_k} + 4\left(\frac{L_{F_y}^2}{\mu^2} + \frac{C_{F_y}^2 L_{f_{yy}}^2}{\mu^4}\right)\|y_k^* - y_{k,T}\|^2\right)$$

$$\leq \left(3L_{F_x}^2 + \frac{3L_{f_{xy}}^2 C_{F_y}^2}{\mu^2} + 12M_{f_{xy}}^2\left(\frac{L_{F_y}^2}{\mu^2} + \frac{C_{F_y}^2 L_{f_{yy}}^2}{\mu^4}\right)\right)\|y_k^* - y_{k,T}\|^2$$

$$+ 6\frac{nLM_{f_{xy}}^2 C_{F_y}^2}{\mu^3\tilde{\xi}Q_k}. \tag{83}$$

Then, based on Lemma D.17, substituting the above inequality into (45) yields:

$$
\begin{aligned}
\|y_k^* - y_{k,T}\|^2 &\leq (1+\varepsilon)c_l^2\kappa^3(\frac{1}{T})^T(1-\beta\mu)^P\|y_{k-1,T} - y_{k-1}^*\|^2 + 2(1+\frac{1}{\varepsilon})c_l^2\kappa^3(\frac{1}{T})^T(1-\beta\mu)^P L_y^2\alpha^2\|\nabla\Phi(x_{k-1})\|^2 \\
&\quad + 2(1+\frac{1}{\varepsilon})c_l^2\kappa^3(\frac{1}{T})^T(1-\beta\mu)^P L_y^2\alpha^2\|\tilde\nabla\Phi(x_{k-1}) - \nabla\Phi(x_{k-1})\|^2 \\
&\leq \tau\|y_{k-1}^* - y_{k-1,T}\|^2 + 2(1+\frac{1}{\varepsilon})c_l^2\kappa^3(\frac{1}{T})^T(1-\beta\mu)^P L_y^2\alpha^2\|\nabla\Phi(x_{k-1})\|^2 \\
&\quad + 12(1+\frac{1}{\varepsilon})c_l^2\kappa^3(\frac{1}{T})^T(1-\beta\mu)^P L_y^2\alpha^2\frac{nLM_{f_{xy}}^2 C_{F_y}^2}{\mu^3\tilde\xi Q_{k-1}},
\end{aligned}
$$
(84)

where $\tau = c_l^2\kappa^3(\frac{1}{T})^T(1-\beta\mu)^P\big((1+\varepsilon) + 6(1+\frac{1}{\varepsilon})L_y^2\alpha^2(L_{F_x}^2 + \frac{L_{f_{xy}}^2 C_{F_y}^2}{\mu^2} + \frac{4M_{f_{xy}}^2 L_{F_y}^2}{\mu^2} + \frac{4M_{f_{xy}}^2 C_{F_y}^2 L_{f_{yy}}^2}{\mu^4})\big)$.

Summing the inequality (84) from 0 to $k$ results in:

$$
\begin{aligned}
\|y_k^* - y_{k,T}\|^2 &\leq \tau^k\|y_0^* - y_{0,T}\|^2 + 2(1+\frac{1}{\varepsilon})c_l^2\kappa^3(\frac{1}{T})^T(1-\beta\mu)^P L_y^2\alpha^2 \sum_{j=0}^{k-1}\tau^j\|\nabla\Phi(x_{k-1-j})\|^2 \\
&\quad + 12(1+\frac{1}{\varepsilon})c_l^2\kappa^3(\frac{1}{T})^T(1-\beta\mu)^P L_y^2\alpha^2\frac{nLM_{f_{xy}}^2 C_{F_y}^2}{\mu^3\tilde\xi}\sum_{j=0}^{k-1}\tau^j\frac{1}{Q_{k-1-j}}.
\end{aligned}
$$

Combining the inequality (83) and $\|y_0^* - y_{0,T}\|^2 \leq c_l^2\kappa^3(\frac{1}{T})^T(1-\beta\mu)^P\|y_0^* - y_0\|^2$, it follows that

$$
\begin{aligned}
\|\tilde\nabla\Phi(x_k) - \nabla\Phi(x_k)\|^2 &\leq \delta_0\tau^k + \omega\alpha^2\sum_{j=0}^{k-1}\tau^j\|\nabla\Phi(x_{k-1-j})\|^2 \\
&\quad + 6\omega\alpha^2\frac{nLM_{f_{xy}}^2 C_{F_y}^2}{\mu^3\tilde\xi}\sum_{j=0}^{k-1}\tau^j\frac{1}{Q_{k-1-j}} + 6\frac{nLM_{f_{xy}}^2 C_{F_y}^2}{\mu^3\tilde\xi Q_k},
\end{aligned}
$$
(85)

where $\delta_0 = 3c_l^2\kappa^3(\frac{1}{T})^T(1-\beta\mu)^P(L_{F_x}^2 + \frac{L_{f_{xy}}^2 C_{F_y}^2}{\mu^2} + \frac{4M_{f_{xy}}^2 L_{F_y}^2}{\mu^2} + \frac{4M_{f_{xy}}^2 C_{F_y}^2 L_{f_{yy}}^2}{\mu^4})\|y_0^* - y_0\|^2$ and $\omega = 6(L_{F_x}^2 + \frac{L_{f_{xy}}^2 C_{F_y}^2}{\mu^2} + \frac{4M_{f_{xy}}^2 L_{F_y}^2}{\mu^2} + \frac{4M_{f_{xy}}^2 C_{F_y}^2 L_{f_{yy}}^2}{\mu^4})(1+\frac{1}{\varepsilon})c_l^2\kappa^3(\frac{1}{T})^T(1-\beta\mu)^P L_y^2$.

Since $\nabla\Phi(\cdot)$ is $L_\Phi-$Lipschitz, it can be obtained that

$$
\begin{aligned}
\Phi(x_{k+1}) &\leq \Phi(x_k) + \langle\nabla\Phi(x_k), x_{k+1} - x_k\rangle + \frac{L_\Phi}{2}\|x_{k+1} - x_k\|^2 \\
&\leq \Phi(x_k) - \alpha\langle\nabla\Phi(x_k), \tilde\nabla\Phi(x_k) - \nabla\Phi(x_k)\rangle - \alpha\|\nabla\Phi(x_k)\|^2 + \alpha^2 L_\Phi\|\nabla\Phi(x_k)\|^2 \\
&\quad + \alpha^2 L_\Phi\|\nabla\Phi(x_k) - \tilde\nabla\Phi(x_k)\|^2 \\
&\leq \Phi(x_k) - \left(\frac{\alpha}{2} - \alpha^2 L_\Phi\right)\|\nabla\Phi(x_k)\|^2 + \left(\frac{\alpha}{2} + \alpha^2 L_\Phi\right)\|\nabla\Phi(x_k) - \tilde\nabla\Phi(x_k)\|^2.
\end{aligned}
$$
(86)

Using the inequality (85) yields:

$$
\begin{aligned}
\Phi(x_{k+1}) &\leq \Phi(x_k) - \left(\frac{\alpha}{2} - \alpha^2 L_\Phi\right)\|\nabla\Phi(x_k)\|^2 + \left(\frac{\alpha}{2} + \alpha^2 L_\Phi\right)\|\nabla\Phi(x_k) - \tilde\nabla\Phi(x_k)\|^2 \\
&\leq \Phi(x_k) - \left(\frac{\alpha}{2} - \alpha^2 L_\Phi\right)\|\nabla\Phi(x_k)\|^2 + \left(\frac{\alpha}{2} + \alpha^2 L_\Phi\right)\delta_0\tau^k \\
&\quad + \omega\alpha^2\left(\frac{\alpha}{2} + \alpha^2 L_\Phi\right)\sum_{j=0}^{k-1}\tau^j\|\nabla\Phi(x_{k-1-j})\|^2 + \left(\frac{\alpha}{2} + \alpha^2 L_\Phi\right)\frac{6nLM_{f_{xy}}^2 C_{F_y}^2}{\mu^3\tilde\xi Q_k} \\
&\quad + 6\omega\left(\frac{\alpha}{2} + \alpha^2 L_\Phi\right)\alpha^2\frac{nLM_{f_{xy}}^2 C_{F_y}^2}{\mu^3\tilde\xi}\sum_{j=0}^{k-1}\tau^j\frac{1}{Q_{k-1-j}}.
\end{aligned}
$$
(87)

Finally, by telescoping the inequality (87) from $k = 0$ to $k = K - 1$, it is derived that

$$
\begin{aligned}
\left(\frac{\alpha}{2} - \alpha^2 L_\Phi\right) \sum_{k=0}^{K-1} \|\nabla\Phi(x_k)\|^2 \leq &\, \Phi(x_0) - \Phi(x_K) + \left(\frac{\alpha}{2} + \alpha^2 L_\Phi\right) \frac{\delta_0}{1 - \tau} \\
&+ \omega\alpha^2 \left(\frac{\alpha}{2} + \alpha^2 L_\Phi\right) \sum_{k=0}^{K-1}\sum_{j=0}^{k-1} \tau^j \|\nabla\Phi(x_{k-1-j})\|^2 \\
&+ \sum_{k=0}^{K-1} \left(\frac{\alpha}{2} + \alpha^2 L_\Phi\right) \frac{6nLM_{f_{xy}}^2 C_{F_y}^2}{\mu^3\tilde{\xi}Q_k} \\
&+ 6\omega\left(\frac{\alpha}{2} + \alpha^2 L_\Phi\right)\alpha^2 \frac{nLM_{f_{xy}}^2 C_{F_y}^2}{\mu^3\tilde{\xi}} \sum_{k=0}^{K-1}\sum_{j=0}^{k-1} \tau^j \frac{1}{Q_{k-1-j}}.
\end{aligned}
\tag{88}
$$

Moreover, due to $\sum_{k=0}^{K-1}\sum_{j=0}^{k-1} a_j b_{k-1-j} \leq \sum_{k=0}^{K-1} a_k \sum_{j=0}^{K-1} b_j$, we can deduce that

$$
\sum_{k=0}^{K-1}\sum_{j=0}^{k-1} \tau^j \|\nabla\Phi(x_{k-1-j})\|^2 \leq \sum_{k=0}^{K-1} \tau^k \sum_{k=0}^{K-1} \|\nabla\Phi(x_k)\|^2 \leq \frac{1}{1-\tau} \sum_{k=0}^{K-1} \|\nabla\Phi(x_k)\|^2,
$$

$$
\sum_{k=0}^{K-1}\sum_{j=0}^{k-1} \tau^j \frac{1}{Q_{k-1-j}} \leq \sum_{k=0}^{K-1} \tau^k \sum_{k=0}^{K-1} \frac{1}{Q_k} \leq \frac{1}{1-\tau} \sum_{k=0}^{K-1} \frac{1}{Q_k}.
$$

Then, the following inequality holds:

$$
\begin{aligned}
&\left(\frac{1}{2} - \alpha L_\Phi - \omega\alpha^2\left(\frac{1}{2} + \alpha L_\Phi\right)\frac{1}{1-\tau}\right) \frac{1}{K}\sum_{k=0}^{K-1} \|\nabla\Phi(x_k)\|^2 \\
\leq &\, \frac{\Phi(x_0) - \Phi(x_K)}{\alpha K} + \left(\frac{1}{2} + \alpha L_\Phi\right)\frac{\delta_0}{K(1-\tau)} \\
&+ \frac{1}{K}\sum_{k=0}^{K-1}\left(\frac{1}{2} + \alpha L_\Phi\right)\frac{6nLM_{f_{xy}}^2 C_{F_y}^2}{\mu^3\tilde{\xi}Q_k} + \frac{1}{K}\sum_{k=0}^{K-1}\frac{6\omega}{1-\tau}\left(\frac{1}{2} + \alpha L_\Phi\right)\alpha^2\frac{nLM_{f_{xy}}^2 C_{F_y}^2}{\mu^3\tilde{\xi}Q_k}.
\end{aligned}
\tag{89}
$$

If $\alpha L_\Phi + \omega\alpha^2\left(\frac{1}{2} + \alpha L_\Phi\right)\frac{1}{1-\tau} \leq \frac{1}{4}$, then

$$
\frac{1}{K}\sum_{k=0}^{K-1} \|\nabla\Phi(x_k)\|^2 \leq \frac{4(\Phi(x_0) - \Phi(x^*))}{\alpha K} + \frac{3\delta_0}{K(1-\tau)} + \frac{1}{K}\sum_{k=0}^{K-1}\frac{18nLM_{f_{xy}}^2 C_{F_y}^2}{\mu^3\tilde{\xi}Q_k},
\tag{90}
$$

where $\Phi(x^*) = \inf_x \Phi(x)$.

Finally, by substituting $Q_k = k + 1$ into (90), (82) is derived. $\qquad\square$

## E    COMPLEXITY AND THEORETICAL DISCUSSION

**Corollary E.1.** *Consider $T = \Theta(\ln\kappa)$ and $\alpha = \Theta(\kappa^{-3})$ such that $\tau < 1$ and $\alpha L_\Phi + \omega\alpha^2\left(\frac{1}{2} + \alpha L_\Phi\right)\frac{1}{1-\tau} \leq \frac{1}{4}$. Under the same setting of Theorem 3.7, we have $\frac{1}{K}\sum_{k=0}^{K-1}\|\nabla\Phi(x_k)\|^2 = \mathcal{O}(\frac{\kappa^3}{K} + \frac{\kappa^3\ln K}{K})$. To achieve an $\epsilon$-stationary point, we require $K = \tilde{\mathcal{O}}(\kappa^3\epsilon^{-1})$, resulting in the gradient complexity of $Gc(f, \epsilon) = \tilde{\mathcal{O}}(\kappa^6\epsilon^{-2}), Gc(F, \epsilon) = \tilde{\mathcal{O}}(\kappa^3\epsilon^{-1})$ and a Jacobian-vector product complexity $JV(\epsilon) = \tilde{\mathcal{O}}(\kappa^3\epsilon^{-1})$.*

*Proof.* For Theorem 3.7, by Theorem D.24, we have

$$
c_3 = 6L_y^2\left(L_{F_x}^2 + \frac{L_{f_{xy}}^2 C_{F_y}^2}{\mu^2} + \frac{4M_{f_{xy}}^2 L_{F_y}^2}{\mu^2} + \frac{4M_{f_{xy}}^2 C_{F_y}^2 L_{f_{yy}}^2}{\mu^4}\right) = \Theta(\kappa^6).
$$

Since $0 < (1 - \beta\mu)^P \leq 1$ and $\alpha = \Theta(\kappa^{-3})$, it is derived that

$$\tau = \kappa(\frac{1}{T})^T (1 - \beta\mu)^P \big((1 + \varepsilon) + (1 + \frac{1}{\varepsilon})\alpha^2 c_3\big) = \Theta(\kappa(1/T)^T),$$

$$\omega = c_3(1 + \frac{1}{\varepsilon})\kappa(\frac{1}{T})^T (1 - \beta\mu)^P = \Theta(\kappa^7 (1/T)^T).$$

Based on Lemma D.12, $\nabla\Phi$ is $L_\Phi$-Lipschitz with $L_\Phi = \Theta(\kappa^3)$. For a suitable choice of $\alpha = \Theta(\kappa^{-3})$, it follows that $\alpha L_\Phi < \frac{1}{8}$. Additionally, with $T = \Theta(\ln\kappa)$, the conditions $0 < \tau \leq \frac{1}{2}$ and $\omega\alpha^2 = \Theta(\kappa(1/T)^T) \leq \frac{1}{10}$ are satisfied. Consequently,

$$\alpha L_\Phi + \omega\alpha^2 \left(\frac{1}{2} + \alpha L_\Phi\right) \frac{1}{1 - \tau} \leq \frac{1}{8} + \frac{1}{10}\left(\frac{1}{2} + \frac{1}{8}\right)\frac{1}{1 - \frac{1}{2}} \leq \frac{1}{4}.$$

Since $\alpha = \Theta(\kappa^{-3})$, it can be obtained from (72) that $\frac{1}{K}\sum_{k=0}^{K-1}\|\nabla\Phi(x_k)\|^2 = \mathcal{O}(\frac{\kappa^3}{K} + \frac{\kappa^3 \ln K}{K})$. Furthermore, in order to achieve an $\epsilon$-stationary point, we have $K = \mathcal{O}(\kappa^3 \epsilon^{-1} \ln\frac{\kappa^3}{\epsilon}) = \tilde{\mathcal{O}}(\kappa^3 \epsilon^{-1})$. Therefore, the following complexity results are derived:

- $Gc(f, \epsilon) = K(T + P) + \sum_{k=0}^{K-1} Q_k = K(T + P) + \frac{K(K+1)}{2} = \tilde{\mathcal{O}}(\kappa^6 \epsilon^{-2})$;

- $Gc(F, \epsilon) = 2K = \tilde{\mathcal{O}}(\kappa^3 \epsilon^{-1})$;

- $JV(\epsilon) = K = \tilde{\mathcal{O}}(\kappa^3 \epsilon^{-1})$.

**Details for obtaining $\tau \leq 1/2$ and $\omega\alpha^2 \leq 1/10$:** Since $\tau = \Theta\left(\kappa(1/T)^T\right)$, $\omega\alpha^2 = \Theta\left(\kappa(1/T)^T\right)$ and $\kappa \geq 1$, it is enough to show that $C_0\kappa(1/T)^T \leq 1/10$ by choosing $T = \Theta(\ln\kappa)$. Here, $C_0 \geq 1$ is a positive constant in $\tau$ and $\omega\alpha^2$, depending explicitly on the Lipschitz constants in the assumptions.

By taking the logarithm on both sides of $C_0\kappa(1/T)^T \leq 1/10$, we get:

$$\ln\kappa - T\ln T \leq -\ln(10C_0).$$

This is equivalent to $T\ln T \geq \ln\kappa + \ln(10C_0)$. Therefore, choosing $T \geq \ln\kappa + \ln(10C_0) + \epsilon$ is sufficient, since $\ln T \geq 1$. Similarly, we can prove the result for Theorem 3.4.

$\square$

