# OpenReview forum: "qNBO: quasi-Newton Meets Bilevel Optimization"
_ICLR.cc/2025/Conference — ICLR 2025 Poster_

### Official Review · Reviewer_XUKY · 2024-10-30

**Soundness:** 4
**Presentation:** 4
**Contribution:** 4
**Rating:** 8
**Confidence:** 4

**Summary:**

The paper presents a bilevel optimization framework, QNBO, which utilizes quasi-Newton methods to address the computational challenges associated with hierarchical learning tasks. The authors integrate quasi-Newton recursion schemes for efficiently approximating inverse Hessian-vector products, specifically using BFGS and SR1 methods. The authors argue that QNBO offers superior convergence and computational efficiency, substantiated through theoretical proofs and numerical experiments.

**Strengths:**

1. The QNBO framework presents an innovative approach by combining bilevel optimization with quasi-Newton methods to improve hypergradient approximation accuracy.

2. The paper provides a non-asymptotic convergence analysis for QNBO, especially for BFGS, detailing convergence under both quadratic and general settings.

3. The authors validate QNBO on real-world tasks such as hyperparameter optimization, meta-learning, and data hyper-cleaning. QNBO’s performance is benchmarked against established algorithms, including SHINE and PZOBO, with performance gains observed in multiple scenarios.

4. I did not notice any major issues with the clarity of the presentation.

**Weaknesses:**

1. The SR1 method, despite its potential in quasi-Newton methods, is noted for its numerical instability in general functions without correction strategies.

2. QNBO offers a flexible choice between subroutine A and B based on performance and computational cost. However, the conditions for selecting one over the other are not rigorously outlined.

**Questions:**

1. Given the success of stochastic adaptations in other optimization frameworks, could the authors discuss potential challenges or advantages of incorporating stochastic quasi-Newton methods into QNBO? Specifically, how might this adaptation impact convergence rates or computational efficiency in practical implementations?

2. The initial matrix $H_0$ plays a significant role in quasi-Newton methods. Here, a scalar multiple of the identity matrix is used. Could the authors provide specific results or theoretical analysis on how different choices of $H_0$ might affect QNBO’s convergence rate and computational efficiency?

---

> ### Author Response · Authors · 2024-11-25
> **Response to concerns raised in Weaknesses.**
>
> >The SR1 method, despite its potential in quasi-Newton methods, is noted for its numerical instability in general functions without correction strategies.
>
> $\textbf{Response:}$
> Thank you for your concern. The main drawback of SR1 updating is that the denominator in its update formula
> $$ H_{t+1}=H_t+\frac{(s_t-H_t g_t)( s_t-H_t g_t)^T}{( s_t-H_t g_t)^T g_t}$$
> can vanish. In fact, even for convex quadratic functions, there may be steps where no symmetric rank-one updating formula satisfies the secant equation. Such situations can lead to numerical instabilities and even a breakdown of the SR1 algorithm (see Section 6.2 of [1]).
>
> To address these issues, a correction strategy can be applied to improve the numerical stability of SR1. The goal of such strategies is to keep the Hessian approximations under control. For detailed discussions on this topic, see Section 4.1 of Ye et al. (2023), which employs a correction strategy introduced in [2]. However, the correction strategy relies on a Hessian-based constant, which varies during iterations of $(x, y)$ in the bilevel setting. This introduces the need for an adaptive correction strategy in qNBO (SR1) to handle the dynamic nature of the Hessian in the bilevel setting. Developing such an adaptive strategy presents a distinct challenge in the bilevel optimization and could be an interesting future extension.
>
>
> [1]Nocedal, Jorge, and Stephen J. Wright, Numerical optimization. New York, NY: Springer New York, 1999.
>
> [2]Rodomanov, Anton, and Yurii Nesterov, Greedy quasi-Newton methods with explicit superlinear convergence. SIAM Journal on Optimization (2021): 785-811.
>
>
>
>
> >QNBO offers a flexible choice between subroutine A and B based on performance and computational cost. However, the conditions for selecting one over the other are not rigorously outlined.
>
>
> $\textbf{Response:}$
> Thanks for the concern. Unfortunately, at this point, we do not have a rigorous selection strategy for choosing between the two options: $Q_k=1$ or $Q_k>1$ in the iteration of $u_{k+1}$. Theoretically, if the ULI assumption described in (Ramzi et al., 2022) holds, then $Q_k=1$ would suffice, and we can share $\\{s_t, g_t\\}_{t=0}^{T-1}$ with subroutine $\mathcal{A}$. However, this assumption is quite strong (cf. Ramzi et al., 2022).
>
> To avoid potential incorrect inversion, we adopt $Q_k-1>0$ quasi-Newton steps in subroutine $\mathcal{B}$ for the direction $\nabla_y F(x_k, y_{k+1})$. Thanks to Reviewer 5z6s’s suggestion on employing a warm-start strategy for  $u_{k+1}$ , we have conducted additional experiments incorporating this strategy. See Appendix C.5.1 (Figure 6) of the revised paper for details. The empirical results show that qNBO performs better when applying the warm-start strategy for $u$. A rigorous analysis would require a more detailed characterization of the relationships between quasi-Newton iterations and their sensitivity to variations in $(x, y)$, which would be an interesting future extension.

---

> ### Author Response · Authors · 2024-11-25
> **Response to concerns raised in Questions (Part 1/2).**
>
> >Given the success of stochastic adaptations in other optimization frameworks, could the authors discuss potential challenges or advantages of incorporating stochastic quasi-Newton methods into QNBO? Specifically, how might this adaptation impact convergence rates or computational efficiency in practical implementations?
>
> $\textbf{Response:}$
> Thank you for your thoughtful questions. qNBO incorporates a generic decoupling structure, wherein all search directions are linear with respect to the objective functions. This structure builds upon existing methods such as HOAG (Pedregosa, 2016), AID-BIO (Ji et al., 2021), AMIGO (Arbel & Mairal, 2022), and SOBA/SABA (Dagréou et al., 2022), enabling qNBO to extend effectively from deterministic to stochastic settings.
>
>
> To elaborate, qNBO consists of three components, with the first two employing quasi-Newton recursion schemes. A straightforward stochastic adaptation involves replacing deterministic quasi-Newton recursion schemes with stochastic variants, such as K-BFGS [1] or Stochastic Block BFGS [2]. Additionally, in Part 3 of qNBO, deterministic gradients are replaced with stochastic gradients throughout the iterative process, consistent with the transition from deterministic to stochastic methods discussed by Dagréou et al. (2022).
>
> Stochastic quasi-Newton methods could be better equipped than stochastic gradient methods to handle ill-conditioning. Incorporating these methods into qNBO offers the advantage of mitigating the adverse effects of high non-linearity and ill-conditioning in the lower-level objective function by leveraging second-order information. Another potential benefit is the improvement of the constants involved in the sublinear convergence rates of stochastic methods. However, the full potential of the stochastic quasi-Newton schemes discussed (and possibly others) is not yet known.
>
> Furthermore, this straightforward extension may compromise convergence rates and computational efficiency under the same smoothness assumptions as in the deterministic setting. For an intuitive comparison (without using quasi-Newton methods), refer to the analysis of SOBA (Dagréou et al., 2022) and its improvement by MA-SOBA (Chen et al., 2023, Optimal Algorithms for Stochastic Bilevel Optimization under Relaxed Smoothness Conditions).
>
> In our opinion, the main challenges in maintaining comparable convergence rates and computational efficiency in a stochastic setting include:
>
> $\textbf{Constructing effective estimators:}$ How can we develop unbiased or biased estimators in Part 3 of qNBO by integrating techniques such as variance reduction or momentum (e.g., moving averages)? Potential candidates for these estimators include SGD, SAGA, SARAH, STORM, PAGE, and even Adam. However, analyzing this extension, particularly its convergence properties, requires significant effort. For recent progress (without using quasi-Newton methods), see SRBA (Dagréou et al., 2024) and SPABA (Chu et al., 2024).
>
> $\textbf{Analyzing convergence rates and complexity:}$ How can we evaluate the proposed stochastic algorithms in a bilevel setting while incorporating noisy second-order (curvature) information? Addressing these difficulties may require theoretical breakthroughs that go beyond existing first-order techniques, leaving this an unresolved challenge and an interesting future extension.
>
>
> [1] Goldfarb, D., Ren, Y., & Bahamou, A. (2020). Practical quasi-newton methods for training deep neural networks.
>
> [2] Gower, R., Goldfarb, D., & Richtárik, P. (2016). Stochastic block BFGS: Squeezing more curvature out of data.

---

> ### Author Response · Authors · 2024-11-25
> **Response to concerns raised in Questions (Part 2/2).**
>
> >The initial matrix $H_0$ plays a significant role in quasi-Newton methods. Here, a scalar multiple of the identity matrix is used. Could the authors provide specific results or theoretical analysis on how different choices of $H_0$ might affect QNBO’s convergence rate and computational efficiency?
>
>
> $\textbf{Response:}$
> Thank you for your thoughtful questions. As discussed in Chapter 6 of [1], the initial matrix $H_0$ is often set as a scalar multiple of the identity matrix, but there is no good general strategy for selecting this scalar. We have conducted experiments to evaluate the impact of the scalar multiple of $H_0$ on the performance of qNBO algorithms in toy and data hyper-cleaning experiments. The results indicate the following: (1) different choices of the scalar multiple for $H_0$ do not affect the convergence rate or computational efficiency of qNBO (SR1) in the toy experiment; and (2) for qNBO (BFGS), the scalar multiple of $H_0$ should not be too small in the data hyper-cleaning experiment. These findings are presented in Figures 1 and 2 of a single-page PDF (containing only figures and available at the anonymous link: https://drive.google.com/file/d/1rKXLVTyE-_iSna_8xBpSknMCSYmqMmlQ/view?usp=sharing).
>
> In theory, for the convergence result in the quadratic case, the condition $H_0 = LI$ should be explicitly stated in the theorem’s description (see Theorem 3.4 in the revised paper). For the general case, we have added a new theorem for qNBO (BFGS) in the Appendix (Theorem D.25 of the revised paper) to justify the selection of $H_0 = LI$. The proof of this theorem builds on the local convergence result of Rodomanov and Nesterov (2021b) (see Lemma D.3 of the revised paper).
>
> [1]Nocedal, Jorge, and Stephen J. Wright, Numerical optimization. New York, NY: Springer New York, 1999.

---

### Official Review · Reviewer_ozni · 2024-10-30

**Soundness:** 3
**Presentation:** 3
**Contribution:** 3
**Rating:** 8
**Confidence:** 4

**Summary:**

This work studies the bilevel optimization problem, where the lower-level problem is strongly convex and the upper-level objective function is nonconvex. The authors introduce a novel framework called qNBO, which approximates the hypergradient by employing quasi-Newton methods for both evaluating the lower-level solution and estimating the inverse Hessian-vector product. When the quasi-Newton method used is BFGS, they establish a non-asymptotic convergence for the proposed algorithm. Finally, empirical experiments demonstrate the superior performance of qNBO compared to other bilevel optimization algorithms.

**Strengths:**

1.The new bilevel algorithms are well motivated. By employing quasi-Newton recursion schemes, they incorporate two subroutines to accelerate hypergradient approximation and avoid incorrect inversion in solving the bilevel optimization problem.

2.Several machine learning tasks, including hyperparameter optimization in logistic regression, data hyper-cleaning, and few-shot meta-learning, are evaluated, demonstrating the superior performance of the proposed methods. Additionally, various ablation studies are provided.

3.A convergence rate guarantee is established when the quasi-Newton method used is BFGS.

**Weaknesses:**

My main concern is that:

Although quasi-Newton methods are used to estimate the inverse Hessian-vector product, the proposed algorithms still require computing Jacobian-vector products, which can be computationally expensive in large-scale cases. Could this computation pose a potential problem?

**Questions:**

For both the theory and experiments, here are a few things that need to be addressed.

1.In Remark 3.4, the authors stated that the qNBO (SR1) algorithm lacks convergence guarantees without specific corrections used to achieve numerical stability in the general case. It was also mentioned that the performance of qNBO (SR1) on the 20News dataset was omitted due to its ineffectiveness in the experiment in Section 4.2. What are the underlying reasons for these issues? A bit more discussion on these points would be helpful.

2.From the first terms on the right hand sides of both (6) and (7), it seems that Theorems 3.3 and 3.6 require a lower bound assumption for $\Phi(x)$. Is this correct? Additionally, in the proof sketch in Appendix D.2, $\tilde{\nabla}\Phi$ is mentioned but not defined. It would be helpful to specify its definition.

3.In the experiments, how to select the step sizes and the number of inner iterations for the proposed algorithms?

4.In Table 1, why does qNBO (BFGS) save so much time compared to SHINE for data hyper-cleaning?

---

> ### Author Response · Authors · 2024-11-25
> **Response to concerns raised in Weaknesses.**
>
> >Although quasi-Newton methods are used to estimate the inverse Hessian-vector product, the proposed algorithms still require computing Jacobian-vector products, which can be computationally expensive in large-scale cases. Could this computation pose a potential problem?
>
> $\textbf{Response:}$
> Thank you for the interesting question. Apart from the computation of Jacobian-vector products, all other operations in qNBO involve inner products and vector summations. Consequently, when Jacobian-vector products are not computationally expensive, qNBO can significantly reduce computational costs.
>
> Modern automatic differentiation frameworks, such as PyTorch and JAX, make the computational complexity of Jacobian-vector products comparable to that of gradient computation. Alternatively, recent methods in the bilevel optimization literature offer different strategies for estimating Jacobian-vector products. For instance, PZOBO (Sow et al., 2022b) computes these products using differences between two optimization-based trajectories, while FdeHBO (Yang et al., 2023) and HJFBiO (Huang, 2024) employ finite-difference approximations. Additionally, ZDSBA (Aghasi & Ghadimi, 2024) uses Gaussian smoothing as an effective alternative.

---

> ### Author Response · Authors · 2024-11-25
> **Response to concerns raised in Questions.**
>
> >In Remark 3.4, the authors stated that the qNBO (SR1) algorithm lacks convergence guarantees without specific corrections used to achieve numerical stability in the general case. It was also mentioned that the performance of qNBO (SR1) on the 20News dataset was omitted due to its ineffectiveness in the experiment in Section 4.2. What are the underlying reasons for these issues? A bit more discussion on these points would be helpful.
>
> $\textbf{Response:}$
> Thank you for your concern. The SR1 update formula is given by:
> $$ H_{t+1}=H_t+\frac{(s_t-H_t g_t)( s_t-H_t g_t)^T}{( s_t-H_t g_t)^T g_t}.$$
> However, even for convex quadratic functions, there are instances where no symmetric rank-one updating formula satisfies the secant equation. This issue arises when $ s_t-H_t g_t \neq 0$ but $( s_t-H_t g_t)^T g_t =0$ (see Section 6.2 of [1]). Such scenarios can lead to numerical instabilities and even a breakdown of the SR1 algorithm. To improve the numerical stability of the SR1 method, a correction strategy should be applied. For discussions on  this topic, refer to Section 4.1 of Ye et al. (2023).
>
>
> [1]Nocedal, Jorge, and Stephen J. Wright, Numerical optimization[M]. New York, NY: Springer New York, 1999.
>
>
> >From the first terms on the right hand sides of both (6) and (7), it seems that Theorems 3.3 and 3.6 require a lower bound assumption for $\Phi(x)$. Is this correct? Additionally, in the proof sketch in Appendix D.2, $\tilde{\nabla}\Phi$ is mentioned but not defined. It would be helpful to specify its definition.
>
> $\textbf{Response:}$
> Yes, your understanding is correct. As noted in the conclusions of the proofs of Theorems 3.3 and 3.6 (Lines 1669 and 1942 of the original version), $\Phi(x^*)$ can be replaced with any lower bound of $\Phi$. This correction has been incorporated into the revised version. To clarify, we have added the standard lower bound assumption for $\Phi(x)$ in the revised version.
>
> Thank you for pointing out this notation issue. $\tilde{\nabla} \Phi$ represents the approximate hypergradient, defined as $\tilde{\nabla} \Phi := \nabla_{x} F(x_k, y_{k+1}) - [\nabla^2_{xy} f(x_k, y_{k+1})]^T u_{k+1}$. This definition has been included in Line 165 of the revised paper.
>
> >In the experiments, how to select the step sizes and the number of inner iterations for the proposed algorithms?
>
> $\textbf{Response:}$ The implementations and hyperparameter settings for the proposed algorithms in all numerical experiments are detailed in Section C of the Appendix.
>
> Based on our experience, the warm-up step size $\beta$ for the inner loop $\mathcal{A}$ is typically set to 0.1. However, for logistic regression, a step size of $0.0001/(j+1)$ has been found to be more effective. The step size $\gamma$ was tuned within ${1, 0.5, 0.1}$. Specifically, we set $\gamma = 1$ in the toy example, while $\gamma = 0.1$ was more effective in other experiments. The number of inner iterations, $T$, significantly influences the parameters $\omega$ and $\tau$, which are critical for satisfying the convergence conditions. Adjustments to $T$ are made based on the condition number of the objective function in the bilevel problem. Experiments, including ablation studies on $T$ from the data hyper-cleaning experiment (Figure 9 in the Appendix of the original version), demonstrate that smaller values of $T$ lead to greater efficiency.
>
> >In Table 1, why does qNBO (BFGS) save so much time compared to SHINE for data hyper-cleaning?
>
> $\textbf{Response:}$
> The reason is that qBNO employs a constant step size strategy instead of the time-consuming line search used in SHINE for the inner solver. To clarify, consider the data hyper-cleaning experiment on the MNIST dataset. The SHINE-OPA algorithm requires 23.19 seconds to achieve a test accuracy of 91.51%. (The 20.25 seconds reported in Table 1 of the paper is the average time across ten runs.) Notably, the line search method in the lower-level solver accounts for 19.61 seconds of the total time.

---

> > ### Comment · Reviewer_ozni · 2024-11-27
> > **Reply to rebuttals**
> >
> > The authors totally deal with my concerns, so I keep my score.

---

> > > ### Author Response · Authors · 2024-12-04
> > >
> > > We are glad to have addressed your concerns. Thank you so much for your constructive suggestions and support!

---

### Official Review · Reviewer_yCCW · 2024-10-31

**Soundness:** 1
**Presentation:** 3
**Contribution:** 3
**Rating:** 5
**Confidence:** 5

**Summary:**

Summary:

The paper considers bilevel optimization problems with strongly convex lower-level objectives and proposes a framework for performing hypergradient approximation by using recursive quasi-Newton schemes to avoid issues in Hessian inversion arising in common BO algorithm.
The proposed algorithms, qNBO (BFGS) and qNBO (SR1), take inspiration from an existing algorithm 'SHINE' (Ramzi et al 2022). SHINE uses quasi-newton for solving the lower-level and reuses the approximate inverse hessian to perform hypergradient computation.
However, such approximate hessian is only accurate when multiplied with a specific vector: the gradient of the inner objective.
Hence, it might yield inaccurate solutions when multiplied by the gradient of the outer objective, as required by implicit differentiation. Although SHINE propose a way to steer the approximation towards the upper-level, it is unlear to what extend the method enjoys good quantitive convergence rates. This paper addresses such limitation by introducing a separate quasi-newton based estimation of the inverse-hessian along the gradient of the outer objective. This results, in principle, in a more accurate estimation of the hypergradient.
The paper then provides finite time convergence guarantees and illustrate the method on a toy example, a regression task and a meta-learning task.

**Strengths:**

- The motivation and approach in the paper are quite interesting, it makes sense to accelerate convergence of the inner problem using quasi-newton approaches. It is also nice that the method provide convergence rates for their algorithm, although I think the analysis do not seem to capture the benefits of qn for computing the hyper-gradient (when computing  the iterates u_k).
- The paper is overall clearly written and well explained, which makes it easy to read.

- The method seems to give quite an improvement compared to SHINE in terms of convergence speed (no full finite time convergence was provided for SHINE). While the experiments seem in favor of the proposed method, I think the implementation of SHINE contains some errors after checking the code (see weakness section).

- I appreciate that the authors provided the code to reproduce the results, this allows for a better assessment for the results. Although there were bugs in it, that unfortunately change some conclusions (see the weakness section), I think this should be seen as an opportunity to improve the experimental study.

**Weaknesses:**

- Experiments: The results in the toy experiment looked suspicious to me: AID-TN, AID-BIO and AmigoCG seemed to work unreasonably worse on a quite simple example where they are supposed to perform quite well. I decided to check the implementation provided in the supplementary and found a number of bugs that explain these results. Please refer to the questions section for the details of these bugs.
I think these can be easily fixed. However, after fixing these bugs, the results do not exactly match the conclusions of the current paper: it is possible to solve the toy problem faster/more accurately using AID-BIO/AmigoCG using a single step of CG (instead of 10). More precisely, after fixing the bugs: AID-BIO/AmigoCG reach 10^-4 relative error on the upper variable (instead the numbers reported in the paper: 10^-2 for AID-BIO and 10^0 for AmigoCG), and further go below 10^-5 (better than qNBO) when using only a single CG step and single gradient step. AID-TN reaches 10^-1, instead of 10^0. I do not exclude other bugs on other methods (the implementation of SHINE already looks incorrect (see questions section)), so I strongly encourage the authors to test the implementation with care.




- Convergence results:
While the convergence analysis exploits the super-linear convergence of quasi-newton for the inner-level iterates, the iterates used for computing the hyper-gradient do not seem to benefit from such super-linear rate. In fact, the convergence of these iterates is sub-linear: 1/Q where Q is the number of quasi-Newton iterates.  This might explain the worse dependence of the algorithm on the number of gradient queries on the lower loss epsilon^{-2} compared to AID-BIO epsilon^{-1}. Still there is an improved dependence on the conditioning for the vector-jacobian complexity (kappa^3) vs AID-BIO (kappa^{3.5}). However, it is unclear how this compensates the increased complexity for the gradient, especially given the results of the toy experiments (without the bugs).
On a positive note, the result is an improvement over the guarantees provided for SHINE as they are quantitative.

Ablation studies: methods such as AID-BIO do not exclude using other solvers for the lower-level problem. One could then use quasi-newton methods and still apply CG for the conjugate gradient. How do you think this would perform?


Although I think the proposed direction is interesting, there is a number of problems with the current version of the paper that need to be addressed. I have many reasons not to trust the empirical results and that would require checking all the details of the code for all experiments. It is unclear to me if the rebuttal period would be enough for that.

**Questions:**

Some of the figures in the appendix seem to be consitent with what the python file toy3.py produces. However, it contains a number of bugs for various implementations:
 1- AID-TN:  a) the implementation of the function TN is wrong, it should be:
	 	v = v - inner_lr * output
		return inner_lr * p
	instead of :
		v = v + inner_lr * output
		return p

	     b) the function TN (truncated-newton) should be called to compute v before computing the cross derivative f_xy, otherwise, one uses an outdated version of v for computing the hyper-gradient i.e.:
		v = TN(fyy,Fy)
            	fyx=f_xy(x,y,v)
	instead of:
		fyx=f_xy(x,y,v)
		v = TN(fyy,Fy)

 2- AID-BIO:  Same issue as in AID-TN: Conjugate gradient should be called before computing the cross-derivative for the same reasons, otherwise this results in unnecessarily inaccurate hyper-gradients which slow performance:
		v = cg(fyy,Fy,v,10)
		fyx=f_xy(x,y,v)
	instead of:
		fyx=f_xy(x,y,v)
		v = cg(fyy,Fy,v,10)
 3- AmigoCG: In the deterministic case, this corresponds exactly to AID-BIO algorithm, so they should give the exact outputs. Similarly to AID-BIO, cg should be called before the cross-derivative. Additionally, there is no need to evalulate f_x(x,y) and f_y(x,y) for hyper-gradient computation as this unnecessarily slows the method. More importantly, there should be a minus sign instead of a + in front of the implicit gradient:
		x_grad=Fx-w
	instead of:
		x_grad=Fx+w

 4- Shine: The implementation is also incorrect though I did not have time to fix it myself. The OPA correction involves only the cross derivatives of the inner-level function not the outer level function:
		ogrady = f_xy(x,y)
	instead of:
		ogrady = F_y(x,y)

---

> ### Author Response · Authors · 2024-11-19
> **The issues encountered with AID-TN, AID-BIO, and AMIGO-CG in the toy experiment do not impact the outcomes of other experiments.**
>
> We sincerely thank Reviewer yCCW for the thorough and valuable feedback on our code. Before addressing the issues identified with AID-TN, AID-BIO, and AMIGO-CG in the toy experiment, we first assessed their potential impact on other experiments.
>
> Our analysis indicates that these issues do not affect the outcomes of other experiments. This conclusion is based on the fact that $\textbf{the toy experiment and other experiments use separate algorithm files.}$ Specifically, all the code for the algorithms in the toy experiment is contained in the file ICLR2025\qNBO\qNBO\toy, which was rewritten exclusively for the toy experiment. Notably, the algorithms AID-TN, AID-BIO, and AMIGO-CG in the toy experiment utilize a closed-form Hessian (discussed later). In contrast, their implementations in other experiments rely on code from the file ICLR2025\other algorithms\solvers, which originates from the open-source implementation cited in the paper.
>
>
> The following provides the specific file paths for thoroughly checking the bugs mentioned by Reviewer yCCW for AID-TN, AID-BIO, and AMIGO-CG during the toy experiment. $\textbf{Notably, these bugs are absent from the codes in}$ ICLR2025\other algorithms\solvers $\textbf{used for other experiments.}$
>
> $\textbf{AID-TN:}$ From ICLR2025\other algorithms\solvers\stocbio.py (lines 84–95) and ICLR2025\other algorithms\benchmark_utils\hessian_approximation.py (lines 116–121), it can be verified that the code of AID-TN used in other experiments aligns with Reviewer yCCW’s debug on AID-TN for the toy experiment.
>
> $\textbf{AID-BIO:}$ From ICLR2025\other algorithms\solvers\stocbio.py (lines 84–95), it can be verified that the code of AID-BIO used in other experiments aligns with Reviewer yCCW’s debug on AID-BIO for the toy experiment.
>
> $\textbf{AMIGO-CG:}$ From ICLR2025\other algorithms\solvers\amigo.py (lines 87–98 and lines 99–102), it can be verified that the code of AMIGO-CG used in other experiments aligns with Reviewer yCCW’s debug on AMIGO-CG for the toy experiment.

---

> ### Author Response · Authors · 2024-11-19
> **Discussion on the impact of Reviewer yCCW’s debug in the toy experiment (Part 1/3).**
>
> We greatly appreciate Reviewer yCCW’s insightful debugging of AID-TN, AID-BIO, and AMIGO-CG in our toy experiment. The suggestions are both accurate and helpful, aligning with the code used in other experiments. Taking this opportunity, we further reviewed the toy experiment and identified two additional directions for improvement in the toy experiment.
>
> First, we need to fine-tune the hyperparameters of all algorithms (e.g., inner loop iterations and step sizes) to ensure a fair comparison. Previously, the inner loop setting (10 steps) for AID-TN, AID-BIO, and AMIGO-CG was adopted directly from other experiments without optimization.
>
> Second, in the toy experiment, we used the closed-form expression $f_{yy} = A$ for the Hessian in the implementation of AID-TN, AID-BIO, and AMIGO-CG. Specifically, the code ICLR2025\qNBO\qNBO\toy\toy3 (lines 750, 814, and 876) calls the function f_yy1 (lines 99–100 of toy3) to directly return the Hessian matrix $A$. However, a more practical approach would involve computing Hessian-vector products using torch.autograd.grad. It is important to note that only the implementations of AID-TN, AID-BIO, and AMIGO-CG require the Hessian.

---

> ### Author Response · Authors · 2024-11-19
> **Discussion on the impact of Reviewer yCCW’s debug in the toy experiment (Part 2/3).**
>
> To address these issues, we have re-conducted the toy experiment, as detailed below:
>
> (1)$\textbf{By retaining the closed-form expression $f_{yy} = A$ of the Hessian in the implementations of AID-TN, AID-BIO, and AMIGO-CG}$, we performed a grid search to fine-tune the hyperparameters of all algorithms, including inner loop iterations and step sizes. As noted by Reviewer yCCW, AID-BIO and AMIGO-CG achieve a relative error of $10^{-5}$ on the upper-level variable (compared to the values reported in the paper: $10^{-2}$ for AID-BIO and $10^0$ for AMIGO-CG) after setting the inner loop iterations for CG steps and gradient steps to 1 while maintaining the outer step size at 0.01.
>
> By further conducting a grid search over TN/CG iterations in the range [1, 5, 10, 15, 20], inner gradient iterations between 1 and 10, and step sizes within [0.1, 0.2, 0.3, 0.4, 0.5], $\textbf{the relative error of AID-BIO and AMIGO-CG can be reduced to below $10^{-6}$ within 1.5 seconds when the outer step size is adjusted to $\alpha = 0.2$}$. At the same time, after fine-tuning the hyperparameters of qNBO (SR1),  $\textbf{the new implementation of qNBO (SR1) can also reduce the error to below $10^{-6}$ within 1.5 seconds}$.
>
> The updated results are presented in Figure 1 of a single-page PDF (containing only figures and available at the anonymous link: https://drive.google.com/file/d/15x5Fp1XYlo1DKrKs4RcquXU6khqv9Y1l/view?usp=drive_link) and in the following tables.
>
> | Time (s) for \|x - x^*\| / \|x^*\| | 0.5      | 1.0      | 1.5      |
> |----------------------------------------|----------|----------|----------|
> | **qNBO(SR1)**                          | 1.8E-6   | 4.0E-7   | 4.3E-7   |
> | **qNBO(BFGS)**                         | 5.6E-3   | 2.8E-4   | 4.0E-5   |
> | **AID-BIO**                            | 9.2E-7   | 7.1E-7   | 7.1E-7   |
> | **AMIGO-CG**                           | 1.6E-5   | 7.1E-7   | 7.1E-7   |
> | **AID-TN**                             | 2.4E-1   | 2.4E-1   | 2.4E-1   |2.4E-1       |
>
>
> | Time (s) for \|y - y^*\| / \|y^*\| | 0.5       | 1.0       | 1.5       |
> |----------------------------------------|-----------|-----------|-----------|
> | **qNBO(SR1)**                          | 1.7E-06   | 7.2E-07   | 8.5E-07   |
> | **qNBO(BFGS)**                         | 7.9E-03   | 3.0E-04   | 6.3E-05   |
> | **AID-BIO**                            | 1.7E-06   | 1.1E-06   | 1.1E-06   |
> | **AMIGO-CG**                           | 2.7E-05   | 1.2E-06   | 1.2E-06   |
> | **AID-TN**                             | 3.1E-01   | 3.1E-01   | 3.1E-01   |
>
> | Time (s) for $\|d_x - \nabla \Phi\|$ | 0.5       | 1.0       | 1.5       |
> |----------------------------------------|-----------|-----------|-----------|
> | **qNBO(SR1)**                          | 1.4E-05   | 7.2E-06   | 8.5E-06   |
> | **qNBO(BFGS)**                         | 2.5E-02   | 1.2E-02   | 8.7E-03   |
> | **AID-BIO**                            | 1.9E-05   | 1.2E-05   | 1.2E-05   |
> | **AMIGO-CG**                           | 3.6E-04   | 1.2E-05   | 1.2E-05   |
> | **AID-TN**                             | 4.1E+00   | 4.3E+00   | 4.0E+00   |
>
>
> Note that the reason the performance of the AID-BIO and AMIGO-CG algorithms differs slightly in the updated results is due to incomplete memory release. To facilitate reproduction of the results, the specific hyperparameter settings are as follows:
>
> $\textbf{AID-BIO/AMIGO-CG:}$ The maximum number of outer iterations is K = 5000, the CG iterations is $N = 1$, the inner gradient iteration is $T = 1$, the inner step size is $\beta= 0.01$ and $\textbf{the outer step size is $\alpha = 0.2$}$.
>
> $\textbf{AID-TN:}$ The maximum number of outer iterations is $K = 5000$, the TN iteration is $N = 1$, the inner gradient iteration is $T = 1$, the inner step size is $\beta = 0.01$ and $\textbf{the outer step size is $\alpha = 0.2$}$.
>
> $\textbf{qNBO (SR1):}$ The maximum number of outer iterations is $K = 5000$, the inner iterations are $T = 6$, $P = 9$, $Q_k = 25$, the inner step sizes are $\beta = 0.01$, $\gamma = 1$, the initial matrix is $H_0 = I$, and the outer step size is $\alpha = 0.5$.

---

> ### Author Response · Authors · 2024-11-19
> **Discussion on the impact of Reviewer yCCW’s debug in the toy experiment (Part 3/3).**
>
> (2)Since the implementations of AID-TN, AID-BIO, and AMIGO-CG in the above experiments used the closed-form expression $f_{yy} = A$ for the Hessian, $\textbf{it would not be}$ $\textbf{ fair to compare them with other algorithms that do not use the explicit Hessian information.}$ Therefore, $\textbf{we also conducted the toy experiment}$  $\textbf{using the more practical torch.autograd.grad for computing Hessian-vector products.}$ Note that this approach is typical in the machine learning literature, as the closed form of the Hessian is generally difficult to compute in practical applications. After performing a grid search to fine-tune the hyperparameters for step sizes tested across [0.01, 0.02, 0.03, 0.04, 0.05, 0.06, 0.07, 0.08, 0.09, 0.1, 0.2, 0.3, 0.4, 0.5], TN/CG iterations evaluated at [1, 10, 20, 30, 40, 50], and inner gradient iterations between 1 and 10, the new results—using torch.autograd.grad for computing the Hessian-vector product instead of the closed Hessian form—are presented in Figure 2 of the single-page PDF at the anonymous link and in the following tables.
>
>
>
> | Time (s) for \|x - x^*\| / \|x^*\|  |  0.5   |  1.0   |  1.5    |
> |-----------------------------------------|------------|------------|------------|
> | **qNBO(SR1)**     | 1.0E-06    | 5.8E-07    | 1.0E-06    |
> | **qNBO(BFGS)**    | 5.0E-03    | 2.7E-04    | 4.5E-05    |
> | **BOME**          | 3.5E-01    | 3.3E-01    | 3.2E-01    |
> | **SHINE-OPA**     | 6.9E-02    | 5.5E-02    | 6.9E-02    |
> | **AID-BIO**       | 1.4E+00    | 7.1E-01    | 4.0E-01    |
> | **AMIGO-CG**      | 1.4E+00    | 7.1E-01    | 4.0E-01    |
> | **PZOBO**         | 3.4E+00    | 3.7E+00    | 4.1E+00    |
> | **F2SA**          | 3.5E-01    | 3.3E-01    | 3.2E-01    |
> | **AID-TN**        | 1.9E+00    | 1.0E+00    | 5.2E-01    |
> |
>
>
> | Time (s) for \|y - y^*\| / \|y^*\| | **0.5**    | **1.0**    | **1.5**    |
> |-------------------|------------|------------|------------|
> | **qNBO(SR1)**     | 1.0E-06    | 4.8E-07    | 2.3E-06    |
> | **qNBO(BFGS)**    | 7.9E-03    | 3.0E-04    | 6.3E-05    |
> | **BOME**          | 2.7E-01    | 2.7E-01    | 2.7E-01    |
> | **SHINE-OPA**     | 1.0E-01    | 4.8E-02    | 4.8E-02    |
> | **AID-BIO**       | 4.1E+00    | 3.3E+00    | 2.7E+00    |
> | **AMIGO-CG**      | 4.1E+00    | 3.3E+00    | 2.7E+00    |
> | **PZOBO**         | 4.1E+00    | 4.2E+00    | 4.5E+00    |
> | **F2SA**          | 5.2E-01    | 4.8E-01    | 4.4E-01    |
> | **AID-TN**        | 2.5E+00    | 1.3E+00    | 6.7E-01    |
>
>
>
>
>
>
>
>
>
>
>
>
>
>
> |Time (s) for $\|d_x - \nabla \Phi\|$  |  0.5    |  1.0    |  1.5    |
> |-------------------|------------|------------|------------|
> | **qNBO(SR1)**     | 8.6E-05    | 2.3E-05    | 7.0E-05    |
> | **qNBO(BFGS)**    | 4.3E-02    | 2.2E-02    | 1.6E-02    |
> | **BOME**          |     -       |         -   |       -     |
> | **SHINE-OPA**     | 4.3E+00    | 1.7E+00    | 1.9E+00    |
> | **AID-BIO**       | 5.6E+01    | 7.8E+01    | 6.9E+01    |
> | **AMIGO-CG**      | 5.6E+01    | 7.8E+01    | 8.9E+01    |
> | **PZOBO**         | 2.2E+02    | 2.2E+02    | 2.4E+02    |
> | **F2SA**          | 1.3E+01    | 1.1E+01    | 1.1E+01    |
> | **AID-TN**        | 3.0E+01    | 1.5E+01    | 9.1E+00    |
>
>
> These results show that: (i) for AID-BIO and AMIGO-CG, 50 CG iterations yielded the best results, while a single CG iteration caused divergence, resulting in much worse performance compared to the scenarios using the closed-form Hessian; (ii) compared to other algorithms, both qNBO (BFGS) and qNBO (SR1) exhibit smaller hypergradient errors and produce iterates that are closer to the optimal solutions. $\textbf{Therefore, after implementing the debugging suggestions from Reviewer yCCW and using the more practical torch.autograd.grad for}$
>
> $\textbf{Hessian-vector products, the results for the toy experiment align with the conclusions of this work.}$
>
> To facilitate the reproduction of the results, the specific hyperparameter settings are as follows:
>
> $\textbf{AID-BIO/AMIGO-CG:}$ The maximum number of outer iterations is $K = 5000$, CG iterations is $N = 50$, the inner gradient iteration is $T = 1$, the inner step size is $\beta = 0.01$, and the outer step size is $\alpha = 0.01$.
>
> $\textbf{AID-TN:}$ The maximum number of outer iterations is $K = 5000$, TN iterations is $N = 50$, the inner gradient iteration is $T = 1$, the inner step size is $\beta = 0.01$, and the outer step size is $\alpha = 0.2$.
>
> The hyperparameters of qNBO (SR1) are the same as those used in the above experiment, and the settings for the other algorithms remain unchanged from those in the paper.

---

> ### Author Response · Authors · 2024-11-19
> **The implementation of SHINE is correct.**
>
> SHINE incorporates two different OPA (Outer Problem Awareness) corrections as described in Ramzi et al. (2022). $\textbf{The first correction involves the cross-derivatives of the inner-level function $f_{xy}(x, y)$ when the upper-level variable $x$ is one dimentianal}$ (i.e., $f_{xy}(x, y)$ is a vector). This correction is outlined in equation (5) of Ramzi et al. (2022) and explained above it. It corresponds to the term $f_{xy}[f_{yy}]^{-1}$ in the computation of the hypergradient and requires $x$ to be one-dimensional.
>
> $\textbf{The second OPA correction, described in equation (8) of Ramzi et al. (2022), involves $F_y(x,y)$}$, which is a vector regardless of the dimensionality of $x$. This correction corresponds to $[f_{yy}]^{-1}F_y$ in the computation of the hypergradient. $\textbf{This is the version we used in both the toy experiment and other experiments.}$
>
> The specific file paths for verifying the implementation of SHINE are as follows:
>
> $\textbf{SHINE:}$ From ICLR2025\qNBO\qNBO\logisticregression\hoag\hoag.py (lines 308 and 383) and ICLR2025\qNBO\qNBO \logisticregression\hoag\lbfgs.py (line 84), it can be confirmed that the code for SHINE aligns with equation (8) of Ramzi et al. (2022). Additionally, it matches the original SHINE open-source code (lines 147–148 in     https://github.com/zaccharieramzi/hoag/blob/shine/hoag/hoag.py and lines 80–83 in https://github.com/zaccharieramzi/hoag/blob/shine/hoag/lbfgs.py).

---

> ### Author Response · Authors · 2024-11-19
> **Additional experiments to explore why 1 step CG iteration, as recommended by Reviewer yCCW, performs better than 10 steps.**
>
> We compared the performance of AID-BIO with different numbers of CG iterations ($T=1, 10$). The results are presented in Figure 3 of the single-page PDF at the anonymous link and in the following tables. Using the number of outer-loop iterations of the algorithm on the x-axis, we observed that both CG iteration strategies exhibited nearly identical performance. This suggests that, due to the closed form of the Hessian provided in the code, a single CG iteration is sufficient to reach the optimal solution. Additional CG iterations do not improve computational accuracy, making $T=1$ more time-efficient than $T=10$.
>
> | Iteration for \|x - x^*\| / \|x^*\| | T=1 | T=10 |
> |-----------|-------------|--------------|
> | 100       | 4.9E-01     | 4.9E-01      |
> | 200       | 4.3E-02     | 4.3E-02      |
> | 300       | 4.1E-03     | 4.2E-03      |
> | 400       | 4.2E-04     | 4.3E-04      |
> | 500       | 4.5E-05     | 4.5E-05      |
>
> | Iteration for \|y - y^*\| / \|y^*\|  | T=1  | T=10  |
> |-----------|-------------|--------------|
> | 100       | 6.3E-01     | 6.3E-01      |
> | 200       | 5.5E-02     | 5.6E-02      |
> | 300       | 5.4E-03     | 5.4E-03      |
> | 400       | 5.5E-04     | 5.6E-04      |
> | 500       | 5.8E-05     | 5.9E-05      |

---

> > ### Comment · Reviewer_yCCW · 2024-11-21
> > **Thank you for the response. However, there are still major flaws in the paper and the new response**
> >
> > Thank you for the response and the additional results. While I appreciate the effort, I do not find the response convincing, in particular the new experiments with Hessian-vector product. As detailed below.
> >
> > **The implementation of SHINE is correct**
> > That's good news, and it is good that the proposed method outperforms SHINE.
> >
> > **Impact of using Hessian-vector product using autograd**
> > While I agree that it makes more sense to use the autograd based implementation of the Hessian-vector product,
> > I am not convinced by the results presented in the anonymous link. In particular figure 2 using autograd Hessian.
> > Using autograd should not change the trajectories of the iterates (although it could affect speed).
> > This means that one doesn't suddenly need to use 50 CG steps instead of 1 just because autograd is used.
> > In fact, I have modified the code of the toy experiment to use autograd and get a small slow-down for AID-BIO and AMIGO-CG when using the same hyper-parameters as when the full hessian is computed.
> > Both methods are still quite fast and reach a lower error near 10^-6 within 1.5s.
> > This is in striking contrast with the results of figure 2 which shows an error of the order of 10^-1.
> > Autograd should not result in significant slow down here, so I suspect there is an error in the implementation.
> > Please find a code snipet for AID-BIO to use autograd in a separate comment.
> > Therefore, I maintain that the conclusions from the toy experiments, when using correct implementation, are different from those in the paper.
> > To me this suggests this example is not a good illustration for the strength of the proposed algorithm.
> >
> > **The reason why 1 CG iteartion was sufficient**
> > I do not agree with the explanation: "due to the closed form of the Hessian provided in the code". That's  not the reason why a single step was enough. It doesn't matter if one provides A directly to perform Hessian vector-product of the form A@x, or to perform autograd to compute this same vector product: this should result in the same number up to numerical precision.
> > The reason why a single step was sufficient is probably because the hessian is very-well conditionned  (conditioning number $\kappa$ close to 1) which implies that it requires few CG steps to converge, since the convergence rate is in $((\sqrt(\kappa)-1)/(\sqrt(\kappa)+1))^K$, so a few iterations (small K) results in a small error already.
> >
> >
> >
> >
> > **The other experiments are not impacted by the bugs in the toy experiments**
> > Also glad to see that this is not the case. However, all the other experiments are missing the AID-BIO/AMIGO-CG which are the strongest baseline according to the toy experiments.
> > Why are some algorithms missing in those comparisons? Why is the meta-learning experiment compares only with PZOBO?
> > The experiments might not be impacted by the bug, but there are still not very convincing due to the missing baselines.
> >
> > **Analysis**
> > Besides the numerical experiments, there is still an unsatisfactory aspect in the theoretical analysis which is not addressed by the authors:
> > The hyper-gradient do not seem to benefit from such super-linear rate, in fact the analysis suggests a very slow rate of 1/Q where Q is the number of quasi-Newton updates, which is even worse than gradient descent.
> > As a result, the final complexity has a degraded dependence in the gradient queries to the lower loss. It is unclear how this degraded complexity is compensated by the improved dependence on the conditioning for vector-jacobian products.

---

> > > ### Comment · Reviewer_yCCW · 2024-11-21
> > > **Code snippet using autograd**
> > >
> > > Code using autograd
> > >
> > > In the implementation of AID-BIO please replace the following:
> > >
> > > ~~~
> > >         fyy=f_yy1(x,y)
> > >         v = cg(fyy,Fy,v,1)
> > >         fyx= f_xy(x,y,v)
> > > ~~~
> > >
> > > by:
> > > ~~~
> > >         fyy_func=f_yy_func(x,y)
> > >         v= cg_func(fyy_func,Fy,v,1)
> > >         fyx_func = f_xy_func(x,y)
> > >         fyx= fyx_func(v)
> > > ~~~
> > >
> > > Here ```cg_func(A,b,x,num_steps)```   is similar to the implemented function ```cg(A,b,x,num_steps)``` , except that the first argument is now a function that takes a vector and performs Hessian-vector multiplication using autograd. This means that ```cg_func``` only needs to replace the two matrix-vector products ```A @ x``` and ```A @ p``` by evaluations ```A(x)``` and ```A(p)```.
> > >
> > > The functions ```f_yy_func``` and ```f_xy_func``` return function handles and are given by:
> > >
> > > ~~~
> > > def f_yy_func(x,y):
> > >
> > >     def f_yy(z):
> > >         val = f(x,y)
> > >
> > >         grad = autograd.grad(outputs=val,
> > >                             inputs=y,
> > >                             grad_outputs=None,
> > >                             retain_graph=True,
> > >                             create_graph=True,
> > >                             only_inputs=True,
> > >                             allow_unused=True)
> > >
> > >         hvp = autograd.grad(outputs=grad, inputs=y, grad_outputs=z)
> > >         return hvp[0]
> > >     return f_yy
> > > ~~~
> > >
> > > ~~~
> > > def f_xy_func(x,y):
> > >     def f_xy(z):
> > >         val = f(x,y)
> > >
> > >         grad = autograd.grad(outputs=val,
> > >                             inputs=y,
> > >                             grad_outputs=None,
> > >                             retain_graph=True,
> > >                             create_graph=True,
> > >                             only_inputs=True,
> > >                             allow_unused=True)
> > >
> > >         hvp = autograd.grad(outputs=grad, inputs=x, grad_outputs=z)
> > >         return hvp[0]
> > >     return f_xy
> > > ~~~

---

> > > ### Author Response · Authors · 2024-11-25
> > > **We greatly appreciate your thorough and valuable feedback. Below is our response to the concerns raised in your new comments (part 1/3).**
> > >
> > > >Hessian-vector product using autograd
> > >
> > > $\textbf{Response:}$
> > > Thank you very much for your reply. Upon carefully reviewing our code, we discovered an error in the iteration variable used during the CG iteration. Specifically, the original code was incorrectly updating the variable $x$ instead of $v$. After correcting this by updating $v$ instead of $x$, our results aligned perfectly with yours. We appreciate your valuable feedback.
> > >
> > > Below are the updated performance metrics:
> > >
> > > | Time (s) for $\parallel \mathbf{x} - \mathbf{x}^*\parallel / \parallel \mathbf{x}^*\parallel $ | 0.5      | 1.0      | 1.5      |
> > > |---------------------------------------------------------------|----------|----------|----------|
> > > | **qNBO (SR1)**                                                | 1.8E-6   | 4.0E-7   | 4.3E-7   |
> > > | **qNBO (BFGS)**                                               | 5.6E-3   | 2.8E-4   | 4.0E-5   |
> > > | **AID-BIO**                                                   | 1.0E-4   | 1.0E-6   | 1.0E-6   |
> > > | **AMIGO-CG**                                                  | 1.7E-4   | 1.0E-6   | 1.0E-6   |
> > >
> > > | Time (s) for $\parallel \mathbf{y} - \mathbf{y}^*\parallel / \parallel \mathbf{y}^*\parallel $ | 0.5       | 1.0       | 1.5       |
> > > |---------------------------------------------------------------|-----------|-----------|-----------|
> > > | **qNBO (SR1)**                                                | 1.7E-6    | 7.2E-7    | 8.5E-7    |
> > > | **qNBO (BFGS)**                                               | 7.9E-3    | 3.0E-4    | 6.3E-5    |
> > > | **AID-BIO**                                                   | 2.5E-4    | 1.0E-6    | 1.0E-6    |
> > > | **AMIGO-CG**                                                  | 4.1E-4    | 1.0E-6    | 1.0E-6    |
> > >
> > > | Time (s) for $\parallel d_x - \nabla \Phi\parallel $                          | 0.5       | 1.0       | 1.5       |
> > > |---------------------------------------------------------------|-----------|-----------|-----------|
> > > | **qNBO (SR1)**                                                | 1.4E-5    | 7.2E-6    | 8.5E-6    |
> > > | **qNBO (BFGS)**                                               | 2.5E-2    | 1.2E-2    | 8.7E-3    |
> > > | **AID-BIO**                                                   | 1.8E-3    | 8.0E-6    | 8.0E-6    |
> > > | **AMIGO-CG**                                                  | 2.9E-3    | 8.0E-6    | 8.0E-6    |
> > >
> > > We have updated the paper to incorporate these revised results. For more details, please refer to Figure 1 in the main text and the details on experiments section in Appendix C of the revised paper.

---

> > > ### Author Response · Authors · 2024-11-25
> > > **We greatly appreciate your thorough and valuable feedback. Below is our response to the concerns raised in your new comments (part 2/3).**
> > >
> > > > The reason why a single step was sufficient is probably because the hessian is very-well conditionned
> > >
> > >
> > > $\textbf{Response:}$
> > > Following your suggestion, we calculated the condition number of matrix $A$ and found it to be 1.98. To further investigate the impact of the condition number on the number of CG convergence steps, we selected condition numbers for $A$ as 10, 50, and 100. The results are as follows:
> > >
> > > $\kappa = 10$
> > >
> > > | Time (s) for $\parallel x - x^*\parallel / \parallel x^*\parallel$ | 0.5      | 1.0      | 1.5      |
> > > |----------------------------------------|----------|----------|----------|
> > > | **AID-BIO(1)**                   | 2.6E-4   | 1.0E-6   | 1.0E-6   |
> > > | **AID-BIO(5)**                   | 4E-2   | 7.4E-4   | 1.7E-6   |
> > >
> > > $\kappa = 50$
> > > | Time (s) for $\parallel x - x^* \parallel / \parallel x^*\parallel$ | 0.5      | 1.0      | 1.5      |
> > > |----------------------------------------|----------|----------|----------|
> > > | **AID-BIO(1)**                     | 1.2E-2   | 9.7E-4   | 7.8E-5 |
> > > | **AID-BIO(5)**                    | 3.2E-1   | 7.2E-2   | 8.3E-2  |
> > >
> > >
> > >
> > > $\kappa = 100$
> > > | Time (s) for $\parallel x - x^*\parallel / \parallel x^*\parallel$ | 0.5      | 1.0      | 1.5      |
> > > |----------------------------------------|----------|----------|----------|
> > > | **AID-BIO(1)**                     | 3E-3   | 7E-6   | 1.0E-6 |
> > > | **AID-BIO(5)**                    | 3E-2   | 7.2E-4   | 6.1E-5  |
> > >
> > >
> > > From the analysis above, it is evident that the condition number significantly impacts the efficiency of the algorithm. However, in most cases, using $P=1$ for the CG iteration remains the optimal choice.
> > > Your conjecture also inspired us to further explore comparisons between AID-BIO and qNBO (SR1) under larger condition numbers. The results indicate that qNBO (SR1) has a distinct advantage.
> > >
> > > Time (s) for $\parallel d_x - \nabla \Phi\parallel $ ($\kappa = 200$ ):
> > >
> > > | Method            | 0.5    | 1.0    | 1.5    |
> > > |-------------------|--------|--------|--------|
> > > | **AID-BIO(1)**    | 2.8    | 2.6    | 2.6    |
> > > | **AID-BIO(5)**    | 2.7    | 2.6    | 2.6    |
> > > | **qNBO (SR1)**           | 1.2E-2 | 7.8E-3 | 7.8E-3 |
> > >
> > > Please refer to the link for the corresponding image results. https://drive.google.com/file/d/11KDfI1dcpDmYpf6K1C12OKteC41NuPz5/view?usp=sharing
> > >
> > > The following is the hyperparameter selection of the algorithm:
> > >
> > > $\textbf{AID-BIO/AMIGO-CG:}$ The CG iterations is $N = 1$, the inner gradient iteration is $T = 1$, the inner step size is $\beta= 0.01$ and $\textbf{the outer step size is $\alpha = 0.01$}$.
> > >
> > > $\textbf{qNBO (SR1):}$ The inner iterations are $T = 14$, $P = 1$, $Q_k = 25$, the inner step sizes are $\beta = 0.01$, $\gamma = 1$, the initial matrix is $H_0 = I$, and the outer step size is $\alpha = 0.5$.
> > >
> > > We have updated all the codes and included them in the supplementary material. If you are interested, please feel free to download and review them. For toy experiments, refer to ICLR2025\qNBO\qNBO\toy\README.md. For logistic regression experiments, see ICLR2025\qNBO\qNBO\logisticregression\README.md. For data hyper-cleaning experiments, refer to ICLR2025\qNBO\qNBO\Dataclean\README.md.

---

> > > ### Author Response · Authors · 2024-11-25
> > > **We greatly appreciate your thorough and valuable feedback. Below is our response to the concerns raised in your new comments (part 3/3).**
> > >
> > > >However, all the other experiments are missing the AID-BIO/AMIGO-CG which are the strongest baseline according to the toy experiments. Why are some algorithms missing in those comparisons? Why is the meta-learning experiment compares only with PZOBO?
> > >
> > > $\textbf{Response:}$ Based on your suggestion, we included AID-BIO/AMIGO-CG as comparison algorithms in the logistic regression and data hyper-cleaning experiments. The results are as follows:
> > >
> > >
> > > MNIST (data hyper-cleaning)
> > > | Time (s) for test accuracy | 1     | 5    | 10    |
> > > |----------------------------------------|----------|----------|----------|
> > > | **AID-BIO**                          | 0.901  | 0.912   | 0.913   |
> > > | **BFGS**                         | 0.896   | 0.913   | 0.917   |
> > > | **SR1**                         | 0.899   | 0.912 | 0.914  |
> > >
> > > Fashion MNIST (data hyper-cleaning)
> > > | Time (s) for test accuracy | 1     | 5    | 10    |
> > > |----------------------------------------|----------|----------|----------|
> > > | **AID-BIO**                          | 0.818 | 0.825   | 0.823   |
> > > | **BFGS**                         | 0.826  |  0.829   | 0.83   |
> > > | **SR1**                         | 0.82   | 0.828  | 0.831   |
> > >
> > > Real-sim (logistic regression)
> > > | Time (s) for test loss | 1     | 2     | 5   |
> > > |----------------------------------------|----------|----------|----------|
> > > | **AID-BIO**                          | 8.1E2 | 6.4E2   | 6E2   |
> > > | **BFGS**                         | 1.7E3  |  3.8E2   | 2.1E2   |
> > > | **SR1**                         | 4.3E2  | 2.4E2  | 2.2E2   |
> > >
> > >
> > > 20news group (logistic regression)
> > > | Time (s) for test loss | 1     | 2     | 5   |
> > > |----------------------------------------|----------|----------|----------|
> > > | **AID-BIO**                          | 1.7E2 | 1.2E2   | 1.1E2   |
> > > | **BFGS**                         | 1.6E3  |  1.2E2   | 8.1E1   |
> > > | **SR1**                         | 3.3E2  | 2.7E2  | 2.2E2   |
> > >
> > > Similarly, we have updated the above results in the latest version of the paper. For more details, please refer to Figures 2-3 of the revised paper. Additionally, we would like to point out that the AMIGO-CG algorithm mentioned in the original paper has been corrected to the AMIGO algorithm, which uses the stochastic gradient descent method to update the auxiliary variable $v$. Since AID-BIO shares the same iteration format as the AMIGO-CG algorithm, we treat them as a single algorithm for comparison.
> > >
> > > Why the only competitor in the meta-learning experiment is PZOBO?
> > > (1)The comparison between PZOBO and MAML and ANIL is conducted in [1]. The results which are shown in Figure 5 of [1] indicate that PZOBO has better performances than MAML and ANIL.
> > > (2)In our research, we primarily focused on assessing whether qNBO could be applied to more complex machine learning experiments. Thus, in the meta-learning experiment, we only compared qNBO with PZOBO, currently recognized as the best algorithm [1] for this type of meta-learning. Due to the extensive duration required for meta-learning experiments and our constrained timeline, along with our focus on other research priorities, we did not explore these algorithms further. We hope the reviewer understand the constraints and focus of our study.
> > >
> > > [1] Sow D, Ji K, Liang Y. On the convergence theory for hessian-free bilevel algorithms[J]. Advances in Neural Information Processing Systems, 2022, 35: 4136-4149.
> > >
> > >
> > >
> > >
> > > >As a result, the final complexity has a degraded dependence in the gradient queries to the lower loss. It is unclear how this degraded complexity is compensated by the improved dependence on the conditioning for vector-jacobian products.
> > >
> > >
> > >
> > >
> > > $\textbf{Response:}$
> > > Thank you for your thoughtful feedback. Yes, your understanding is correct—there are unsatisfactory aspects in the theoretical analysis of this work, as stated in the theoretical comparisons at the end of Section 3. In theory, qNBO is not more efficient than AID-BIO in terms of $\epsilon$-order; it is only more efficient with respect to the $\kappa$ term within matrix-vector complexity. This degraded complexity appears to result from the cold-start strategy for $u_{k+1}$. Following Reviewer 5z6s’s suggestion to employ a warm-start strategy for $u_{k+1}$, we conducted additional experiments incorporating this approach. Details are provided in Figure 6 of the revised paper. The empirical results demonstrate that qNBO achieves better performance when the warm-start strategy is applied to $u$. A rigorous analysis would require a more detailed characterization of the relationships between quasi-Newton iterations and their sensitivity to variations in $(x, y)$, which we consider an interesting direction for future research.

---

> > > > ### Comment · Reviewer_yCCW · 2024-11-28
> > > > **Thank you. The response addresses my major concerns.**
> > > >
> > > > It is good to see that you have fixed the implementations. I have raised my score to 5 since I believe that the method is interesting, but that there is still some work to do in order to ensure the experimental comparaisons are correctly conducted. In particular:
> > > > - For the meta learning experiment, the justification for restricting the comparaison to PZOBO is unconvincing, especially that many of the considered methods appeared around the same time as PZOBO. The paper should at least compare with SHINE, since it is quite related.
> > > >
> > > > - Effect of conditioning: In the reported results comparing qNBO with AID-BIO for higher conditioning number, the number of iterations (T, N) are fixed to 1 for AID-BIO while T=15 and Q_k=25 for qNBO. Compared to the better conditionned problem, T was increased from 6 to 15 for qNBO, but not for  AID-BIO. How were these choices made? To have a fair comparaison, one most perform a grid search on these hyper-parameters for each method using a similar budget for each method. Was this the case? For instance, have you tried running AID-BIO with T=N=14 on this problem?

---

> > > > > ### Author Response · Authors · 2024-12-04
> > > > > **Thank you for your constructive feedback and for increasing the score. We have invested significant effort in conducting the meta-learning experiment. Below is our detailed response (part 1/2).**
> > > > >
> > > > > >Comment 1: Meta learning experiment
> > > > >
> > > > > $\textbf{Response:}$
> > > > > The meta-learning experiment was conducted to assess whether qNBO could be applied to more complex machine learning tasks and achieve comparable or superior performance. To the best of our knowledge, PZOBO is regarded as the baseline algorithm for this type of meta-learning experiment, while other algorithms developed after PZOBO have not been applied to such experiments. Consequently, we focused primarily on comparing qNBO with PZOBO across various datasets (miniImageNet, Omniglot, and FC100).
> > > > >
> > > > > We also attempted to test BOME but faced challenges when applying it to the miniImageNet dataset. The highest accuracy achieved by BOME was only 22.6%, compared to 59.8% for PZOBO and 62.1% for qNBO. Since BOME has not been previously applied to this type of meta-learning, its performance has not been publicly reported. Therefore, we chose not to include this result to maintain rigor.
> > > > >
> > > > > Thank you for your suggestion. We tested SHINE in the context of meta-learning. Due to the large size of the datasets used in our experiments (miniImageNet, FC100, and Omniglot—specific details are provided in Appendix C4.1, page 21 of the paper), we initially employed a constant step size of 1 for the inner iteration, as an alternative option in Algorithm 1 of SHINE, instead of the time-consuming line-search. For testing, we selected the smaller FC100 dataset, where the highest test accuracy achieved by SHINE was only 33% (compared to 47.01% for PZOBO and 47.3% for qNBO). Incorporating the line-search increased the highest test accuracy to 37.5%, but this approach was computationally expensive and still underperformed compared to our proposed algorithm. Since meta-learning experiments are more complex and involve larger datasets than data cleaning tasks, we dedicated considerable time to fine-tuning the hyperparameters and testing the experiments over the past few days. We appreciate your understanding.

---

> > > > > ### Author Response · Authors · 2024-12-04
> > > > > **Thank you for your constructive feedback and for increasing the score. We have invested significant effort in conducting the meta-learning experiment. Below is our detailed response (part 2/2).**
> > > > >
> > > > > >Comment 2: Effect of conditioning
> > > > >
> > > > > $\textbf{Response:}$
> > > > > In the experiment with a condition number $\kappa = 200$, we conducted a grid search for AID-BIO. The search range for the outer step size was [0.01, 0.02, 0.03, 0.04, 0.05, 0.1, 0.2, 0.3, 0.4, 0.5]. For the inner gradient iteration number $T$ and the conjugate gradient (CG) iteration number $N$, the search range was [1, 5, 10].
> > > > >
> > > > > Compared to other hyperparameter selections, $T = N = 1$ may exhibit a slight disadvantage in speed when the ordinate is $|| d_x - \nabla \Phi ||$ or $|| y - y^* || / || y^* ||$. However, it demonstrates significant advantages in both accuracy and speed when the ordinate is $|| x - x^* || / || x^* ||$. Below are some of the results:
> > > > > | Time (s) for $\parallel x - x^*\parallel / \parallel x^*\parallel$ | 0.5      | 1.0      | 1.5      |
> > > > > |-------------------------------------------------------------------|----------|----------|----------|
> > > > > | **T=N=1**                            | 2.6E-1   | 1.4E-2   | 2.1E-3   |
> > > > > | **T=1,N=10**                          | 2.8E-1  | 1.4E-2   | 1.0E-2   |
> > > > > | **T=10,N=1**                          | 4.8E-1   | 2.98E-1   | 1.9E-1   |
> > > > > | **T=10,N=10**                         | 1.9E-1   | 1.9E-1   | 1.9E-1   |
> > > > > | **qNBO(SR1)**                         | 8.7E-3   | 8.7E-3    | 8.7E-3  |
> > > > >
> > > > > | Time (s) for $\parallel y - y^*\parallel / \parallel y^*\parallel$ | 0.5      | 1.0      | 1.5      |
> > > > > |-------------------------------------------------------------------|----------|----------|----------|
> > > > > | **T=N=1**                            | 7.6E-1   | 6.8E-1    | 7.0E-1   |
> > > > > | **T=1,N=10**                          | 9.5E-1  | 7.1E-1   | 7.0E-1   |
> > > > > | **T=10,N=1**                          | 7.8E-1   | 6.8E-1    | 7.0E-1   |
> > > > > | **T=10,N=10**                         | 6.8E-1   | 6.8E-1   | 6.8E-1  |
> > > > > | **qNBO(SR1)**                         | 1.1E-2  |  1.1E-2    |  1.1E-2  |
> > > > >
> > > > > | Time (s) for $\|\| d_x-\nabla \Phi \|\|$   | 0.5      | 1.0      | 1.5      |
> > > > > |-------------------------------------------------------------------|----------|----------|----------|
> > > > > | **T=N=1**                            | 4.6E+00   | 2.7E+00   | 2.8E+00  |
> > > > > | **T=1,N=10**                          | 5.7E+00   | 3.1E+00  | 3.2E+00  |
> > > > > | **T=10,N=1**                          | 5.5E+00    | 2.6E+00  | 2.6E+00   |
> > > > > | **T=10,N=10**                         |  2.6E+00  |  2.6E+00 |  2.6E+00  |
> > > > > | **qNBO(SR1)**                         | 1.7E-1   | 1.7E-1   | 1.7E-1   |
> > > > >
> > > > > Therefore, overall, $T = N = 1$ is the optimal strategy. Based on your suggestion, we conducted an additional experiment with $T = N = 14$ under the condition number $\kappa = 200$. The results are presented below:
> > > > > | Time (s) for $\parallel x - x^*\parallel / \parallel x^*\parallel$ | 0.5      | 1.0      | 1.5      |
> > > > > |-------------------------------------------------------------------|----------|----------|----------|
> > > > > | **T=N=1**                            | 2.6E-1   | 1.4E-2   | 2.1E-3   |
> > > > > | **T=N=14**                            | 1.9E-1   | 1.9E-1   | 1.9E-1   |
> > > > > | **qNBO(SR1)**                         | 8.7E-3   | 8.7E-3    | 8.7E-3  |
> > > > >
> > > > > | Time (s) for $\parallel y - y^*\parallel / \parallel y^*\parallel$ | 0.5      | 1.0      | 1.5      |
> > > > > |-------------------------------------------------------------------|----------|----------|----------|
> > > > > | **T=N=1**                            | 7.6E-1   | 6.8E-1    | 7.0E-1   |
> > > > > | **T=N=14**                           | 6.8E-1   | 6.8E-1  |  6.8E-1   |
> > > > > | **qNBO(SR1)**                         | 1.1E-2  |  1.1E-2    |  1.1E-2  |
> > > > >
> > > > >
> > > > > | Time (s) for $\|\| d_x-\nabla \Phi \|\|$   | 0.5      | 1.0      | 1.5      |
> > > > > |----------------------------------------|----------|----------|----------|
> > > > > | **T=N=1**                      | 4.6E+00   | 2.7E+00   | 2.8E+00  |
> > > > > | **T=N=14**                   | 2.8E+00  | 2.7E+00   | 2.7E+00   |
> > > > > | **qNBO(SR1)**                      | 1.7E-1   | 1.7E-1   | 1.7E-1   |
> > > > >
> > > > > Due to the random matrix $A$ generated in the toy experiment, the results above slightly differ from the previous response in specific values, but the orders of magnitude remain the same. Additionally, in the previous response, the result for qNBO(SR1) with $\kappa = 200$ and $\|\| d_x-\nabla \Phi \|\|$ at $T = 14$ was reported as 1.2E-2 and 7.8E-3. However, due to an oversight, the correct values should have been 1.7E-1 (the results in the figure are correct), as shown above. We apologize for the confusion.

---

### Official Review · Reviewer_5z6s · 2024-10-31

**Soundness:** 3
**Presentation:** 3
**Contribution:** 3
**Rating:** 8
**Confidence:** 4

**Summary:**

This paper proposes an algorithmic framework, qNBO, which leverages quasi-Newton methods to solve bilevel optimization problems in the nonconvex-strongly-convex setting. Some practical implementations are also provided under the theoretical framework. It establishes non-asymptotic convergence for the BFGS adaptation within the framework and provides numerical experiments demonstrating the superior performance of the proposed algorithms across various machine-learning applications.

**Strengths:**

1. The paper is well written and easy to follow. Quasi-Newton type methods for solving bilevel optimization have not been well studied, even in the nonconvex-strongly-convex setting. This work provides a quite general algorithmic framework, allowing any quasi-Newton method to be applied in qNBO.
2. A convergence rate and complexity analysis are provided for qNBO (BFGS). Technical derivations seem to be nontrivial. The authors incorporate the superlinear convergence of BFGS into the non-asymptotic convergence framework commonly used in bilevel optimization literature.
3. Experiments validate the promise of the proposed methods.

**Weaknesses:**

1. For the theory, the results in Theorems 3.3 and 3.6 are limited to the setting where $Q_k=k+1$. As a result, qNBO is a double-loop algorithm. Is it possible to design a single-loop version? Recent progress has been made toward single-loop bilevel optimization algorithms, especially in the nonconvex-strongly-convex setting, by using a warm-start strategy.
2. For the experiments, since the value of $Q_k$ affects the running time, it would be beneficial to empirically demonstrate how increasing $Q_k$ influences performance gains. Note that the numerical results in Fig. 1(d) illustrate only the impact of $Q_k$ as the outer iteration $k$ varies.
3. Page 6: Do Theorems 3.3 and 3.6 require the assumption that an optimal solution $x^*$ exists? Note that $\Phi(x^*)$ appears in both (6) and (7).
4. Page 7: Why is it stated that qNBO is more efficient than AID-BIO? Note that the notation $\tilde{\mathcal{O}}$ omits the $\log\frac{1}{\epsilon}$ term.
5. In Appendix C.5, the authors report various ablation studies on the parameters $T$ and $Q$, but they do not specify which experiment these ablation studies are based on.

**Questions:**

Please see the questions in the weakness part. Here, I want to add some minor questions:

1. Page 6: In Eq. (6) of Theorem 3.3, the constant $M_{f_{xy}}$ can be computed, as $f$ takes the quadratic form given in (5).
2. Page 8: In Eq. (9), ``${a_i}’$, ${b_i}’$” should be ``$a’_i$, $b’_i$”.

I think this paper provides a good attempt to use second-order optimization to accelerate bilevel optimization. Overall, I like the approach. I am raising my score to 8 if the questions and the typos can be addressed well in the rebuttal.

---

> ### Author Response · Authors · 2024-11-25
> **Response to the concerns raised in Weaknesses.**
>
> >For the theory, the results in Theorems 3.3 and 3.6 are limited to the setting where $Q_k=k+1$. As a result, qNBO is a double-loop algorithm. Is it possible to design a single-loop version? Recent progress has been made toward single-loop bilevel optimization algorithms, especially in the nonconvex-strongly-convex setting, by using a warm-start strategy.
>
> $\textbf{Response:}$
> Thank you for the suggestion. A warm-start strategy is reasonable based on the Lipschitz continuity of $u^*(x, y)$ under the given assumptions. We have incorporated experiments using the warm-start strategy you suggested for $u$. As shown in Appendix C.1 (Figure 6 on Page 17) of the revised paper, the empirical results demonstrate that qNBO(BFGS) performs better when applying the warm-start strategy for $u_k$ (denoted as $Q_k = \text{ws} $ in the Figure 6). A rigorous analysis would require a more explicit characterization of the relationships between quasi-Newton iterations and their sensitivity with respect to $(x, y)$, which would be an interesting future extension. In our opinion, this could serve as a first step toward developing a single-loop version of qNBO. Thank you for your insightful question; we will continue to explore this idea.
>
>
>
> >For the experiments, since the value of $Q_k$ affects the running time, it would be beneficial to empirically demonstrate how increasing $Q_k$ influences performance gains. Note that the numerical results in Fig. 1(d) illustrate only the impact of $Q_k$ as the outer iteration $k$ varies.
>
> $\textbf{Response:}$
> Thank you for the suggestion. In the original submitted version (Figure 10 on Page 22 in Appendix C.5), an ablation study on the iteration number $Q_k$ of qNBO (BFGS) in the data hyper-cleaning experiment was performed using running times. We have added additional ablation study experiments based on running times for the toy experiment; see Appendix C.5.1 (Page 17, Figure 6) in the revised version. As shown in Figure 6, when $Q_k$ is constant, an increase in $Q_k$ initially slows the decline in various indicators over time but ultimately leads to a smaller minimal error. However, $Q_k = k+1$ demonstrates superior performance in terms of both the rate of decline and the minimal error achieved, even when its initial number of steps is fewer than that of constant $Q_k$. Furthermore, employing a warm-start strategy for $u_k$ improves the performance of qNBO (BFGS), highlighting the potential advantages of this approach, as noted by the reviewer.
>
>
>
> >Page 6: Do Theorems 3.3 and 3.6 require the assumption that an optimal solution $x^∗$ exists? Note that $\Phi(x^*)$ appears in both (6) and (7).
>
> $\textbf{Response:}$ Thank you for your questions. Theorems 3.3 and 3.6 do not assume the existence of optimal solutions. As noted at the conclusions of their proofs (Lines 1669 and 1942 of the original version), $\Phi(x^*)$ can be replaced with any lower bound of $\Phi$. Thus, we add the lower bound assumption on $\Phi(x)$ (Assumption 3.3 in the revised paper) and replace $\Phi(x^*)$ with $\inf_x \Phi(x)$ in Theorems 3.4 and 3.7 and the corresponding proofs in the appendix of the revised version.
>
> >Page 7: Why is it stated that qNBO is more efficient than AID-BIO? Note that the notation $\tilde{\mathcal{O}}$ omits the $\log(1/\epsilon)$ term.
>
> $\textbf{Response:}$ Thank you for your concern. Your understanding is correct—it is inaccurate to claim that qNBO is more efficient than AID-BIO in terms of matrix-vector complexity. According to Table 1 of AID-BIO (Ji et al., 2021), the number of Jacobian-vector products for AID-BIO is of the order $\mathcal{O}(\kappa^{3}\epsilon^{-1})$, and the number of Hessian-vector products is of the order $\mathcal{O}(\kappa^{3.5}\epsilon^{-1})$. When the dimension of the lower-level variable exceeds that of the upper-level variable (e.g., the model parameter’s dimension is larger than that of the hyperparameter), Hessian-vector products become more computationally intensive. Consequently, the matrix-vector complexity for AID-BIO is $\mathcal{O}(\kappa^{3.5}\epsilon^{-1})$, compared to $\tilde{\mathcal{O}}(\kappa^{3}\epsilon^{-1})$ for qNBO. Thus, qNBO is not more efficient than AID-BIO in terms of $\epsilon$-order; it is only more efficient in the $\kappa$ term within matrix-vector complexity. This correction has been included in the revised version.
>
> >In Appendix C.5, the authors report various ablation studies on the parameters $T$ and $Q$, but they do not specify which experiment these ablation studies are based on.
>
> $\textbf{Response:}$ The ablation studies described in Appendix C.5 are based on the data hyper-cleaning experiment. This information has been added to Appendix C.5.2 of the revised version of the paper. Thank you for bringing this to our attention.

---

> ### Author Response · Authors · 2024-11-25
> **Response to the concerns raised in  Questions.**
>
> >Page 6: In Eq. (6) of Theorem 3.3, the constant Mfxy can be computed, as f takes the quadratic form given in (5).
>
> $\textbf{Response:}$ Thank you for your suggestion. Since $\nabla^2_{xy} f(x, y) = -I$ for the lower-level objective function given in Eq. (5), the constant $M_{f_{xy}} = 1$. We have made the corresponding revisions in the paper.
>
>
> >Page 8: In Eq. (9), ${a_i}’$, ${b_i}’$” should be ai′, bi′”.
>
> $\textbf{Response:}$
> Thank you for pointing this out. The corrections to Eq. (9) have been made in the revised version.

---

> > ### Comment · Reviewer_5z6s · 2024-12-01
> > **Thanks for the response**
> >
> > I thank the authors for the detailed responses. I went through the responses and other reviewers' comments. I am satisfied with the answers to my questions, so I increase my score accordingly.
> >
> > Best,
> > Reviewer

---

> > > ### Author Response · Authors · 2024-12-04
> > >
> > > Thank you so much for your support! We appreciate your constructive suggestions in enhancing the quality of our work.

---

### Official Review · Reviewer_E1d6 · 2024-11-04

**Soundness:** 2
**Presentation:** 2
**Contribution:** 3
**Rating:** 5
**Confidence:** 3

**Summary:**

This paper integrates a quasi-Newton approach into a bilevel optimization algorithm to approximate the inverse Hessian-vector product.

**Strengths:**

The proposed method avoids costly second-order computations for approximating the Hessian-vector product while achieving a comparable convergence rate.

**Weaknesses:**

The paper is challenging to follow, with a vague explanation of the quasi-Newton recursion scheme. The design of the proposed method appears overly complex compared to existing methods, and there is limited explanation regarding the validity of  $u_{k+1}$ as an accurate approximation for the Hessian-vector product.

**Questions:**

Can this method be effectively applied in a stochastic setting while maintaining comparable convergence rates and computational efficiency? The required adjustments in iteration rounds and step size suggest potential difficulties in achieving the same convergence speed as existing methods.

---

> ### Author Response · Authors · 2024-11-25
> **Response to the concerns raised regarding the design of the proposed method and its stochastic adaptation (part 1/2).**
>
> >Weaknesses: The paper is challenging to follow, with a vague explanation of the quasi-Newton recursion scheme. The design of the proposed method appears overly complex compared to existing methods, and there is limited explanation regarding the validity of $u_{k+1}$ as an accurate approximation for the Hessian-vector product.
>
> $\textbf{Response:}$
> Thank you for your comment. Appendix B provides a comprehensive explanation of the quasi-Newton recursion scheme, including the algorithmic framework and key details omitted from the main text for brevity. Below, we elaborate on the necessity of subroutine $\mathcal{B}$ and and present explicit lemmas to justify why $u_{k+1}$ serves as an accurate approximation of the Hessian-vector product.
>
> In quasi-Newton methods, the convergence of the quasi-Newton matrix  $H^{-1}$  to the true Hessian matrix  $\nabla_{yy} f$  depends on strong assumptions [1], which are often not satisfied in practice [2]. Nevertheless, even without these assumptions, the quasi-Newton matrix  $H^{-1}$  can still converge to the true Hessian  $\nabla_{yy} f$  along specific directions [1,3]. Accordingly, in Step 2 of Algorithm 1, we apply the quasi-Newton recursion along the direction  $\nabla_y F$  to compute  $u_{k+1}$ , which serves as an approximation of the Hessian-vector product  $[\nabla_{yy} f]^{-1} \nabla_y F$ . Recent studies [4,5,6] have explicitly characterized the accuracy of the quasi-Newton matrix in approximating the Hessian along specific directions. Building on these findings, Lemmas D.16 (Lemma D.18 in the revised paper) and D.21 (Lemma D.23 in the revised paper）in Appendix D demonstrate how  $u = H^{-1} \nabla_y F$  effectively approximates  $[\nabla_{yy} f]^{-1} \nabla_y F$ .
>
>
> [1] Nocedal, Jorge, and Stephen J. Wright, Numerical optimization[M]. New York, NY: Springer New York, 1999.
>
>
> [2] Mannel, F. On the convergence of the Broyden-like matrices[J]. 2020.
>
> [3] Dennis J E, Moré J J. A characterization of superlinear convergence and its application to quasi-Newton methods[J]. Mathematics of computation, 1974, 28(126): 549-560.
>
> [4] Rodomanov A, Nesterov Y. Rates of superlinear convergence for classical quasi-Newton methods[J]. Mathematical Programming, 2022: 1-32.
>
> [5] Rodomanov A, Nesterov Y. New results on superlinear convergence of classical quasi-Newton methods[J]. Journal of optimization theory and applications, 2021, 188: 744-769.
>
> [6] Jin Q, Mokhtari A. Non-asymptotic superlinear convergence of standard quasi-Newton methods[J]. Mathematical Programming, 2023, 200(1): 425-473.

---

> ### Author Response · Authors · 2024-11-25
> **Response to the concerns raised regarding the design of the proposed method and its stochastic adaptation (part 2/2).**
>
> >Questions: Can this method be effectively applied in a stochastic setting while maintaining comparable convergence rates and computational efficiency? The required adjustments in iteration rounds and step size suggest potential difficulties in achieving the same convergence speed as existing methods.
>
> $\textbf{Response:}$
> Thank you for your thoughtful questions. This work introduces a flexible algorithmic framework, qNBO, designed to improve hypergradient approximation. qNBO incorporates a generic decoupling structure, wherein all search directions are linear with respect to the objective functions. This structure builds upon existing methods such as HOAG (Pedregosa, 2016), AID-BIO (Ji et al., 2021), AMIGO (Arbel & Mairal, 2022), and SOBA/SABA (Dagréou et al., 2022), enabling qNBO to extend effectively from deterministic to stochastic settings.
>
>
> To elaborate, qNBO consists of three components, with the first two employing quasi-Newton recursion schemes. A straightforward stochastic adaptation involves replacing deterministic quasi-Newton recursion schemes with stochastic variants, such as K-BFGS [1] or Stochastic Block BFGS [2]. Additionally, in Part 3 of qNBO, deterministic gradients are replaced with stochastic gradients throughout the iterative process, consistent with the transition from deterministic to stochastic methods discussed by Dagréou et al. (2022).
>
> However, this straightforward extension may compromise convergence rates and computational efficiency under the same smoothness assumptions as in the deterministic setting. For an intuitive comparison (without using quasi-Newton methods), refer to the analysis of SOBA (Dagréou et al., 2022) and its improvement by MA-SOBA (Chen et al., 2023, Optimal Algorithms for Stochastic Bilevel Optimization under
> Relaxed Smoothness Conditions).
>
> Addressing your question about maintaining comparable convergence rates and computational efficiency in a stochastic setting, we consider this a promising research direction. The main challenges in achieving this include:
>
> $\textbf{Constructing effective estimators:}$ How can we develop unbiased or biased estimators in Part 3 of qNBO by integrating techniques such as variance reduction or momentum (e.g., moving averages)? Potential candidates for these estimators include SGD, SAGA, SARAH, STORM, PAGE, and even Adam. However, analyzing this extension, particularly its convergence properties, requires significant effort. For recent progress (without using quasi-Newton methods), see SRBA (Dagréou et al., 2024) and SPABA (Chu et al., 2024).
>
> $\textbf{Analyzing convergence rates and complexity:}$ How can we evaluate the proposed stochastic algorithms in a bilevel setting while incorporating noisy second-order (curvature) information?Addressing these difficulties may require theoretical breakthroughs that go beyond existing first-order techniques, leaving this an unresolved challenge and an interesting future extension.
>
>
> [1] Goldfarb, D., Ren, Y., & Bahamou, A. (2020). Practical quasi-newton methods for training deep neural networks.
>
> [2] Gower, R., Goldfarb, D., & Richtárik, P. (2016). Stochastic block BFGS: Squeezing more curvature out of data.

---

### Meta-Review · Area_Chair_zoCa · 2024-12-20

**Metareview:**

This paper introduces qNBO, a novel bilevel optimization framework that leverages quasi-Newton methods to enhance hypergradient approximation. The authors provide a  convergence analysis and demonstrate strong empirical performance across various machine learning tasks. While some areas require further investigation, such as the stability of the SR1 method and the stochastic setting, the paper's innovative approach and promising results justify its acceptance.

**Additional Comments On Reviewer Discussion:**

There was lengthy back and forth with Reviewer yCCW who had identified some implementation issues, and helped solve it with the authors. This was tremendously helpful.

---

### Decision · Program_Chairs · 2025-01-22

Accept (Poster)